# Bayesian decision-making under misspecified priors with applications to meta-learning

**Max Simchowitz**[*]      Christopher Tosh[†]      Akshay Krishnamurthy[‡]      Daniel Hsu[†]
Thodoris Lykouris[§]      Miroslav Dudík[‡]      Robert Schapire[‡]

## Abstract

Thompson sampling and other Bayesian sequential decision-making algorithms are among the most popular approaches to tackle explore/exploit trade-offs in (contextual) bandits. The choice of prior in these algorithms offers flexibility to encode domain knowledge but can also lead to poor performance when misspecified. In this paper, we demonstrate that performance degrades gracefully with misspecification. We prove that the expected reward accrued by Thompson sampling (TS) with a misspecified prior differs by at most $\tilde{O}(H^2\epsilon)$ from TS with a well-specified prior, where $\epsilon$ is the total-variation distance between priors and $H$ is the learning horizon.

Our bound does not require the prior to have any parametric form. For priors with bounded support, our bound is independent of the cardinality or structure of the action space, and we show that it is tight up to universal constants in the worst case.

Building on our sensitivity analysis, we establish generic PAC guarantees for algorithms in the recently studied Bayesian meta-learning setting and derive corollaries for various families of priors. Our results generalize along two axes: (1) they apply to a broader family of Bayesian decision-making algorithms, including a Monte-Carlo implementation of the knowledge gradient algorithm (KG), and (2) they apply to Bayesian POMDPs, the most general Bayesian decision-making setting, encompassing contextual bandits as a special case. Through numerical simulations, we illustrate how prior misspecification and the deployment of one-step look-ahead (as in KG) can impact the convergence of meta-learning in multi-armed and contextual bandits with structured and correlated priors.

## 1   Introduction

Bayesian decision-making algorithms are widely popular, due to both strong empirical performance and the flexibility afforded by incorporating inductive biases and domain knowledge through priors. However, in practical applications, any chosen prior is at best an approximation of the true environment in which the algorithm is deployed. This raises a critical question:

*How sensitive are Bayesian decision-making algorithms to prior misspecification?*

For decision-making problems with a very large horizon, it suffices that the misspecified prior places a vanishingly small probability mass on the ground truth environment; this condition is referred to informally as a "grain of truth." This is because, in the large-horizon limit, Bayesian algorithms (like many non-Bayesian methods) should converge to the optimal policy.

---

[*]Massachusetts Institute of Technology, msimchow@mit.edu

[†]Columbia University

[‡]Microsoft Research NYC

[§]Massachusetts Institute of Technology

35th Conference on Neural Information Processing Systems (NeurIPS 2021).

But in many practical settings, decision-making takes place on shorter time scales. Consider a news recommendation website that, when presented with a new user, sequentially offers a selection of currently trending articles. Such a system may only have a few opportunities to make recommendations before the user decides to navigate away, leaving little time to correct for misspecified or underspecified prior knowledge. Such examples are described more broadly by the meta-learning paradigm, where a single learning agent must complete multiple disparate-though-related tasks.

In meta-learning problems, and in short-horizon problems more broadly, the "grain of truth" argument paints a rather uninformative picture. Consequently, recent work has begun to explore sensitivity bounds in applications with shorter horizons [LL16, KKZ$^+$21]. However, these recent works focus on particular classes of priors and/or reward models, as well as on the Thompson sampling algorithm specifically. Notably, this leaves open questions about the extent to which prior sensitivity is determined by properties of the Bayesian decision-making algorithm, the reward model, and the prior itself.

## 1.1    Our Contributions

Motivated by meta-learning problems with short task horizons, we establish general, distribution-independent, and worst-case optimal bounds on the sensitivity of Bayesian algorithms to prior misspecification. We focus on the Bayesian bandit setting, where a mean-vector "environment" $\boldsymbol{\mu}$ is drawn from a distribution $P$, and rewards for each action are drawn in accordance with $\boldsymbol{\mu}$. We study the performance of Bayesian algorithms which operate according to a *misspecified* prior $P'$.

**Sensitivity of Thompson Sampling and Related Bayesian Bandit Algorithms.**    As a concrete example, we consider the expected reward obtained by Thompson sampling with misspecified prior $P'$ under environments drawn from true prior $P$.

When the mean rewards lie in the range $[0, 1]$, as in the Bernoulli reward setting, we show that the difference in expected reward between Thompson sampling with $P'$ and with $P$ is at most twice the total variation distance between $P$ and $P'$ multiplied by the *square* of the horizon length. We prove a lower bound demonstrating that, for worst-case priors, this result is tight up to constants. Moreover, our upper bound holds for any two priors $P$ and $P'$ and suffers no dependence on the complexity of the decision space.

We extend this result in two directions. First, we remove the boundedness requirement on the mean reward range, showing that so long as certain tail probability conditions on the prior means are satisfied, a similar result holds. Second, we generalize beyond Thompson sampling, bounding the prior sensitivity of a broad class of Bayesian bandit algorithms, which we term $n$-*Monte Carlo algorithms*. Our lower bounds extend to this class, verifying sharp dependence on the parameter $n$.

**Sample Complexity of Bayesian Meta-Learning.**    We apply our prior sensitivity results to the Bayesian bandit meta-learning setting, in which a meta-learner iteratively interacts on bandit instances that are sampled from an unknown prior distribution. Motivated by our sensitivity analysis we describe a generic algorithmic recipe for Bayesian meta-learning, in which the meta-learner explores for several episodes to estimate the prior and then exploits by instantiating a Bayesian decision-maker with the learned prior. We formally consider two instantiations of this setup: (1) the Beta-Bernoulli setting where the rewards are Bernoulli and the prior is a product of Beta distributions and (2) the Gaussian-Gaussian setting where the rewards are Gaussian and the prior is a Gaussian (with arbitrary covariance structure) over the means. We note that the Gaussian-Gaussian setting was recently studied in [KKZ$^+$21] but only for the diagonal covariance setting.

**Bayesian Decision-Making Beyond the Bandit Setting.**    A striking feature of our proof is that it makes no explicit reference to the structure of bandit decision-making. As a consequence, our results extend seamlessly to both contextual bandits and the most general Bayesian decision-making problem: Bayesian POMDPs. While our sensitivity bounds hold almost verbatim in these settings, we note that estimating the prior may be statistically much more challenging in these scenarios, so there is no free lunch. To facilitate readability of the paper, we defer all further discussion and formal results to Appendix E.

**Experimental results.**    We complement our meta-learning theory with synthetic experiments in multi-armed and contextual bandit settings. Our experiments show the benefits of (a) meta-learning broadly, (b) estimating higher-order moments of the prior distribution, and (c) using less myopic

algorithms like the Knowledge Gradient [RPF12] over Thompson sampling when faced with structured environments.

## 1.2 Related Work

**Bayesian Decision-Making.** Bayesian decision-making broadly refers to a class of algorithms that use Bayesian methods to estimate various problem parameters, and then derive decision/allocation rules from these estimates. The study of Bayesian decision-making began with the seminal work of Thompson [Tho33], who introduced the Thompson sampling algorithm for adaptive experiment design in clinical trials. Thompson sampling later gained popularity in the reinforcement learning community as a means to solve multi-armed bandit and tabular reinforcement learning problems [Str00, OVR17], and has been extended in many directions [AL17, AL18, GMM14]. Recent years have seen the proliferation of other Bayesian decision-making and learning algorithms, including Information Directed Sampling [RVR16], Top-Two Thompson Sampling [Rus16], and Knowledge Gradient [RPF12].

**Sensitivity Analysis and Frequentist Regret.** The field of robust Bayesian analysis examines the sensitivity of Bayesian inference to prior and model misspecification (c.f., [BMP+94]). These approaches typically do not consider decision-making, so they do not account for multi-step adaptive sampling inherent in our setting. More recent works study frequentist regret for Thompson sampling [AG12, KKM12]. These guarantees can be interpreted as controlling the sensitivity to arbitrary degrees of prior misspecification, but consequently, they do not provide a precise picture of how misspecification affects performance. Moreover, frequentist guarantees for Thompson sampling focus on relatively long learning horizons, so they are less relevant in the context of meta-learning with many short-horizon tasks.

**Short-Horizon Sensitivity.** Most closely related to our paper are two previous works on sensitivity of Thompson sampling to small amounts of misspecification in short-horizon settings. [LL16] study the sensitivity of Thompson sampling for two-armed bandits when the prior has finite support. More recently, [KKZ+21] study meta-learning with Thompson sampling and derive sensitivity bounds for Thompson sampling in multi-armed bandits with Gaussian rewards and independent-across-arm Gaussian priors. In contrast to both of these works, the bounds presented in this work apply to arbitrary families of priors, more general decision-making problems, and to more general families of decision-making algorithms. Further, as illustrated in Remark 2, our bounds are also tighter than those achieved by [KKZ+21] when specialized to their precise setting. Finally, our lower bounds demonstrate that the square-horizon factor incurred in [KKZ+21] is unavoidable for worst-case priors (though perhaps not for their special case).

[BSLZ19] study sensitivity of general Bayesian algorithms in a dynamic pricing context. Their approach requires "sufficiently random" reward noise and applies only to algorithms with a non-adaptive initial exploration phase (unlike true Thompson Sampling); under these conditions, they show that the trajectories under a well-specified and misspecified Bayesian decision maker can be coupled so that the two algorithms maintain the same posteriors with good probability. In contrast, our analysis applies only to Bayesian decision-making algorithms which have sufficient "internal randomness" (e.g., Thompson sampling, and more generally, the $n$-Monte Carlo algorithms). Our approach obviates assumptions about reward noise and initial exploration at the expense of slightly restricting the class of algorithms to which our guarantees apply.

**Meta-learning and Meta-RL.** Meta-learning is a classical learning paradigm in which a learner faces many distinct-but-related tasks [Thr96, Thr98, Bax98, Bax00, HYC01]. While the classical work primarily considered supervised learning tasks, recent, predominantly empirical work has focused on meta-reinforcement learning (Meta-RL), where each task is itself a decision-making problem (c.f., [WKNT+17, DSC+16]). This includes some Bayesian approaches [HGH+20]. While there have been some theoretical results on Meta-RL [ALB13, CLP20, YHLD21, HCJ+21], apart from [KKZ+21] we are not aware of other theoretical treatments with a Bayesian flavor.

**Learning under model misspecification.** This work studies a specific notion of misspecification: running Bayesian decision-making algorithms with inexact approximations of a true underlying prior. Numerous other types of mispecification have been considered by the learning theory community more

broadly, although typically in the absence of meta-learning and in frequentist settings. These include models where rewards may be changing with time [GM11, BGZ14, CSLZ18, WL21] or adversarially corrupted [LMPL18, GKT19, ZS21], or where a simple function class (e.g., linear models) is used to approximate a more complex reward function [DKWY20, VRD19, LSW20, FGMZ20]. The Bayesian analog of these models is that the *likelihood* is misspecified, which is quite different from the prior misspecification considered here. Translating these notions of misspecification to Bayesian decision-making and unifying these lines of work remains an exciting direction for future research.

## 2 Setting and Notation

Throughout, we use bold $\mathbf{v}$ to denote vectors and non-bold $v_a$ to denote scalars. When the vector $\mathbf{v}_h$ has a subscript, $v_{h,a}$ denotes its coordinates.

**Bayesian Bandit Learning under Misspecification.** A Bayesian bandit learning instance is specified by (a) an abstract action space $\mathcal{A}$, (b) a parametric family of priors $P_\theta$ indexed by parameters $\theta \in \Theta$ over mean vectors $\boldsymbol{\mu} \in \mathbb{R}^{\mathcal{A}}$ with coordinates $\mu_a$, and (c) a function $\mathcal{D} : \mathbb{R}^{\mathcal{A}} \to \Delta(\mathbb{R}^{\mathcal{A}})$ mapping mean vectors $\boldsymbol{\mu}$ to distributions over reward vectors $\mathbf{r} \in \mathbb{R}^{\mathcal{A}}$ such that the expected reward under $\mathcal{D}(\boldsymbol{\mu})$ is $\boldsymbol{\mu}$, i.e., $\mathbb{E}_{\mathbf{r} \sim \mathcal{D}(\boldsymbol{\mu})}[\mathbf{r}] = \boldsymbol{\mu}$.[5] Note that this general setup allows the prior $P_\theta$ to encode complex dependencies between the mean rewards $\mu_a$ of actions $a \in \mathcal{A}$.

We consider an episodic bandit protocol with horizon $H$. First, $\boldsymbol{\mu} \sim P_\theta$ is drawn from the prior. Then, at each time step $h = 1, 2, \ldots, H$, the learner's policy, specified by an algorithm alg, selects an action $a_h \in \mathcal{A}$. Simultaneously, a reward vector $\mathbf{r}_h$ is drawn independently from $\mathcal{D}(\boldsymbol{\mu})$, and the learner observes reward $r_h = r_{h,a_h}$. The choice of action $a_h$ may depend on the partial trajectory $\tau_{h-1} = (a_1, r_1, \ldots, a_{h-1}, r_{h-1})$. We let $P_{\theta,\text{alg}}$ denote the joint law over $\boldsymbol{\mu}$ and the full trajectory $\tau_H$, while expectations are denoted $E_{\theta,\text{alg}}$. We abbreviate the full trajectory $\tau = \tau_H$. We denote the cumulative reward

$$R(\theta, \text{alg}) := E_{\theta,\text{alg}}\left[\sum_{h=1}^H r_h\right] = E_{\theta,\text{alg}}\left[\sum_{h=1}^H \mu_{a_h}\right].$$

**Bayesian Learning Algorithms.** We study a class of algorithms $\text{alg}(\theta)$ also parameterized by $\theta \in \Theta$. For concreteness, the reader may think of $\text{alg}(\theta)$ as corresponding to Thompson sampling, where the learner internally computes posteriors using $P_\theta$ as its prior. More general classes of Bayesian algorithms are defined in Section 3.1. We are interested in the consequences of misspecification; that is, interacting with $\boldsymbol{\mu} \sim P_\theta$, but executing $\text{alg}(\theta')$ for some other $\theta' \neq \theta$. Note that our notation for the induced law on the trajectory is $P_{\theta,\text{alg}(\theta')}$.

**Episodic Bayesian Meta-Learning.** We apply the above framework to the problem of Bayesian meta-learning. Let $\theta^\star \in \Theta$ be a ground-truth parameter. At each episode $t = 1, 2, \ldots, T$, a mean parameter $\boldsymbol{\mu}^{(t)}$ is drawn i.i.d. from $P_{\theta^\star}$. Simultaneously, the learner commits to a (potentially non-Bayesian) exploration strategy $\text{explore}^{(t)}$ and collects the induced trajectory $\tau^{(t)}$. At the end of $T$ episodes, the learner selects a parameter $\hat{\theta} \in \Theta$ as a function of $\tau^{(1)}, \ldots, \tau^{(T)}$. The learner's performance is evaluated on the expected reward of the plug-in algorithm on $\hat{\theta}$: $R(\theta^\star, \text{alg}(\hat{\theta}))$.

**Further notation.** Given two probability distributions $P$ and $Q$ over the same probability space $(\Omega, \mathcal{F})$, we denote their total variation $\text{TV}(P \parallel Q) := \sup_{\mathcal{E} \in \mathcal{F}} |P[\mathcal{E}] - Q[\mathcal{E}]|$ and Kullback-Leibler divergence $\text{KL}(P \parallel Q)$. If $P$ is a joint distribution of random variables $(X, Y, Z, \ldots)$, $P(X)$ denotes the marginal of $X$ under $P$, and $P(Y|X)$ the conditional distribution (as a function of random variable $X$). We define the diameter of a mean vector as $\text{diam}(\boldsymbol{\mu}) := \sup_{a \in \mathcal{A}} \mu_a - \inf_{a \in \mathcal{A}} \mu_a$, which is a random variable when $\boldsymbol{\mu}$ is drawn from $P_\theta$. Throughout, $\log(\cdot)$ denotes the natural logarithm. Given a space $\mathcal{X}$, we let $\Delta(\mathcal{X})$ be the set of probability distributions on $\mathcal{X}$; see Appendix B.1 for measure-theoretic considerations.

---

[5] In fact, our analysis extends to more general cases where the reward distribution is parameterized by more than just the mean vectors, but we restrict ourselves to the current setting for ease of exposition.

# 3 Prior Sensitivity in Bayesian Learning

This section states sensitivity bounds for various Bayesian bandit algorithms and families of priors, starting with the concrete instance of Thompson sampling under priors with bounded-range means. Our results extend almost verbatim to more general decision-making tasks such as contextual bandits; see Appendix E for further details. Throughout, we use the fact that the posterior distribution of the mean $\boldsymbol{\mu}$ given trajectories $\tau_h$ does not depend on the choice of learning algorithm alg; hence, we denote these posteriors $P_\theta[\cdot \mid \tau_{h-1}]$.[6]

Recall the classical Thompson sampling algorithm: at each step $h$, $\mathsf{TS}(\theta)$ draws a mean $\tilde{\boldsymbol{\mu}}_h \sim P_\theta[\cdot \mid \tau_{h-1}]$ and selects the reward-maximizing action $a_h \in \arg\max_a \tilde{\mu}_{h,a}$. We say that the prior $P_\theta$ is *B-bounded* if $P_\theta[\mathrm{diam}(\boldsymbol{\mu}) \leq B] = 1$. For Thompson sampling under $B$-bounded priors, we have the following result:

**Corollary 3.1.** *Let $P_\theta$ be $B$-bounded. Then, the suboptimality of misspecified Thompson sampling $\mathsf{TS}(\theta')$ on instance $\theta$ is at most*

$$|R(\theta, \mathsf{TS}(\theta)) - R(\theta, \mathsf{TS}(\theta'))| \leq 2H^2 \cdot \mathrm{TV}(P_\theta \parallel P_{\theta'}) \cdot B.$$

Corollary 3.1 follows directly from Theorem 3.2, which we state in Section 3.2, and which generalizes the statement of the corollary along two axes: to a more general family of Bayesian algorithms that we call "$n$-Monte Carlo" and to less restrictive conditions on the behavior of $\mathrm{diam}(\boldsymbol{\mu})$, such as sub-Gaussian tails. Due to lack of space, we focus on the first such generalization; the second direction is more technical in nature, and we leave its exposition to Appendix B.2.

## 3.1 $n$-Monte Carlo algorithms

Unfortunately, for arbitrary Bayesian bandit algorithms, the behavior under two different priors cannot always be controlled in terms of the total variation distance of their priors. Indeed, consider an algorithm that always pulls a particular arm $a^\star$ if the prior places any probability mass on a mean for which $a^\star$ is best; clearly, this algorithm's behavior is not robust to small changes in its prior distribution. However, many important Bayesian bandit algorithms, such as Thompson sampling, are not arbitrary functions of their priors; rather, they select actions based on their internal posterior distribution in a relatively stable manner. We call such algorithms $n$-*Monte Carlo algorithms*.

**Definition 3.1** ($n$-Monte Carlo algorithm). Given $n > 0$, we say that a family of algorithms $\mathsf{alg}(\cdot)$ parameterized by $\theta \in \Theta$ is $n$-Monte Carlo if, for any $\theta, \theta'$, step $h \geq 1$, and partial trajectory $\tau_{h-1}$,

$$\mathrm{TV}(P_{\mathsf{alg}(\theta)}(a_h \mid \tau_{h-1}) \parallel P_{\mathsf{alg}(\theta')}(a_h \mid \tau_{h-1})) \leq n \cdot \mathrm{TV}(P_\theta(\boldsymbol{\mu} \mid \tau_{h-1}) \parallel P_{\theta'}(\boldsymbol{\mu} \mid \tau_{h-1})).$$

In words, $n$-Monte Carlo algorithms are those Bayesian algorithms for which small changes in the posterior distribution result in small changes (up to a multiplicative factor of $n$) in the distribution over actions. Note that on the left-hand side, we do not need to specify the true $\theta^\star$, because each algorithm's choice of an action can only depend on $\tau_{h-1}$. The nomenclature arises because any algorithm that selects actions based exclusively on $n$ *samples* from its posterior $P_\theta(\boldsymbol{\mu} \mid \tau_{h-1})$ is $n$-Monte Carlo. However, the definition is more general and in Appendix C we describe various algorithms that satisfy the $n$-Monte Carlo property, summarizing key insights here:

- We show that $\mathsf{TS}(\theta)$ is 1-Monte Carlo.
- We introduce a generalization of Thompson sampling, which we call $k$-shot Thompson sampling ($k$-$\mathsf{TS}(\theta)$), that samples $k$ means $\tilde{\boldsymbol{\mu}}_1, \ldots, \tilde{\boldsymbol{\mu}}_k$ i.i.d. from the posterior $P_\theta[\cdot \mid \tau_{h-1}]$, and selects the action $a_h \in \arg\max_a \max\{\tilde{\mu}_{1,a}, \ldots, \tilde{\mu}_{k,a}\}$. We show that $k$-$\mathsf{TS}(\theta)$ is $k$-Monte Carlo.
- We introduce a Monte Carlo approximation of the knowledge gradient algorithm [RPF12], which we call two-step Receding Horizon Control (2-RHC($\theta$)). This algorithm is non-myopic in that it chooses an action that maximizes the expected value at the subsequent time (according to its own posterior updates). We show that when $\mathcal{A}$ is finite, 2-RHC($\theta$) is $n$-Monte Carlo for some $n$ that is polynomial in $|\mathcal{A}|$ and the number of Monte Carlo samples it draws from its posterior.

---

[6]Note that whenever $\tau_h$ lies in the support of $P_\theta$, the posterior $P_\theta[\boldsymbol{\mu} \mid \tau_{h-1}]$ is well-defined and unique, even if $\tau_{h-1}$ was generated by interacting with mean $\boldsymbol{\mu} \sim P_{\theta'}$ for some $\theta' \neq \theta$. When $\tau_{h-1}$ does not lie in the support of $P_\theta$, we allow $P_\theta[\boldsymbol{\mu} \mid \tau_{h-1}]$ to be any distribution over $\boldsymbol{\mu}$ (for concreteness, one may default to $P_\theta[\boldsymbol{\mu}]$). Note, however, that although $P_\theta[\boldsymbol{\mu} \mid \tau_{h-1}]$ may not be uniquely defined, $P_{\theta,\mathsf{alg}}[\tau_{h-1} \mid \boldsymbol{\mu}]$ is always uniquely defined and independent of $\theta$.

## 3.2 General Sensitivity Upper and Lower Bounds

We are now ready to state a general prior sensitivity bound for $n$-Monte Carlo algorithms. For simplicity, we state our bounds for $B$-bounded priors, that is, $P_\theta[\text{diam}(\boldsymbol{\mu}) \leq B] = 1$, and under a natural sub-Gaussian tail condition stated formally in Appendix B.2 (Theorem B.2).

**Theorem 3.2.** *Let $\text{alg}(\cdot)$ be an $n$-Monte Carlo family of algorithms on horizon $H \in \mathbb{N}$, and let $\theta, \theta' \in \Theta$. Setting $\varepsilon = \text{TV}(P_\theta \parallel P_{\theta'})$, we have the following guarantees.*

*(a) If $P_\theta$ is $B$-bounded, then $|R(\theta, \text{alg}(\theta)) - R(\theta, \text{alg}(\theta'))| \leq 2nH^2\varepsilon \cdot B$.*
*(b) If $P_\theta$ is coordinate-wise $\sigma^2$-sub-Gaussian, then*

$$|R(\theta, \text{alg}(\theta)) - R(\theta, \text{alg}(\theta'))| \leq 2nH^2\varepsilon \left( \text{diam}(E_\theta[\boldsymbol{\mu}]) + \sigma \left( 8 + 5\sqrt{\log\left( \frac{|\mathcal{A}|^2}{\min\{1, 2nH\epsilon\}} \right)} \right) \right).$$

Next, we complement our upper bound with a lower bound that matches Theorem 3.2(a) for $n$-shot Thompson sampling (an $n$-Monte Carlo algorithm) up to a multiplicative constant:

**Theorem 3.3** (Lower Bound, Informal)**.** *For any parameter $n \in \mathbb{N}$, horizon $H \gg 1$, number of arms $N = |\mathcal{A}| \gg H$, and separation $\epsilon \ll 1/nH$, there exist two priors $P_\theta$ and $P_{\theta'}$ over bounded means $\boldsymbol{\mu} \in [0,1]^N$ such that $\text{TV}(P_\theta \parallel P_{\theta'}) = \epsilon$ and*

$$R(\theta, n\text{-TS}(\theta)) \geq R(\theta, n\text{-TS}(\theta')) + (1 - o(1)) \cdot \frac{nH^2\epsilon}{2},$$

*where the $o(1)$ decays to zero as $1/H$, $H/N$, $\epsilon nH \to 0$.*

See Theorem D.1 for a precise, quantitative statement and Appendix D for a full proof.

**Remark 1** (Comparison to $\tilde{O}(\sqrt{H})$ regret guarantees)**.** At first glance, Theorem 3.3 appears inconsistent with known upper bounds for Thompson sampling which show that regret relative to the best action in hindsight scales sublinearly as $\tilde{O}(\sqrt{|\mathcal{A}|H})$ in the horizon $H$. Notice however that our lower bound requires the number of actions $|\mathcal{A}|$ to scale at least linearly in $H$, so the bound applies in a regime where regret upper bounds are in fact vacuous. Instead, the purpose of our lower bound is to quantify the influence of the misspecified prior for *fixed* horizons, but where the magnitude $\text{TV}(P_\theta \parallel P_{\theta'})$ of the misspecification may be arbitrarily small.

**Proof ideas.** One of the key ingredients in the proof of Theorem 3.2, and a result which may be of independent interest, is the following bound on the total variation of the trajectory of an algorithm run with the true prior and the same algorithm run with an incorrect prior.

**Proposition 3.4.** *Let $\text{alg}(\cdot)$ be an $n$-Monte Carlo family of algorithms on horizon $H \in \mathbb{N}$. Then,*

$$\text{TV}(P_H \parallel P'_H) \leq 2nH \cdot \text{TV}(P_\theta \parallel P_{\theta'}),$$

*where $P_H = P_{\theta, \text{alg}(\theta)}(\boldsymbol{\mu}, \tau_H)$ and $P'_H = P_{\theta, \text{alg}(\theta')}(\boldsymbol{\mu}, \tau_H)$.*

A full proof is given in Appendix B.6 and relies on a careful coupling argument between the two trajectories detailed therein. As a concrete warmup, we illustrate this coupling for a Gaussian bandit instance in Appendix B.5. The factor of $H$ arises from a telescoping argument (Lemma B.9) based on the performance-difference lemma [Kak03].

For $B$-bounded priors, Proposition 3.4 directly translates into the sensitivity bound in Theorem 3.2(a), where the difference in rewards can be bounded as $BH$ times the probability that the trajectory of $\text{alg}(\theta)$ differs from the trajectory of $\text{alg}(\theta')$. Addressing more general tail conditions like sub-Gaussianity requires more care; see Appendix B for details.

## 4 Meta-learning

In this section, we apply the above prior sensitivity guarantees to episodic Bayesian meta-learning and obtain sample-efficiency guarantees for canonical Bayesian bandit setups.

Suppose an episodic Bayesian meta-learner uses an exploration strategy $\text{explore}^{(t)}$ in $T$ episodes and computes an estimate $\hat{\theta} = \hat{\theta}(\tau^{(1)}, \ldots, \tau^{(T)})$ of the ground-truth parameter $\theta^\star$. Suppose further that, for any $\varepsilon, \delta \in (0, 1)$, with probability at least $1 - \delta$ over the realizations of the episodes and internal

randomization of the meta-learner, the estimate $\hat{\theta}$ satisfies $\text{TV}(P_{\theta^\star} \parallel P_{\hat{\theta}}) \leq \varepsilon$. Then, Theorem 3.2 implies that, for any $n$-Monte Carlo algorithm $\text{alg}(\cdot)$, the relative performance of $\text{alg}(\hat{\theta})$ compared to $\text{alg}(\theta^\star)$ is (essentially) bounded as $\tilde{O}(nH^2\varepsilon)$ over horizon $H$.

Our task of designing meta-learners is thus reduced to that of designing estimators (and exploration strategies) for $\theta^\star$ that enjoy convergence guarantees in TV distance. This is quite a general recipe that can produce concrete meta-learning algorithms in many Bayesian bandit settings. We explain how to do so in two setups: (1) $P_{\theta^\star}$ is a product of Beta distributions, and the rewards are Bernoulli; (2) $P_{\theta^\star}$ is a multivariate Gaussian and the rewards are Gaussian.

## 4.1 Beta Priors and Bernoulli Rewards

We first consider the situation where the prior distribution is a product of Beta distributions $P_{\theta^\star} = \bigotimes_{a \in \mathcal{A}} \text{Beta}(\alpha_a^\star, \beta_a^\star)$ and the reward distribution is a product of Bernoulli distributions $\mathcal{D}(\boldsymbol{\mu}) = \bigotimes_{a \in \mathcal{A}} \text{Bern}(\mu_a)$. Recall that $\text{Beta}(\alpha, \beta)$ for $\alpha > 0$ and $\beta > 0$ is a continuous probability distribution supported on $(0, 1)$, and hence our parameter space $\Theta$ is the (strictly) positive orthant in $\mathbb{R}^{2|\mathcal{A}|}$.

Our approach is to directly estimate the parameters $\boldsymbol{\theta}^\star = (\boldsymbol{\alpha}^\star, \boldsymbol{\beta}^\star)$ from the observed rewards in the $T$ episodes. Since the family of Beta distributions is an exponential family [Bro86] (with $(\alpha, \beta)$ being the natural parameters), we can appeal to general statistical theory to bound the total variation distance between two such distributions in terms of their parameter distance.

Suppose we adopt the exploration strategy where arm 1 is selected in the first $n$ rounds in each of the first $T/|\mathcal{A}|$ episodes, arm 2 in the next $T/|\mathcal{A}|$ episodes, and so on. (We assume the horizon $H$ and $n$ satisfy $H \geq n \geq 2$.) We focus on the estimation of $(\alpha_1^\star, \beta_1^\star)$, as the exact same approach works for all of the arms. Let $X_t$ denote the cumulative reward collected in the first $n$ rounds of episode $t$. Then, the random variables $X_1, \ldots, X_{T/|\mathcal{A}|}$ are i.i.d. draws from a Beta-Binomial distribution with parameters $(\alpha_1^\star, \beta_1^\star, n)$, where $n$ denotes the number of trials of the binomial component. The first and second moments of $X_t$ are

$$m_1^\star = \mathbb{E}[X_t] = \frac{n\alpha_1^\star}{\alpha_1^\star + \beta_1^\star} \quad \text{and} \quad m_2^\star = \mathbb{E}[X_t^2] = \frac{n\alpha_1^\star(n(1 + \alpha_1^\star) + \beta_1^\star)}{(\alpha_1^\star + \beta_1^\star)(1 + \alpha_1^\star + \beta_1^\star)}.$$

These moments uniquely determine $\alpha_1^\star$ and $\beta_1^\star$ as long as $n \geq 2$. Therefore, we can estimate $(\alpha_1^\star, \beta_1^\star)$ using plug-in estimates of the first two moments $(m_1^\star, m_2^\star)$ via the method of moments [TGG94]. Using this approach, we obtain the following sample complexity guarantee for estimating the prior distribution:

**Theorem 4.1.** *The exploration strategy and estimator described above enjoy the following guarantee. If $P_{\boldsymbol{\theta}^\star} = \bigotimes_{a \in \mathcal{A}} \text{Beta}(\alpha_a^\star, \beta_a^\star)$ and $\mathcal{D}(\boldsymbol{\mu}) = \bigotimes_{a \in \mathcal{A}} \text{Bern}(\mu_a)$, then there is a constant $C$ depending only on $(\boldsymbol{\alpha}^\star, \boldsymbol{\beta}^\star)$ such that, for any $\varepsilon, \delta \in (0, 1)$, if $H \geq 2$ and*

$$T \geq \frac{C \cdot |\mathcal{A}|^2 \log(|\mathcal{A}|/\delta)}{\varepsilon^2},$$

*then $\mathbb{P}[\text{TV}(P_{\boldsymbol{\theta}^\star} \parallel P_{\hat{\boldsymbol{\theta}}}) \leq \varepsilon] \geq 1 - \delta$.*

The proof of the theorem is given in Appendix F.1.

## 4.2 Gaussian Priors and Gaussian Rewards

We now consider the situation where the prior distribution is a multivariate Gaussian $P_{\boldsymbol{\theta}^\star} = \mathcal{N}(\boldsymbol{\nu}_\star, \boldsymbol{\Psi}_\star)$ in $\mathbb{R}^{\mathcal{A}}$, and the reward distribution is a spherical Gaussian distribution $\mathcal{D}(\boldsymbol{\mu}) = \mathcal{N}(\boldsymbol{\mu}, \sigma^2 \mathbf{I})$. Note that such a prior distribution is able to capture correlations between the arms' mean rewards in an episode, which cannot be captured by the product-form priors in the previous subsection (nor in previous work [KKZ+21]).

We again directly estimate the parameters $\boldsymbol{\theta}^\star = (\boldsymbol{\nu}_\star, \boldsymbol{\Psi}_\star)$ using a simple exploration strategy and the method of moments. In each episode (which we assume to have $H \geq 2$), we select independent and uniformly random actions in the first two rounds. Let $a_t$ and $b_t$ denote the actions taken in episode $t$, and let $r_t$ and $s_t$ denote the corresponding observed rewards. Our estimates for $\boldsymbol{\nu}_\star$ and $\boldsymbol{\Psi}_\star$ based on

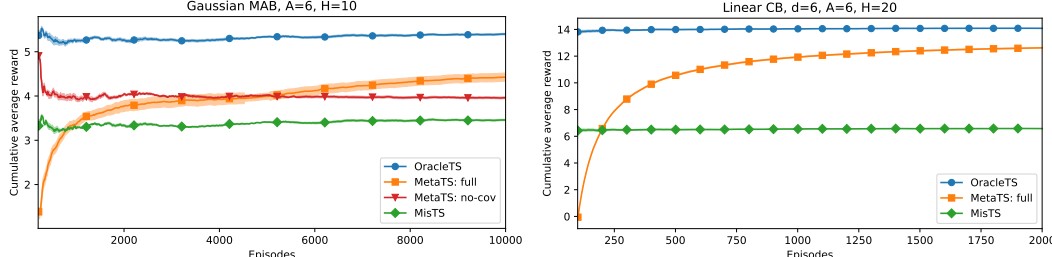

Figure 1: Learning curves for Gaussian MAB and linear CB experiments. We run 100 replicates per algorithm and visualize two standard errors with error bands. For meta-learners we tune the number of exploration rounds and display the performance of the best configuration at each point, which we call the upper envelope.

the information collected in $T$ episodes are[7]

$$\widehat{\boldsymbol{\nu}} := \frac{|\mathcal{A}|}{T} \sum_{t=1}^{T} r_t \mathbf{e}_{a_t} \quad \text{and} \quad \widehat{\boldsymbol{\Psi}} := \frac{|\mathcal{A}|^2}{T} \sum_{t=1}^{T} r_t s_t \left( \mathbf{e}_{a_t} \mathbf{e}_{b_t}^{\mathsf{T}} + \mathbf{e}_{b_t} \mathbf{e}_{a_t}^{\mathsf{T}} \right) - \widehat{\boldsymbol{\nu}} \widehat{\boldsymbol{\nu}}^{\mathsf{T}}.$$

For these estimators, we have the following theorem.

**Theorem 4.2.** *The exploration strategy and estimator described above enjoy the following guarantee. If $P_{\boldsymbol{\theta}^\star} = \mathcal{N}(\boldsymbol{\nu}_\star, \boldsymbol{\Psi}_\star)$ and $\mathcal{D}(\boldsymbol{\mu}) = \mathcal{N}(\boldsymbol{\mu}, \sigma^2 \mathbf{I})$, then there is a constant $C$ depending only on $(\boldsymbol{\nu}_\star, \boldsymbol{\Psi}_\star)$ and $\sigma^2$ such that, for any $\varepsilon, \delta \in (0,1)$, if $H \geq 2$ and*

$$T \geq \frac{C \cdot (|\mathcal{A}|^4 + |\mathcal{A}|^3 \log(1/\delta))}{\varepsilon^2},$$

*then $\mathbb{P}[\mathrm{TV}(P_{\boldsymbol{\theta}^\star} \parallel P_{\widehat{\boldsymbol{\theta}}}) \leq \varepsilon] \geq 1 - \delta$.*

The proof of the theorem and the precise dependence on $\boldsymbol{\nu}_\star$, $\boldsymbol{\Psi}_\star$, and $\sigma^2$ are given in Appendix F.2. The quartic dependence on $|\mathcal{A}|$ is due to estimating $\boldsymbol{\Psi}_\star$; it improves to $|\mathcal{A}|^2$ if $\boldsymbol{\Psi}_\star$ is known.

**Remark 2** (Comparison to [KKZ+21]). [KKZ+21] study the case where $P_{\boldsymbol{\theta}^\star} = \mathcal{N}(\boldsymbol{\nu}_\star, \sigma_0^2 \mathbf{I})$, which is a product-form prior over means $\boldsymbol{\mu}$ with known $\sigma_0^2$. For $\tilde{\epsilon} = |\mathcal{A}| \cdot \|\boldsymbol{\nu}_\star - \widehat{\boldsymbol{\nu}}\|_\infty / \sigma_0$, they show that[8]

$$|R(\boldsymbol{\theta}, \mathsf{TS}(\boldsymbol{\theta})) - R(\boldsymbol{\theta}, \mathsf{TS}(\widehat{\boldsymbol{\theta}}))| \leq O\left( \|\boldsymbol{\nu}_\star\|_\infty + \sigma_0 \sqrt{\log(H/\tilde{\epsilon})} \right) \cdot H^2 \tilde{\epsilon}.$$

On the other hand, Theorem 3.2 applied to the 1-Monte Carlo Thompson Sampling algorithm (and bounding $\mathrm{diam}(\boldsymbol{\nu}_\star) \leq \|\boldsymbol{\nu}_\star\|_\infty$) yields the same inequality, but with $\tilde{\epsilon}$ replaced by $\epsilon = \mathrm{TV}(\mathcal{N}(\boldsymbol{\nu}_\star, \sigma_0^2 I) \parallel \mathcal{N}(\widehat{\boldsymbol{\nu}}, \sigma_0^2 I)) \leq \|\boldsymbol{\nu}_\star - \widehat{\boldsymbol{\nu}}\|_2 / \sigma_0$. Note that $\tilde{\epsilon}$ is always larger than $\epsilon$ by a factor of at least $\sqrt{|\mathcal{A}|}$; thus, our result is strictly sharper.

## 5 Experiments

We demonstrate the generality of our results in three distinct meta-learning experimental settings. First, we study a simple multi-armed bandit scenario with Gaussian prior and Gaussian rewards, where we demonstrate how meta-learning higher-order moments of the prior can significantly improve performance. Next, we consider a Gaussian linear contextual bandits scenario, to demonstrate the generality of Bayesian meta-learning. Finally, we study a more interesting multi-armed bandit problem with discrete priors, where, in addition to the value of meta-learning, we see that look-ahead algorithms can substantially outperform Thompson sampling. Additional experimental details are presented in Appendix A.

**Gaussian MAB.** Our first scenario is a multi-armed bandit problem with Gaussian prior and Gaussian reward. The instance has $|\mathcal{A}| = 6$ arms and each episode has horizon $H = 10$. The prior is $\mathcal{N}(\boldsymbol{\nu}_\star, \boldsymbol{\Psi}_\star)$ where $\boldsymbol{\nu}_\star = [0.5, 0, 0, 0.1, 0, 0]$ and $\boldsymbol{\Psi}_\star$ has block structure so that arms $1, 2, 3$ are

---

[7]This estimator can be generalized to explore for more of the episode and use more of the observed rewards.

[8]The following optimizes Lemma 5 of [KKZ+21] over its free parameter $\delta > 0$ for $\tilde{\epsilon}$ small.

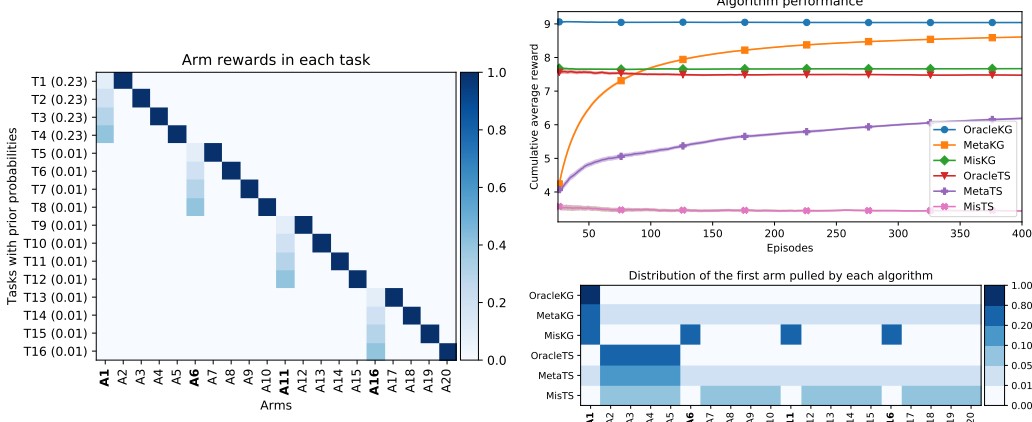

Figure 2: Synthetic experiments with discrete MAB for $|\mathcal{A}| = 20$ and $H = 10$. *Left*: visualization of the instance showing the reward for each of the arms in each of the 16 possible tasks along with the prior distribution over tasks (probabilities rounded, actual values are $9/40$ and $1/120$). *Top right*: learning curves for 6 algorithms (100 replicates, error bands at 2 standard errors, we tune the number of exploration rounds and plot upper envelopes for meta-learners). *Bottom right*: empirical distribution of the first arm pulled in each episode by each algorithm. Note that the color scale is non-linear.

highly correlated, and analogously for arms $4, 5, 6$. The rewards are Gaussian with variance 1, which is known to all learners.

We run four algorithms. Two are non-meta-learning Thompson sampling algorithms: **OracleTS**, which uses the correct prior, and **MisTS**, which uses the misspecified prior $\mathcal{N}(\mathbf{0}, \mathbf{I})$. We also run **MetaTS:no-cov** which only attempts to meta-learn the prior mean $\mu_0$ and assumes that the prior covariance matrix is the identity (this algorithm is essentially the one studied in [KKZ+21]). Finally, our algorithm is **MetaTS:full** which meta-learns both the prior mean and covariance. Both meta-learners are run in an explore-then-commit fashion where the first $T_0$ episodes are used for exploration.[9]

In Figure 1, we plot the cumulative average per-episode reward for each algorithm, where for the meta-learners we sweep over many choices of $T_0$ and display the pointwise best (i.e., the upper envelope). The experiment clearly shows the value of meta-learning as both **MetaTS:no-cov** and **MetaTS:full** quickly outperform misspecified TS. Additionally, we also see the importance of learning the covariance matrix, even though it can require many samples. Indeed, the final performance of **MetaTS:full** with $T_0 = 5\mathrm{K}$, ignoring the regret incurred due to exploration, is competitive with **OracleTS**, while **MetaTS:no-cov** asymptotes to a much lower performance (see Figure 3 in Appendix A).

**Gaussian linear contextual bandits.** Our second experiment concerns Gaussian linear contextual bandits. Here we run **OracleTS**, **MisTS**, and **MetaTS:full**, on a synthetic linear contextual bandit problem where there are $|\mathcal{A}| = 6$ actions each with a $d = 6$ dimensional action feature (generated stochastically at each time step), and with horizon $H = 20$. The prior is over the linear parameter $\boldsymbol{\mu}$ that determines the reward for action-feature $\mathbf{x}_a \in \mathbb{R}^d$ as $r(a) \sim \mathcal{N}(\langle \boldsymbol{\mu}, \mathbf{x}_a \rangle, 1)$. We set the prior as $\mathcal{N}(\mathbf{1}, \boldsymbol{\Psi}_\star)$ where $\boldsymbol{\Psi}_\star$ is a scaled-down version of the block diagonal matrix used in the previous experiment. In the right panel of Figure 1 we again see that by meta-learning the prior, we quickly outperform the misspecified approach and asymptotically achieve the oracle performance. This demonstrates that Bayesian meta-learning is quite broadly applicable and highlights the importance of our general theoretical development.

**Discrete bandits.** Finally, we study a synthetic MAB setting with $|\mathcal{A}| = 20$ arms and a prior supported on a finite set of 16 reward distributions (tasks), under each of which rewards are deterministic. The instance is visualized in the left panel of Figure 2. It is constructed so that each task has a unique optimal arm and there are four arms that can quickly identify which task the agent is in (arms A1, A6,

---

[9]For **MetaTS:no-cov**, we follow [KKZ+21] and only use the first step of each exploration episode for exploration, switching to TS with the current prior estimate for the rest of the episode. On the other hand, **MetaTS:full** explores for all time steps in the first $T_0$ episodes.

A11, A16), so that it can infer the optimal arm. Additionally, the prior is concentrated on the first four tasks, so that pulling the first identifying arm almost always reveals the current task.

We evaluate 6 algorithms: Oracle, Misspecified, and Meta-learning each with TS and Monte-Carlo Knowledge Gradient (an instantiation of the 2-RHC($\theta$) algorithm detailed in Appendix C) as the base learners, and we visualize the results in the top right panel of Figure 2. Perhaps more revealing is the bottom right panel of Figure 2, where we visualize the empirical distribution over the first arm pull in each episode for each algorithm. We see that `OracleTS` typically plays uniformly over arms A2–A5 in the first round as these are highly likely to be the optimal arm under the prior, while `MisTS` plays uniformly over the 16 plausibly optimal arms. `MetaTS` quickly learns to play uniformly over arms A2–A5 and is asymptotically competitive with `OracleTS`.

The interesting property of this instance is that playing the identifying arms is crucial for optimal behavior. However, since TS is myopic and these arms never produce large rewards, TS will never play them. Thus, to achieve optimal behavior, we must use a less myopic base learner like Knowledge Gradient. As can be seen, both `OracleKG` and `MisKG` first play the identifying arms, where the oracle almost always pulls the first one while `MisKG` plays them uniformly. The performance of `OracleKG` is much better than all TS configurations. Finally, the meta-learning configuration of Knowledge Gradient quickly learns to pull the first identifying arm and competes with `OracleKG`.

## 6 Discussion

In our simulations, we demonstrated the superiority of more expressive prior families (e.g., modeling means and covariances) and non-myopic base algorithms (e.g., Knowledge Gradient) over less expressive priors (e.g., product measures) and greedy base learners (e.g., Thompson sampling). Notably, the generality and flexibility of our theoretical contributions ensure robustness to prior misspecification even for these richer priors and sophisticated base learners.

Still, theory and experiments alike point to a tradeoff: despite the potential for improved performance, richer prior families are harder to learn, and some base learners (e.g., $n$-Monte Carlo algorithms for large $n$) can be more sensitive to incorrect priors. It is an exciting direction for future work to investigate the joint problems of *model selection* (over priors) and *algorithm selection* (over base learners) in order to optimally navigate these tradeoffs. Perhaps model and algorithm selection can be coupled so that certain base learners exhibit improved performance, or greater robustness, over certain classes of priors. We would like to further understand how these tradeoffs interface with computational burdens of using certain priors and base learners, and whether our sensitivity analysis extends to computationally efficient approximations of sampling-based decision-making algorithms (e.g., via Laplace approximations, MCMC, Gibbs Sampling, and Variational Methods; the long-horizon performance of Thompson sampling under approximate inference has already been studied [PAYD19]). Finally, we hold hope that a more instance-dependent analysis may improve our sensitivity bounds for certain families of priors, which may in turn inform more clever exploration strategies that circumvent worst-case tradeoffs.

**Acknowledgements**

The authors thank Wen Sun for many discussions that helped shape the current paper. At the time of writing, Max Simchowitz was generously supported by an Open Philanthropy AI Fellowship, and the research was initiated during his internship at MSR New York and PhD at UC Berkeley. Research was initiated during Thodoris Lykouris' postdoc at MSR New York. Daniel Hsu is supported by NSF grants CCF-1740833, DMREF-1534910, IIS-1563785; a Bloomberg Data Science Research Grant, a JP Morgan Faculty Award, and a Sloan Research Fellowship; Christoper Tosh is also supported by NSF grant CCF-1740833 and Daniel Hsu's JP Morgan Faculty Award.

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
