# Contents

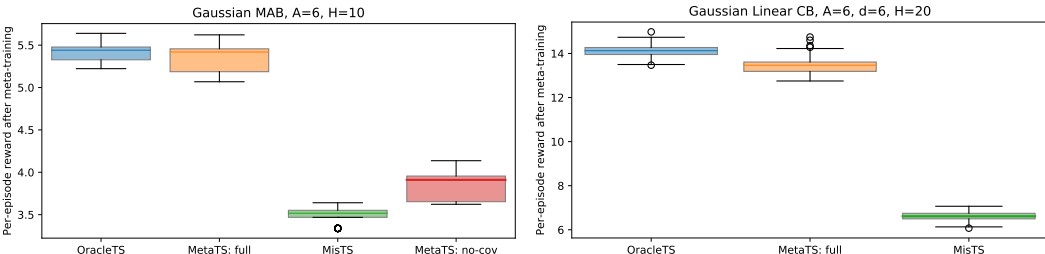

Figure 3: Test performance in Gaussian MAB and Gaussian linear CB experiments.

# A  Additional Experimental Details

In this section, we provide additional experimental details for each setting. As a prelude, the total amount of compute is very minimal and primarily inflated by the large number of replicates used in each experiment. On a standard CPU cluster the experiments can easily be completed in 2-4 hours, even with running 100 replicates for each algorithm/configuration.

## A.1  Multi-armed bandit experiments

As described, the left panel of Figure 1 is based on a $|\mathcal{A}| = 6$ arm bandit problem with horizon $H = 10$ and prior $\mathcal{N}(\boldsymbol{\nu}_\star, \boldsymbol{\Psi}_\star)$, where

$$\boldsymbol{\nu}_\star = [0.5, 0, 0, 0.1, 0, 0], \quad \text{and} \quad \boldsymbol{\Psi}_\star = \begin{pmatrix} 1 & 0.9 & 0.9 & 0 & 0 & 0 \\ 0.9 & 1 & 0.9 & 0 & 0 & 0 \\ 0.9 & 0.9 & 1 & 0 & 0 & 0 \\ 0 & 0 & 0 & 1 & 0.9 & 0.9 \\ 0 & 0 & 0 & 0.9 & 1 & 0.9 \\ 0 & 0 & 0 & 0.9 & 0.9 & 1 \end{pmatrix}.$$

The rewards are Gaussian, with variance $1.0$.

The four algorithms we run are:

- **OracleTS**: The standard implementation of Gaussian Thompson sampling, with the correct prior $(\boldsymbol{\nu}_\star, \boldsymbol{\Psi}_\star)$.
- **MisTS**: The standard implementation of Gaussian Thompson sampling, with the incorrect prior $(\mathbf{0}, \mathbf{I})$.
- **MetaTS:full**: A meta-learning implementation of Gaussian Thompson sampling with an "explore-then-commit" strategy. This algorithm has a hyperparameter $T_0$ which determines the number of exploration rounds. In the first $T_0$ rounds, the algorithm simply selects all actions uniformly at random. Then at the end of the $T_0$ exploration rounds, it forms an estimate $(\widehat{\boldsymbol{\nu}}, \widehat{\boldsymbol{\Psi}})$ as follows:

$$\widehat{\boldsymbol{\nu}} = \frac{|\mathcal{A}|}{T_0} \sum_{i=1}^{T_0} \widehat{\boldsymbol{\mu}}_i = \frac{|\mathcal{A}|}{T_0} \sum_{i=1}^{T_0} \left( \frac{1}{H} \sum_{h=1}^{H} \sum_{a \in \mathcal{A}} \mathbf{1}\{a_{i,h} = a\} e_a r_{i,h} \right),$$

$$\widehat{\boldsymbol{\Psi}} = \frac{1}{T_0} \sum_{i=1}^{T_0} \left( \widehat{\boldsymbol{\mu}}_i \widehat{\boldsymbol{\mu}}_i^\top - \text{diag}(\widehat{\boldsymbol{\mu}}_i \widehat{\boldsymbol{\mu}}_i^\top) + \text{diag}\left( \frac{|\mathcal{A}|}{H} \sum_{h=1}^{H} \sum_a \mathbf{1}\{a_{i,h} = a\} e_a r_{i,h}^2 \right) - \mathbf{I} \right) - \widehat{\boldsymbol{\nu}} \widehat{\boldsymbol{\nu}}^\top.$$

Here $a_{i,h}$ is the action played at the $h^{\text{th}}$ time step of the $i^{\text{th}}$ episode and $r_{i,h}$ is the corresponding reward. It is not difficult to verify that both of these are unbiased estimators for $\boldsymbol{\nu}_\star$ and $\boldsymbol{\Psi}_\star$ respectively. We additionally project $\widehat{\boldsymbol{\Psi}}$ onto the positive semidefinite cone. After the $T_0$ exploration rounds, **MetaTS:full** forms the above estimators and runs standard Gaussian Thompson sampling with the estimates $(\widehat{\boldsymbol{\nu}}, \widehat{\boldsymbol{\Psi}})$.

- **MetaTS:no-cov**: A meta-learning implementation of Gaussian Thompson sampling with an "explore-then-commit" strategy, which does not estimate the prior covariance. As above, it has

a hyperparameter $T_0$ determining the number of exploration rounds. In the first $T_0$ rounds, the algorithm chooses just the first action uniformly at random and then chooses the remaining actions by instantiating Gaussian Thompson sampling with the current estimate of the prior mean and the incorrect prior covariance $\mathbf{I}$. The prior mean at round $t$ is estimated as

$$\widehat{\boldsymbol{\nu}}_t = \frac{|\mathcal{A}|}{t-1} \sum_{i=1}^{t-1} \sum_{a \in \mathcal{A}} \mathbf{1}\{a_{i,1} = a\} e_a r_{i,1},$$

which is analogous to the estimate above. After $T_0$ rounds, we set $\widehat{\boldsymbol{\nu}} = \widehat{\boldsymbol{\nu}}_{T_0}$ and we run standard Gaussian Thompson sampling with prior $(\widehat{\boldsymbol{\nu}}, \mathbf{I})$ for the remaining rounds.

**Experimental Protocol and Results.** In the left panel of Figure 1 we run each algorithm (with each hyperparameter configuration) for 100 replicates with different random seeds. For both **MetaTS** variants, we choose $T_0$ from the set $\{200, 400, 600, \ldots, 5000\}$. In the figure, we record the average (across replicates) performance at each episode number with error bands corresponding to $\pm 2$ standard errors.

For the algorithms with a hyperparameter, we plot the performance of the pointwise best hyperparameter configuration. That is, we optimize hyperparameters (based on average-across-replicates performance) for each episode number $n$ individually.

In the left panel of Figure 3 we visualize the "test performance" of the various algorithms, which corresponds to the average per-episode performance for the last $5,000$ episodes. Here the box plots visualize the 100 different replicates. For both **MetaTS** variants, we use $T_0 = 5,000$ as the hyperparameter. Note that since the total number of episodes is $10,000$, both algorithms do not update their prior estimate for the episodes during which we record performance.

## A.2 Linear contextual bandit experiments

The experimental protocol is similar to the one above. Here we consider a Gaussian linear contextual bandit setup with $|\mathcal{A}| = 6$ actions and $d = 6$ dimensional action features and horizon $H = 20$. The prior is $\mathcal{N}(\boldsymbol{\nu}_\star, \boldsymbol{\Psi}_\star)$ where

$$\boldsymbol{\nu}_\star = \mathbf{1}, \quad \text{and} \quad \boldsymbol{\Psi}_\star = 0.1 \times \begin{pmatrix} 1 & 0.9 & 0.9 & 0 & 0 & 0 \\ 0.9 & 1 & 0.9 & 0 & 0 & 0 \\ 0.9 & 0.9 & 1 & 0 & 0 & 0 \\ 0 & 0 & 0 & 1 & 0.9 & 0.9 \\ 0 & 0 & 0 & 0.9 & 1 & 0.9 \\ 0 & 0 & 0 & 0.9 & 0.9 & 1 \end{pmatrix}.$$

In each round the action features are generated by sampling each entry from a standard normal distribution and then normalizing so that the feature vector has $\ell_2$ norm equal to 1. For action feature $\mathbf{x}_a$ the reward is given by $r(a) \sim \mathcal{N}(\langle \boldsymbol{\mu}, \mathbf{x}_a \rangle, 1)$.

We run three algorithms here. The first two **OracleTS** and **MisTS** are standard implementations of Gaussian linear Thompson sampling with well-specified and mis-specified priors respectively. Here **MisTS** is initialized with prior $\mathcal{N}(\mathbf{0}, \mathbf{I})$. The final algorithm, **MetaTS:full** is implemented in the explore-then-commit fashion described above. The only difference is the estimator for the prior. Here in each episode of the exploration stage, we choose actions uniformly at random and use ordinary least squares to estimate the parameter $\boldsymbol{\mu}$ of the episode. The prior mean is simply estimated using the average of these OLS solutions. The prior covariance is estimated as

$$\widehat{\boldsymbol{\Psi}} = \left( \frac{1}{T_0} \sum_{i=1}^{T_0} \widehat{\boldsymbol{\mu}}_i \widehat{\boldsymbol{\mu}}_i^\top - \Sigma_i^{-1} \right) - \widehat{\boldsymbol{\nu}} \widehat{\boldsymbol{\nu}}^\top,$$

where $\Sigma_i = \sum_{h=1}^H \mathbf{x}_{h,a_h} \mathbf{x}_{h,a_h}^\top$ is the second moment matrix of the action features chosen in the episode. As above, this is an unbiased estimator of the prior covariance.

**Experimental Protocol and Results.** We follow the same protocol as above, running each algorithm for 100 replicates and, for **MetaTS:full**, we plot the pointwise best performance across hyperparameter configurations. Here we tune $T_0 \in \{100, 200, \ldots, 1000\}$. In the right panel of Figure 3 we plot the test performance of each algorithm, measured as the average performance in the final $1,000$ episodes. We use $T_0 = 1000$ for **MetaTS:full**.

### A.3 Discrete bandits

The final experiment is with the discrete MAB instance visualized in Figure 2. As the instance is visualized in the left panel, we only describe the algorithms and the experimental protocol. As the reward distributions are singular, posteriors collapse frequently in this experiment. Once this happens, all algorithms simply play the best arm from then on.

Thompson sampling as a base learner is standard. We maintain a posterior distribution over tasks, sample an instance/task from this distribution, and play the best arm for that task. Posterior updates are straightforward due to the singular nature of the reward distributions.

For Knowledge Gradient, we implement a one-step look-ahead variant, which is exactly as described in Algorithm 3, with $k_1 = k_2 = 10$ and $\alpha = 1$. We also implement a random tie breaking scheme where we choose randomly among actions with the maximum $V_a$.

We implement the meta-learners in a straightforward explore-then-commit manner. In each exploration round, we choose actions uniformly at random. If the posterior collapses, then we increment a counter associated with the current task. If the posterior does not collapse during the episode then we do not increment any counter. After $T_0$ exploration rounds we estimate the posterior by the empirical fraction of times we observed each task.

As above, we run 100 replicates of each algorithm. Misspecified variants are initialized with the uniform prior over tasks. For the meta-learners we tune $T_0 \in \{25, 50, \dots, 200\}$. We plot the point-wise best (across hyperparameters) mean performance across replicates, with bands corresponding to $\pm 2$ standard errors. Note that there is very little variance here since we run many replicates and the problem has little noise.

In the bottom panel of Figure 2 we plot the empirical distribution of the first action chose by each algorithm, where we compute this distribution using all 400 episodes and all 100 replicates of each algorithm. For both meta-learners we use $T_0 = 100$ here.

## B   Proof of Sensitivity Bounds

In this appendix, we give the proofs of Theorem 3.2 and Proposition 3.4. The results in this appendix are much more general than those stated in Section 3 and require us to introduce some new concepts. The following roadmap may be useful in navigating the rest of this appendix.

- Appendix B.1 provides some key properties of total variation distance that are used in the rest of Appendix B, as well as in Appendix C.
- Appendix B.2 provides the statements of the main results of this section. In particular, we define our notion of upper tail expectation, we introduce our tail conditions, and we provide the statements of our generalizations of Theorem 3.2 (Theorem B.1 and Theorem B.2).
- Appendix B.3 and Appendix B.4 provide key properties of our upper tail expectation and bound the upper tail expectation under our tail conditions.
- Appendix B.5 provides an analysis of a Gaussian bandit instance.
- Appendix B.6 and Appendix B.7 together give the proof of Proposition 3.4.
- Appendix B.8 finishes the proof of Theorem B.1.

### B.1   Key Properties of the Total Variation Distance

**Technical disclaimer.**   In what follows, we will need that our probability space $(\Omega, \mathscr{F})$ allows for the equivalence between total variation distance and couplings. One way that this can be guaranteed is if (a) our space $\Omega$ is Polish, i.e., that $\Omega$ is metrizable by a metric that makes it complete and separable and (b) our $\sigma$-algebra $\mathscr{F}$ is the Borel algebra $\mathscr{B}(\Omega)$, i.e., the $\sigma$-algebra generated by open sets in $\Omega$ [Lin02]. Furthermore, we assume all random variables $X : (\Omega, \mathscr{F}) \to \mathcal{X}$ take values in a Polish space $\mathcal{X}$. We endow $\mathcal{X}$ with the Borel $\sigma$-algebra $\mathscr{B}(\mathcal{X})$, and assume that $X$ is measurable from $(\Omega, \mathscr{F}) \to (\mathcal{X}, \mathscr{B}(\mathcal{X}))$; that is, $X^{-1}(\mathcal{E}) \in \mathscr{F}$ for all $\mathcal{E} \in \mathscr{B}(\mathcal{X})$. We let $\Delta(\mathcal{X})$ denote the set of all Borel-measurable distributions on $\mathcal{X}$.

**Randomized Algorithms.** Throughout, we often refer to *randomized algorithms*. Formally, a family of randomized bandit algorithms $\mathrm{alg}(\theta)$ is a specified by a distribution $\mathcal{D}_{\mathrm{seed}}$ (independent of $\theta$), a domain $\Xi$ over random seeds $\boldsymbol{\xi}$, and step-wise mappings $f_1, \ldots, f_H$ from trajectories, the random seed, and parameters $\theta$ to distributions over actions:

$$f_h(\tau_{h-1}, \boldsymbol{\xi} \mid \theta) : \{h\text{-trajectories}\} \times \Xi \times \Theta \to \Delta(\mathcal{A}).$$

Each $\mathrm{alg}(\theta)$ operates as follows:

- $\boldsymbol{\xi}$ is drawn from $\mathcal{D}_{\mathrm{seed}}$ at the start of the episode before interaction.
- At each step $h$, $a_h$ is chosen as $a_h \sim f_h(\tau_h, \boldsymbol{\xi} \mid \theta)$, independently of the past

**Remark 3** (Sources of Randomness). Note that we allow for *two* sources of randomness: the draw of $a_h$ from the distribution $f_h(\tau_{h-1}, \boldsymbol{\xi} \mid \theta)$, and the initial random seed $\boldsymbol{\xi}$ at the start of the episode. For many natural algorithms - such as those Appendix C - we do not need $\boldsymbol{\xi}$, and can just represent the randomness via actions selected independently for trajectory-dependent distributions. However, in some case, it may be desirable for there to be a random seed $\boldsymbol{\xi}$ encoding randomness shared across stages. Moreover, the assumption that $\mathcal{D}_{\mathrm{seed}}$ does not depend on $\theta$ is very mild, and can be satisfied by all families $\mathrm{alg}(\cdot)$ which can be run on a single random number generator independent of $\theta$.

**Total Variation and its Key Properties.** Recall the definition of the total variation distance.

**Definition B.1.** Let $P, P'$ be two probability measures on a space $(\Omega, \mathscr{F})$. Then $\mathrm{TV}(P \parallel P') = \sup_{\mathcal{E}} |P(\mathcal{E}) - P'(\mathcal{E})|$ is the maximal difference in probabilities of measurable events $\mathcal{E} \in \mathscr{F}$.

In our proofs, we make use of the following elementary properties of the total variation distance.

**Lemma B.1** (Total Variation Properties). *Let $P, P', P''$ be any three probability measures of the same probability space $(\Omega, \mathscr{F})$.*

(a) Coupling Form: *Let $Q$ be a coupling of $P$ and $P'$, i.e. a joint distribution over $(X, Y)$ such that its marginal distribution over $X$ is $P$ and its marginal distribution over $Y$ is $P'$. Then for any such coupling $Q$, we have*

$$\mathrm{TV}(P \parallel P') \le Q(X \ne Y).$$

*Moreover, there exists a maximal coupling $Q$ such that*

$$\mathrm{TV}(P \parallel P') = Q(X \ne Y).$$

(b) Variational Forms: $\mathrm{TV}(P \parallel P') = \sup_{\mathcal{E}} P(\mathcal{E}) - P'(\mathcal{E})$ *(that is, without the absolute value). Moreover, if $E, E'$ denote the associated expectations, and letting $V$ quantify $[0, 1]$-bounded random variables on $(\Omega, \mathscr{F})$,*

$$\mathrm{TV}(P \parallel P') = \sup\{E[V] - E'[V] \text{ s.t. } V : (\Omega, \mathscr{F}) \to [0, 1]\}$$

(c) Symmetry: $\mathrm{TV}(P \parallel P') = \mathrm{TV}(P' \parallel P)$.

(d) Triangle Inequality:

$$\mathrm{TV}(P \parallel P'') \le \mathrm{TV}(P \parallel P') + \mathrm{TV}(P' \parallel P'').$$

(e) Data Processing: *Let $(X, Y)$ be random variables on $(\Omega, \mathscr{F})$. Then*

$$\mathrm{TV}(P(X) \parallel P'(X)) \le \mathrm{TV}(P(X, Y) \parallel P'(X, Y)).$$

(f) Tensorization: *Let $(X_1, \ldots, X_n)$ be $n$ random variables on $(\Omega, \mathscr{F})$ which are independent under both $P$ and $P'$. Then,*

$$\mathrm{TV}(P(X_1, \ldots, X_n) \parallel P'(X_1, \ldots, X_n)) \le \sum_{i=1}^{n} \mathrm{TV}(P(X_i) \parallel P'(X_i)).$$

*Proof.* The coupling and variational forms can be found in [Lin02, Chapter 1]. Symmetry follows immediately from the definition.

To see the triangle inequality, note that for any measurable $\mathcal{E} \subset \Omega$, we have

$$|P(\mathcal{E}) - P''(\mathcal{E})| \le |P(\mathcal{E}) - P'(\mathcal{E})| + |P'(\mathcal{E}) - P''(\mathcal{E})| \le \mathrm{TV}(P \parallel P') + \mathrm{TV}(P' \parallel P'').$$

As the above holds for any such $\mathcal{E}$, we can conclude

$$\text{TV}(P \parallel P'') \leq \text{TV}(P \parallel P') + \text{TV}(P' \parallel P'').$$

For the data processing inequality, say $(X, Y)$ follow distribution $P$ and $(X', Y')$ follow $P'$, and let $Q$ be the maximal coupling of $P(X, Y)$ and $P'(X, Y)$. Then

$$\text{TV}(P(X) \parallel P'(X)) \leq Q(X \neq X') \leq Q((X, Y) \neq (X', Y')) = \text{TV}(P(X, Y) \parallel P'(X, Y)).$$

To prove the tensorization inequality, say $(X_1, \ldots, X_n)$ follow $P$ and $(X_1', \ldots, X_n')$ follow $P'$. For each $i$, let $Q_i$ be the maximal coupling of $P(X_i)$ and $P'(X_i')$, and let $Q$ denote the product distribution of the $Q_i$'s. Note that $Q$ is a valid coupling of $P$ and $P'$, which are each product distributions. Then we have

$$\text{TV}(P \parallel P') \leq Q(X \neq X') \leq \sum_{i=1}^{n} Q_i(X_i \neq X_i') = \sum_{i=1}^{n} \text{TV}(P(X_i) \parallel P'(X_i')). \qquad \square$$

**Lemma B.2** (Total Variation with Shared Marginal). *Let $P$ and $P'$ be joint distributions over random variables $(X, Y)$ such that the* marginals $P(X)$ *and* $P'(X)$ *coincide. Then,*

$$\text{TV}(P(X, Y) \parallel P'(X, Y)) = \mathbb{E}_{X \sim P} \text{TV}(P(Y \mid X) \parallel P'(Y \mid X)).$$

*Proof.* We first show that $\text{TV}(P(X, Y) \parallel P'(X, Y)) \geq \mathbb{E}_{X \sim P} \text{TV}(P(Y \mid X) \parallel P'(Y \mid X))$. To see this let $(B_x)_{x \in \Omega}$ be any set of measurable events indexed by $\Omega$. Letting $V(X, Y) = \mathbb{1}[Y \in B_X]$, the variational form of total variation distance (Lemma B.1) implies that

$$\mathbb{E}_{(X,Y) \sim P}[V(X, Y)] - \mathbb{E}_{(X,Y) \sim P'}[V(X, Y)] \leq \text{TV}(P(X, Y) \parallel P'(X, Y)).$$

On the other hand, because $P(X) = P'(X)$, we have

$$\mathbb{E}_{(X,Y) \sim P}[V(X, Y)] - \mathbb{E}_{(X,Y) \sim P'}[V(X, Y)] = \mathbb{E}_{X \sim P}[P(Y \in B_X) - P'(Y \in B_X)].$$

Since the choice of $(B_x)_{x \in \Omega}$ was arbitrary, we can conclude that

$$\text{TV}(P(X, Y) \parallel P'(X, Y)) \geq \mathbb{E}_{X \sim P} \text{TV}(P(Y \mid X) \parallel P'(Y \mid X)).$$

Now to prove $\text{TV}(P(X, Y) \parallel P'(X, Y)) \leq \mathbb{E}_{X \sim P} \text{TV}(P(Y \mid X) \parallel P'(Y \mid X))$, we construct a coupling $Q((X, Y), (X', Y'))$ of $P(X, Y)$ and $P'(X, Y)$ as follows. First draw $X \sim P(X)$ and set $X' = X$. Then let $(Y, Y')$ be drawn from the maximal coupling of $P(Y \mid X)$ and $P'(Y \mid X)$ (guaranteed by Lemma B.1). By construction, this satisfies that $Q(X, Y) = P(X, Y)$ and $Q(X', Y') = P'(X, Y)$. By the coupling inequality (Lemma B.1), we have

$$\text{TV}(P(X, Y) \parallel P'(X, Y)) \leq Q((X, Y) \neq (X', Y')) = \mathbb{E}_{X \sim P}[\text{TV}(P(Y \mid X) \parallel P'(Y \mid X))].$$
$$\square$$

**Lemma B.3** (Total Variation with Shared Conditional). *Let $P$ and $P'$ be joint distributions over random variables $(X, Y)$ such that the* conditionals $P(Y \mid X)$ *and* $P'(Y \mid X)$ *coincide. Then,*

$$\text{TV}(P(X, Y) \parallel P(X, Y)) = \text{TV}(P(X) \parallel P'(X)).$$

*Proof.* By the data processing property of total variation (Lemma B.1), we know

$$\text{TV}(P(X, Y) \parallel P(X, Y)) \geq \text{TV}(P(X) \parallel P'(X)).$$

To prove the lemma, we need to show that the opposite inequality also holds. To do so, let $Q_X$ be the maximal coupling of $P(X)$ and $P'(X)$, and let $Q$ denote the distribution over $((X, Y), (X', Y'))$ induced by first drawing $(X, X')$ from $Q_X$ and then drawing $Y, Y'$ as follows:

- If $X = X'$, draw $Y \sim P(Y \mid X)$ and set $Y' = Y$.
- Otherwise, draw $Y$ and $Y'$ independently from $P(Y \mid X)$ and $P'(Y' \mid X')$, respectively

It is clear that $Q(X, Y) = P(X, Y)$. To show that $Q$ is a valid coupling, it remains to check that $Q(X', Y') = P'(X, Y)$. Since $Q'(X) = P'(X)$ by construction, it suffices to check that

$Q(Y \mid X' = x') = P'(Y \mid X = x')$ for all $x'$ in the (almost-sure) support of $P(X')$. This follows since

$$Q(Y \mid X' = x', X) = \begin{cases} P(Y \mid X = x') & \text{if } X = x' \\ P'(Y \mid X = x') & \text{if } X \neq x' \end{cases} = P'(Y \mid X = x')$$

where we use $P(Y \mid X) = P'(Y \mid X)$. Hence, marginalizing over $X$, $Q(Y \mid X') = P'(Y \mid X = x')$, as needed. Lastly, observe that our construction of $Q$ ensures $Y = Y'$ whenever $X = X'$. Therefore, we conclude

$$\mathrm{TV}(P(X, Y) \parallel P'(X, Y)) \leq Q((X, Y) \neq (X', Y')) = Q_X(X \neq X') = \mathrm{TV}(P(X) \parallel P(X')).$$

$\square$

**Lemma B.4** (Coupled Transport Form). *Let $P$ and $P'$ be joint distributions over random variables $(X, Y)$ with coinciding marginals $P(X) = P(X')$ in the first variable. Then there exists a distribution $Q(X, Y, Y')$ whose marginals satisfy $Q(X, Y) = P(X, Y)$ and $Q(X, Y') = P'(X, Y)$, and for which we have*

$$\mathrm{TV}(P(X, Y) \parallel P'(X, Y)) = Q[Y \neq Y'].$$

*Proof.* We construct $Q(X, Y, Y')$ as follows. First draw $X \sim P(X)$. Then let $(Y, Y')$ be drawn from the maximal coupling of $P(Y \mid X)$ and $P'(Y \mid X)$ (guaranteed by Lemma B.1). By construction, this satisfies that $Q(X, Y) = P(X, Y)$ and $Q(X, Y') = P'(X, Y)$. Moreover, one can see that

$$Q[Y \neq Y'] = \mathbb{E}_{X \sim P} \mathrm{TV}(P(Y \mid X) \parallel P'(Y \mid X)) = \mathrm{TV}(P(X, Y) \parallel P'(X, Y)),$$

where the first equality follows from the use of the maximal coupling of conditional distributions and the second equality is Lemma B.2. $\square$

## B.2 General Sensitivity Bounds: Generalizing Theorem 3.2

In general, we address priors over means which are unbounded. We use the following functional to control expectation over their upper tails:

**Definition B.2** (Upper Tail Expectation). Let $X$ be a nonnegative random variable on a probability space $(\Omega, \mathscr{F})$ with law $\mathbb{P}$ and finite expectation $\mathbb{E}[X] < \infty$. We define its tail expectation, as a function of probabilities $p \in (0, 1]$, as

$$\Psi_X(p) := \frac{1}{p} \sup_Y \mathbb{E}[XY]$$
$$\text{s.t. } Y : (\Omega, \mathscr{F}) \to [0, 1] \text{ and } \mathbb{E}[Y] \leq p.$$

For $p > 1$, we extend $\Psi_X(p) = \mathbb{E}[X]$. Overloading notation, we let $\Psi_\theta(p)$ denote the upper tail function over $\mathrm{diam}(\boldsymbol{\mu})$ when drawn from $P_\theta$:

$$\Psi_\theta(p) := \Psi_{\mathrm{diam}(\boldsymbol{\mu})}(p) \quad \text{where } \boldsymbol{\mu} \sim P_\theta.$$

By taking conditional expectations, one can equivalently verify that $\Psi_X(p) := \frac{1}{p} \sup_f \mathbb{E}[X f(X)]$ is the supremal expected correlation between $X$ and $f(X)$, over functions $f : [0, 1] \to \mathbb{R}$ satisfying $\mathbb{E}[f(X)] = p$. Intuitively, $\Psi_\theta(p)$ considers large how conditional expectation of $\frac{1}{p} \mathbb{E}[X f(X)]$ can be made by concentrating all the mass of $f$ on the upper tail of $X$. We establish key properties, estimates, and a closed form for $\Psi_\theta(p)$ in terms of quantiles of $X$ in Appendix B.3.

Given this definition, our general sensitivity bound takes the following form:

**Theorem B.1.** *Let $\mathrm{alg}(\cdot)$ be an $n$-Monte Carlo family of algorithms on horizon $H \in \mathbb{N}$, and let $\theta, \theta' \in \Theta$. Setting $\varepsilon = \mathrm{TV}(P_\theta \parallel P_{\theta'})$, we have that*

$$|R(\theta, \mathrm{alg}(\theta)) - R(\theta, \mathrm{alg}(\theta'))| \leq 2nH^2 \varepsilon \cdot \Psi_\theta(2nH\epsilon),$$

*where $\Psi_\theta(\cdot)$ is the tail expectation defined in Definition B.2.*

We specialize upper bounds on the upper tail expectation for priors satsifying the following tail conditions:

**Definition B.3** (Tail Conditions). We set $\bar{\boldsymbol{\mu}}_\theta := E_\theta[\boldsymbol{\mu}]$. We say that $P_\theta$ is

(a) $B$-bounded if $P_\theta[\mathrm{diam}(\boldsymbol{\mu}) \le B] = 1$.
(b) Coordinate-wise $\sigma^2$-sub-Gaussian if for all $a \in \mathcal{A}$,

$$P_\theta(|\mu_a - \bar{\mu}_a| \ge t) \le 2\exp\left(-\frac{t^2}{2\sigma^2}\right).$$

(c) Coordinate-wise $(\sigma^2, \nu)$-sub-Gamma if for all $a \in \mathcal{A}$,

$$P_\theta(|\mu_a - \bar{\mu}_a| \ge t) \le 2\max\left\{\exp\left(-\frac{t^2}{2\sigma^2}\right), \exp\left(-\frac{t}{2\nu}\right)\right\}.$$

For priors satisfying the above tail conditions, Theorem B.1 specializes as follows:

**Theorem B.2.** *Let $\mathsf{alg}(\cdot)$ be an $n$-Monte Carlo family of algorithms on horizon $H \in \mathbb{N}$, and let $\theta, \theta' \in \Theta$. Setting $\varepsilon = \mathrm{TV}(P_\theta \| P_{\theta'})$, we have the following guarantees.*

(a) *If $P_\theta$ is B-bounded, then $|R(\theta, \mathsf{alg}(\theta)) - R(\theta, \mathsf{alg}(\theta'))| \le 2nH^2 \varepsilon B$.*
(b) *If $P_\theta$ is coordinate-wise $\sigma^2$-sub-Gaussian and $\epsilon$, then*

$$|R(\theta, \mathsf{alg}(\theta)) - R(\theta, \mathsf{alg}(\theta'))| \le 2nH^2\varepsilon\left(\mathrm{diam}(E_\theta[\boldsymbol{\mu}]) + \sigma\left(8 + 5\sqrt{\log\left(\frac{|\mathcal{A}|^2}{\min\{1, 2nH\epsilon\}}\right)}\right)\right)$$

(c) *If $P_\theta$ is coordinate-wise $(\sigma^2, \nu)$-sub-Gamma, then*

$$|R(\theta, \mathsf{alg}(\theta)) - R(\theta, \mathsf{alg}(\theta'))|$$
$$\le 2nH^2\varepsilon\left(\mathrm{diam}(E_\theta[\boldsymbol{\mu}]) + \sigma\left(8 + 5\sqrt{\log\left(\frac{|\mathcal{A}|^2}{\min\{1, 2nH\epsilon\}}\right)}\right) + \nu\left(11 + 7\log\left(\frac{|\mathcal{A}|^2}{\min\{1, 2nH\epsilon\}}\right)\right)\right).$$

The proof of Theorem B.2 is a direct consequence of Theorem B.1 and the estimates from Lemma B.6 given in Appendix B.4. Note that Theorem 3.2 comprises of the first two statements of Theorem B.2.

### B.3 Quantiles, CDFs and Tail Expectations

Recall classical definitions of quantile and CDF:

**Definition B.4** (Quantile and CDF). Given a real-valued random variable $X$ with law $P$, we define its cumulative distribution function, or CDF, by $F_X(t) := P[X \le t]$, and the *quantile function* $q_X(p) := \inf\{t : 1 - F_X(t) \le p\}$.

With these definitions in place, we expose the essential properties of $\Psi_X(p)$:

**Lemma B.5** (Properties of the Upper Tail expectation). *Then upper tail expectation satisfies the following properties:*

(a) Monotonicity: *$p \mapsto \Psi_X(p)$ is non-increasing in $p$, and $p \mapsto p \cdot \Psi_X(p)$ is non-decreasing.*
(b) Dominance Preservation: *Let $X'$ stochastically dominate $X$, that is, $F_X(t) \ge F_{X'}(t)$ for all $t$. Then, $\Psi_X(p) \le \Psi_{X'}(p)$ for all $p$.*
(c) Translation: *$\Psi_X(p) \le c + \Psi_{\max\{X-c,\, 0\}}(p)$ for any constant $c > 0$.*
(d) Closed Form: *We have that*

$$\Psi_X(p) = \frac{1}{p}\mathbb{E}[Xg_p(X)], \quad \text{where}$$
$$g_p(u) := \mathbb{I}\{u > q_X(p)\} + (p - \mathbb{P}[X > q_X(p)])\,\mathbb{I}\{u = q_X(p)\}.$$

*In particular, if $F_X(\cdot)$ is continuous, then*

$$\Psi_X(p) = \frac{1}{p}\mathbb{E}[X\mathbb{I}\{X \ge q_X(p)\}] = \mathbb{E}[X \mid X \ge q_X(p)]$$

(e) Useful Estimate: *For any $\alpha > 0$ and $p \in (0, 1]$*

$$\Psi_X(p) \leq \frac{p - \mathbb{P}[X > q_p(X)]}{p} + \sum_{i=0}^{\infty} \alpha^{-i} q_X(\alpha^{-i-1} p)$$

*In particular, if $F_X(\cdot)$ is continuous, then*

$$\Psi_X(p) \leq \sum_{i=0}^{\infty} \alpha^{-i} q_X(\alpha^{-i-1} p).$$

*Proof.* (a) We can rewrite

$$\Psi_X(p) := \sup_Y \mathbb{E}[XY]$$

$$\text{s.t. } Y : (\Omega, \mathscr{F}) \to [0, \tfrac{1}{p}] \text{ and } \mathbb{E}[Y] \leq 1. \tag{B.1}$$

Hence, the constraint on $Y$ becomes strictly less restrictive as $p$ decreases, meaning that $\Psi_X(p)$ is non-increasing. Similarly, $p \cdot \Psi_X(p)$ is the supremum over $\mathbb{E}[XY]$ with $Y \in [0, 1]$ and $\mathbb{E}[Y] \leq p$, so the constraint becomes more restrictive as $p$ decays, and thus $p \mapsto p\Psi_X(p)$ is non-decreasing.

(b) Stochastic domination implies that one can construct a joint distribution $(X, X')$ such that $X' \geq X$ almost surely (see for example the coupling at the beginning of Section 2.3.1 in [SWDN09]). This implies that, for any $Y \geq 0$ jointly distributed with $X$ via $(X, Y)$, we can create a joint distribution $(X', Y)$ such that $\mathbb{E}[XY] \leq \mathbb{E}[X'Y]$. The bound follows.

(c) For any random variable $Y \in [0, 1]$ with $\mathbb{E}\,Y = p$, we have $\frac{1}{p}\,\mathbb{E}[XY] = c + \frac{1}{p}\,\mathbb{E}[(X - c)Y] \leq c + \frac{1}{p}\,\mathbb{E}[\max\{X - c, 0\}Y] \leq c + \Psi_{\max\{X-c,0\}}(p)$.

(d) It is clear from the definition that $\mathbb{E}[g_p(X)] = p$ and $0 \leq g_p(\cdot) \leq 1$. Hence, $\mathbb{E}[Xg_p(X)] \leq \Psi_X(p)$. To prove the converse, first observe that for any random variable $Y$, we have

$$\mathbb{E}[XY] = \mathbb{E}[X\,\mathbb{E}[Y \mid X]].$$

Since $X$ is non-negative, it suffices to restrict our attention to random variables of the form $Y = f(X)$ where $f : \mathbb{R} \to [0, 1]$ and $\mathbb{E}[f(X)] = p$. We will show that for any such function $f$, if we do not have $f(X) = g_p(X)$ almost surely, then we must have $\mathbb{E}[Xf(X)] < \mathbb{E}[Xg_p(X)]$. To see this, it suffices to show that (i) conditioned on the event $X > q_X(p)$, we must have $f(X) = 1$ almost surely, and (ii) if $\mathbb{P}[X = q_X(p)] > 0$, then conditioned on the event $X = q_X(p)$, we must have $f(X) = p - \mathbb{P}[X = q_X(p)]$ almost surely. As the arguments are symmetrical, we will only provide the proof of (i).

Suppose that (i) does not hold. Then there exist sets $S_+ \subset (q_X(p), \infty)$ and $S_- \subset [0, q_X(p)]$ such that $P(S_+), P(S_-) > 0$, $\mathbb{E}[f(X) \mid X \in S_+] < 1$, and $\mathbb{E}[f(X) \mid S_-] > 0$. Define the function $h : \mathbb{R} \to [0, 1]$ satisfying

$$h(x) = \begin{cases} \alpha_+ + (1 - \alpha_+)f(x) & \text{if } x \in S_+ \\ (1 - \alpha_-)f(x) & \text{if } x \in S_- \\ f(x) & \text{otherwise.} \end{cases}$$

where $\alpha_+ = P(S_-)\,\mathbb{E}[f(X) \mid X \in S_-]$ and $\alpha_- = P(S_+)\,\mathbb{E}[1 - f(X) \mid X \in S_+]$. By assumption, we have $\alpha_-, \alpha_+ \in [0, 1]$, so that $h(x) \in [0, 1]$. Further, we can calculate

$$\begin{aligned} \mathbb{E}[h(X)] &= \mathbb{E}[f(X)\mathbb{1}[X \notin S_+ \cup S_-]] + \mathbb{E}[(\alpha_+ + (1 - \alpha_+)f(X))\mathbb{1}[X \in S_+]] \\ &\quad + \mathbb{E}[(1 - \alpha_-)f(X)\mathbb{1}[X \in S_-]] \\ &= \mathbb{E}[f(X)] + \alpha_+ P(S_+)\mathbb{E}[1 - f(X) \mid X \in S_+] + \alpha_- P(S_-)\,\mathbb{E}[f(X) \mid X \in S_-] \\ &= \mathbb{E}[f(X)] = p. \end{aligned}$$

On the other hand, we can see that

$$\mathbb{E}[Xh(X)] - \mathbb{E}[Xf(X)]$$
$$= \mathbb{E}[f(X)X\mathbb{1}[X \notin S_+ \cup S_-]] + \mathbb{E}[(\alpha_+ + (1-\alpha_+)f(X))X\mathbb{1}[X \in S_+]]$$
$$\quad + \mathbb{E}[(1-\alpha_-)f(X)X\mathbb{1}[X \in S_-]] - \mathbb{E}[Xf(X)]$$
$$= \alpha_+ P(S_+)\,\mathbb{E}[(1-f(X))X \mid X \in S_+] - \alpha_- P(S_-)\,\mathbb{E}[f(X)X \mid X \in S_-]$$
$$= P(S_+)P(S_-)\,\mathbb{E}[1-f(X) \mid X \in S_+]\,\mathbb{E}[f(X) \mid X \in S_-]$$
$$\quad \cdot \left( \mathbb{E}\left[ \frac{(1-f(X))X}{\mathbb{E}[1-f(X) \mid X \in S_+]} \;\Big|\; X \in S_+ \right] - \mathbb{E}\left[ \frac{f(X)X}{\mathbb{E}[f(X) \mid X \in S_-]} \;\Big|\; X \in S_- \right] \right)$$
$$> 0.$$

where the last inequality comes from the fact that

$$\mathbb{E}\left[ \frac{(1-f(X))X}{\mathbb{E}[1-f(X) \mid X \in S_+]} \;\Big|\; X \in S_+ \right]$$

is a convex combination of elements from $S_+$ and

$$\mathbb{E}\left[ \frac{f(X)X}{\mathbb{E}[f(X) \mid X \in S_-]} \;\Big|\; X \in S_- \right]$$

is a convex combination of elements from $S_-$, and every element in $S_-$ is strictly smaller than every element of $S_+$.

(e) For $\epsilon > 0$, define the sequence of integers $t_i = t_i(\epsilon) = 2^i\epsilon + q_p(\alpha^i X)$. Note that $\lim_{i \to \infty} \mathbb{P}[X \le t_i] = 1$ for each $\epsilon$. Hence,

$$\Psi_p(X) = \frac{1}{p}\mathbb{E}[Xg_{p,X}(X)]$$

$$\le \frac{1}{p}\mathbb{E}[Xg_p(X)\mathbb{I}\{X \le t_0\}] + \frac{1}{p}\sum_{i=0}^{\infty}\mathbb{E}[Xg_p(X) \cdot \mathbb{I}\{t_i < X < t_{t+i}\}]$$

$$\le 2\epsilon + \frac{1}{p}q_X(p)\,\mathbb{E}[\mathbb{I}\{X \le t_0\}g_p(X)] + \frac{1}{p}\sum_{i=0}^{\infty} q_p(\alpha^{-(i+1)}X)\,\mathbb{P}[X > t_i]$$

Since $t_i > q_X(\alpha^{-i}p)$, we have that $\mathbb{P}[X > t_i] \le \alpha^{-i}p$. Thus, taking $\epsilon \to 0$,

$$\Psi_p(X) \le \frac{1}{p}q_X(p)\lim_{\epsilon \to 0}\mathbb{E}[\mathbb{I}\{X \le t_0(\epsilon)\}g_p(X)] + \sum_{i=0}^{\infty} q_X(\alpha^{-(i+1)}X)\alpha^{-i}p.$$

Finally, we observe that $\mathbb{E}[\mathbb{I}\{X \le t_0(\epsilon)\}g_p(X)] \le \mathbb{P}[q_X(p) < X < q_X(p) + \epsilon] + \mathbb{E}[\mathbb{I}\{X \le q_X(p)\}g_p(X)\}$. By continuity of probability measures, the first term tends to 0 as $\epsilon \to 0$. The second term is precisely $p - \mathbb{P}[X > q_X(p)]$. The first bound follows. Note that for continuous CDFs, we necessarily have that $\mathbb{P}[X > q_p(X)] = p$, yielding the specialization to continuous CDFs.

$\square$

## B.4 Upper Tail Expectations under Tail Conditions

Each of the conditions in Definition B.3 yields a transparent upper bound on $\Psi_\theta(p)$:

**Lemma B.6.** *Let $\bar{\mu}_\theta = E_\theta[\mu]$. Then, for $p \in [0,1]$.*

(a) *If $P_\theta$ is B-bounded, then $\Psi_\theta(p) \le B$ for all $p$.*

(b) *If $P_\theta$ is coordinate-wise $\sigma^2$-sub-Gaussian and $\mathcal{A}$ is finite, then*

$$\Psi_\theta(p) \le \operatorname{diam}(\bar{\mu}_\theta) + \sigma(8 + 5\sqrt{\log \tfrac{2|\mathcal{A}|}{p}})$$

(c) *If $P_\theta$ is coordinate-wise $(\sigma^2, \nu)$-sub-Gamma and $\mathcal{A}$ is finite, then*

$$\Psi_\theta(p) \le \operatorname{diam}(\bar{\mu}_\theta) + \sigma(8 + 5\sqrt{\log \tfrac{2|\mathcal{A}|}{p}}) + \nu(11 + 7\log \tfrac{2|\mathcal{A}|}{p}).$$

*By Definition B.2, the above extend to $p \ge 1$ by replacing $p \leftarrow \min\{1, p\}$.*

The bounds in parts (b) and (c) may be extended to infinite $\mathcal{A}$ via covering arguments.

*Proof of Lemma B.6.* We prove each part in sequence

(a) Suppose $\Psi_\theta$ is $B$-bounded. Then, for any random variable $Y \in [0,1]$ with $\mathbb{E}[Y] = p$, we have $\frac{1}{p}\mathbb{E}[\mathrm{diam}(\boldsymbol{\mu})Y] \leq \frac{1}{p}B\,\mathbb{E}[Y] = B$. Hence, $\Psi_\theta(p) \leq B$.

(b) Define the random variable $X = \max\{0, \mathrm{diam}(\boldsymbol{\mu}) - \mathrm{diam}(E_\theta[\boldsymbol{\mu}])\}$. By Lemma B.5 part (c), we have
$$\Psi_\theta(p) \leq \mathrm{diam}(E_\theta[\boldsymbol{\mu}]) + \Psi_X(p).$$
Further, observe that
$$X = \max\{0, \mathrm{diam}(\boldsymbol{\mu}) - \mathrm{diam}(E_\theta[\boldsymbol{\mu}])\} \leq 2\max_{a \in \mathcal{A}}|\mu_a - E_\theta[\mu_a]|. \tag{B.2}$$
By a union bound over all $a \in \mathcal{A}$, the sub-Gaussian tail implies
$$F_X(t) = P_\theta[X \leq t] \leq F_{X'}(t), \quad \text{where } F_{X'}(t) := 1 - \begin{cases} 1 & t \leq 0 \\ \min\left\{1, 2|\mathcal{A}|e^{-\frac{t^2}{8\sigma^2}}\right\} & t > 0, \end{cases}$$
where above $e^{-\frac{t^2}{8\sigma^2}} = e^{-\frac{(t/2)^2}{2\sigma^2}}$ accounts for the factor of 2 in Equation (B.2), and minimum with 1 accounts for boundedness of probabilities.

Note that $F_{X'}(t)$ is continuous and is a valid CDF of a random variable, say $X'$, which stochastically dominates $X$ (that is, $F_X(t) \leq F_{X'}(t)$ for all $t$). Hence, Lemma B.5 part (b) implies that $\Psi_X(p) \leq \Psi_{X'}(p)$. Moreover, we can compute that the quantile function of $X'$ is
$$q_{X'}(p) = 2\sqrt{2\sigma^2 \log\frac{2|\mathcal{A}|}{p}}.$$

Hence, by continuity of $X'$, Lemma B.5 part (e) implies
$$\Psi_{X'}(p) \leq 2\sum_{i=0}^{\infty} e^{-i}q_{X'}(e^{-i-1}p)$$
$$\leq 2\sum_{i=0}^{\infty} e^{-i}\sqrt{2\sigma^2 \log\frac{2|\mathcal{A}|}{e^{i+1}p}}$$
$$\leq 2\left(1 - \frac{1}{e}\right)^{-1}\sqrt{2\sigma^2 \log\frac{2|\mathcal{A}|}{p}} + 2\sqrt{2\sigma^2}\sum_{i=0}^{\infty}\underbrace{\sqrt{(i+1)}}_{\leq i+1}e^{-i}$$
$$\leq 2\left(1 - \frac{1}{e}\right)^{-1}\sqrt{2\sigma^2 \log\frac{2|\mathcal{A}|}{p}} + 2\sqrt{2\sigma^2}\left(1 - \frac{1}{e}\right)^{-2}$$
$$\leq \sigma\left(8 + 5\sqrt{\log\frac{2|\mathcal{A}|}{p}}\right).$$

(c) The proof is analogous, except now we use the bound
$$q_{X'}(p) \leq 2\sqrt{2\sigma^2 \log\frac{2|\mathcal{A}|}{p}} + 4\nu \log\frac{2|\mathcal{A}|}{p}.$$
After some computation, this yields
$$\Psi_{X'}(p) \leq 2(1 - \frac{1}{e})^{-1}\sqrt{2\sigma^2 \log\frac{2|\mathcal{A}|}{p}} + 2\sqrt{2\sigma^2}\left(1 - \frac{1}{e}\right)^{-2}$$
$$+ 4\left(1 - \frac{1}{e}\right)^{-1}\nu \log\frac{2|\mathcal{A}|}{p} + 4\nu\left(1 - \frac{1}{e}\right)^{-2}$$
$$\leq \sigma(8 + 5\sqrt{\log\frac{2|\mathcal{A}|}{p}}) + \nu(11 + 7\log\frac{2|\mathcal{A}|}{p}). \qquad \square$$

## B.5 A warmup to Proposition 3.4

To illustrate the concepts involved in the proof, we will directly analyze Thompson sampling in the case of Gaussian priors and Gaussian rewards. Specifically, we will assume that the prior distributions are of the form $\mathcal{N}(\theta, I_{|\mathcal{A}|})$ and the reward distributions are of the form $\mathcal{N}(\mu, \sigma^2)$, where $\sigma^2$ is known, in which case we can show the (slightly tighter) bound of

$$\text{TV}(P_{\theta,\mathsf{TS}(\theta)}(\boldsymbol{\mu}, \tau_H) \parallel P_{\theta,\mathsf{TS}(\theta')}(\boldsymbol{\mu}, \tau_H)) \leq H \cdot \sum_{a \in \mathcal{A}} \text{TV}(\mathcal{N}(\theta_a, 1) \parallel \mathcal{N}(\theta'_a, 1)). \tag{B.3}$$

We construct a coupling of $P_{\theta,\mathsf{TS}(\theta)}(\boldsymbol{\mu}, \tau_H)$ and $P_{\theta,\mathsf{TS}(\theta')}(\boldsymbol{\mu}, \tau_H)$ inductively, using the notation that $\boldsymbol{\mu}, \tau_H$ marginally is distributed according to $P_{\theta,\mathsf{TS}(\theta)}(\boldsymbol{\mu}, \tau_H)$ and $\boldsymbol{\mu}', \tau'_H$ is marginally distributed according to $P_{\theta,\mathsf{TS}(\theta')}(\boldsymbol{\mu}, \tau_H)$. The construction is as follows.

- First, draw $\boldsymbol{\mu} \sim P_\theta$ and set $\boldsymbol{\mu}' = \boldsymbol{\mu}$.
- For $h = 1, 2, \ldots, H$:
  - If $\tau_{h-1} = \tau'_{h-1}$:
    * Let $\hat{\mu}_h, \hat{\mu}'_h$ be drawn from the maximal coupling of the posteriors $P_\theta(\mu \mid \tau_h)$ and $P_{\theta'}(\mu \mid \tau'_h)$.
    * Play actions $a_h = \arg\max_a \hat{\mu}_{h,a}$ and $a'_h = \arg\max_a \hat{\mu}'_{h,a}$.
    * Draw $r_h \sim \mathcal{N}(\boldsymbol{\mu}_{a_h}, \sigma^2)$. If $a_h = a'_h$, set $r'_h = r_h$. Otherwise, draw $r'_h \sim \mathcal{N}(\boldsymbol{\mu}_{a'_h}, \sigma^2)$.
  - Otherwise: run $\mathsf{TS}(\theta)$ and $\mathsf{TS}(\theta')$ independently.

It is not too hard to see that this produces a valid coupling as the marginals are respected throughout the construction. Letting $Q$ denote the joint measure of this coupling, we have

$$\text{TV}(P_{\theta,\mathsf{TS}(\theta)}(\boldsymbol{\mu}, \tau_H) \parallel P_{\theta,\mathsf{TS}(\theta')}(\boldsymbol{\mu}, \tau_H)) \leq Q((\boldsymbol{\mu}, \tau_H) \neq (\boldsymbol{\mu}', \tau'_H))$$
$$\leq Q(\boldsymbol{\mu} \neq \boldsymbol{\mu}') + Q(\tau_H \neq \tau'_H \mid \boldsymbol{\mu} = \boldsymbol{\mu}')$$
$$\leq Q(\boldsymbol{\mu} \neq \boldsymbol{\mu}') + \sum_{h=1}^{H} Q((a_h, r_h) \neq (a'_h, r'_h) \mid \boldsymbol{\mu} = \boldsymbol{\mu}', \tau_{h-1} = \tau'_{h-1})$$
$$\leq \sum_{h=1}^{H} Q(\hat{\mu}_h \neq \hat{\mu}'_h \mid \boldsymbol{\mu} = \boldsymbol{\mu}', \tau_{h-1} = \tau'_{h-1}).$$

Here, the first inequality follows from Lemma B.1 and the last inequality comes from the particular construction of the coupling $Q$. We now turn to bounding this last quantity. We will require two lemmas. The first of these gives the form of the posterior distributions in this setting.

**Lemma B.7** ([Mur12]). *Let $\theta \in \mathbb{R}$. Consider the generative model where $\mu \sim \mathcal{N}(\theta, 1)$ and $x_1, \ldots, x_n \sim \mathcal{N}(\mu, \sigma^2)$. Then the posterior distribution of $\mu$ given $x_1, \ldots, x_n$ is $\mathcal{N}(\theta_n, \sigma_n^2)$ where*

$$\theta_n = \theta \cdot \frac{\sigma^2}{\sigma^2 + n} + \bar{x}_n \cdot \frac{n}{\sigma^2 + n}$$
$$\sigma_n^2 = \frac{\sigma^2}{\sigma^2 + n}$$

*and $\bar{x}_n$ is the empirical mean of $x_1, \ldots, x_n$.*

Lemma B.7 implies that for any fixed trajectory $\tau_h$, the posterior $P_\theta(\mu \mid \tau_h)$ is $\mathcal{N}(\theta_h, \Sigma_h)$

$$\theta_{h,a} = \theta \cdot \frac{\sigma^2}{\sigma^2 + n_{h,a}} + \bar{r}_{h,a} \cdot \frac{n_{h,a}}{\sigma^2 + n_{h,a}}$$

and $\Sigma_h$ is a diagonal matrix satisfying

$$\Sigma_{h,a,a} = \frac{\sigma^2}{\sigma^2 + n_{h,a}}.$$

Here we have used the convention that $n_{h,a}$ is the number of times that arm $a$ has been pulled at time $h$ and $\bar{r}_{h,a}$ is the mean of the observed rewards for arm $a$ at time $h$.

The next lemma quantifies the total variation distance between two Gaussians that share the same covariance matrix.

**Lemma B.8.** *Let $\theta, \theta' \in \mathbb{R}^d$ and let $\Sigma \in \mathbb{R}^{d \times d}$ be symmetric positive definite. The total variation distance between normal distributions $\mathcal{N}(\theta, \Sigma)$ and $\mathcal{N}(\theta', \Sigma)$ is exactly $2\Phi\left(\frac{\|\Sigma^{-1/2}(\theta - \theta')\|}{2}\right) - 1$, where $\Phi(\cdot)$ is the c.d.f. for the standard normal distribution.*

*Proof.* Let $\nu_\theta$ and $\nu_{\theta'}$ denote the measures of $\mathcal{N}(\theta, \Sigma)$ and $\mathcal{N}(\theta', \Sigma)$, respectively. Similarly, let $q_\theta$ and $q_{\theta'}$ denote their respective densities. It is not too hard to see that when the measures admit densities the variational form of total variation distance in Lemma B.1 is attained at the set $\mathcal{E} = \{x \in \mathbb{R}^d \ : \ q_\theta(x) \geq q_{\theta'}(x)\}$, i.e. we have

$$\text{TV}(\nu_\theta \parallel \nu_{\theta'}) = \nu_\theta(\mathcal{E}) - \nu_{\theta'}(\mathcal{E}).$$

By expanding the densities $q_\theta$ and $q_{\theta'}$, we have

$$\mathcal{E} = \left\{x \in \mathbb{R}^d \ : \ x^\top \Sigma^{-1}(\theta - \theta') \geq \frac{1}{2}\left(\theta^\top \Sigma^{-1}\theta - \theta'^\top \Sigma^{-1}\theta'\right)\right\}.$$

Observe that for $X \sim \mathcal{N}(\theta, \Sigma)$, we have $X^\top \Sigma^{-1}(\theta - \theta')$ is distributed as $\mathcal{N}(\theta^\top \Sigma^{-1}(\theta - \theta'), \sigma^2)$, where $\sigma^2 = (\theta - \theta')^\top \Sigma^{-1}(\theta - \theta')$. Thus, we can calculate

$$\nu_\theta(\mathcal{E}) = P_{Z \sim \mathcal{N}(\theta^\top \Sigma^{-1}(\theta - \theta'), \sigma^2)}\left(Z \geq \frac{1}{2}\left(\theta^\top \Sigma^{-1}\theta - \theta'^\top \Sigma^{-1}\theta'\right)\right)$$

$$= 1 - \Phi\left(\frac{-\frac{1}{2}\theta^\top \Sigma^{-1} - \frac{1}{2}\theta'^\top \Sigma^{-1}\theta' + \theta^\top \Sigma^{-1}\theta'}{\sigma}\right)$$

$$= 1 - \Phi\left(-\frac{\sigma}{2}\right) = \Phi\left(\frac{\sigma}{2}\right).$$

Similar calculations show that

$$\nu_{\theta'}(\mathcal{E}) = 1 - \Phi\left(\frac{\sigma}{2}\right).$$

Putting it all together gives us the lemma statement. $\qquad\square$

These two lemmas together show that, regardless of the trajectories, the total variation distance of the posterior distributions $P_\theta(\mu \mid \tau_h)$ and $P_{\theta'}(\mu \mid \tau_h)$ is *contracting*:

$$\text{TV}(P_\theta(\mu \mid \tau_h) \parallel P_{\theta'}(\mu \mid \tau_h)) = \text{TV}(\mathcal{N}(\theta_h, \Sigma_h) \parallel \mathcal{N}(\theta'_h, \Sigma_h))$$

$$= 2\Phi\left(\frac{\|\Sigma_h^{-1/2}(\theta_h - \theta'_h)\|}{2}\right) - 1$$

$$= 2\Phi\left(\frac{1}{2}\left(\sum_{a \in \mathcal{A}} \frac{\sigma^2 + n_{h,a}}{\sigma^2}(\theta \cdot \frac{\sigma^2}{\sigma^2 + n_{h,a}} - \theta' \cdot \frac{\sigma^2}{\sigma^2 + n_{h,a}})^2\right)^{1/2}\right) - 1$$

$$\leq 2\Phi\left(\frac{\|\theta - \theta'\|}{2}\right) - 1$$

$$= \text{TV}(\mathcal{N}(\theta, I_\mathcal{A}) \parallel \mathcal{N}(\theta'_h, I_\mathcal{A}))$$

where the inequality uses the fact that $n_{h,a}$ is non-negative. Putting it all together, we have

$$\text{TV}(P_{\theta, \text{TS}(\theta)}(\boldsymbol{\mu}, \tau_H) \parallel P_{\theta, \text{TS}(\theta')}(\boldsymbol{\mu}, \tau_H)) \leq \sum_{h=1}^{H} Q(\hat{\mu}_h \neq \hat{\mu}'_h \mid \boldsymbol{\mu} = \boldsymbol{\mu}', \tau_{h-1} = \tau'_{h-1})$$

$$\leq \sum_{h=1}^{H} \text{TV}(P_\theta(\mu \mid \tau_h) \parallel P_{\theta'}(\mu \mid \tau_h))$$

$$\leq \sum_{h=1}^{H} \text{TV}(\mathcal{N}(\theta, I_\mathcal{A}) \parallel \mathcal{N}(\theta'_h, I_\mathcal{A}))$$

$$= H \cdot \text{TV}(\mathcal{N}(\theta, I_\mathcal{A}) \parallel \mathcal{N}(\theta'_h, I_\mathcal{A}))$$

where the second inequality makes use of Lemma B.1.

## B.6 Proof of Proposition 3.4

To prove Proposition 3.4, we first decompose the total variation across steps $h$ via an analogue of the performance difference lemma.

**Lemma B.9.** *For any two algorithms* $\mathsf{alg}, \mathsf{alg}'$*, it holds that*

$$\mathrm{TV}(P_{\theta,\mathsf{alg}}(\boldsymbol{\mu}, \tau_H) \| P_{\theta,\mathsf{alg}'}(\boldsymbol{\mu}, \tau_H)) \leq \sum_{h=1}^{H} \mathop{\mathbb{E}}_{\tau_{h-1} \sim P_{\theta,\mathsf{alg}}} \left[ \mathrm{TV}(P_{\mathsf{alg}}(a_h \mid \tau_{h-1}) \| P_{\mathsf{alg}'}(a_h \mid \tau_{h-1})) \right],$$

*where on the right hand side, we consider the total variation distance between the conditional distribution of* $a_h$ *under* $\mathsf{alg}, \mathsf{alg}'$ *given* $\tau_{h-1}$ *and take expectation over* $\tau_{h-1}$ *under* $P_{\theta,\mathsf{alg}}$*.*

*Proof Sketch.* Using the variational characterization of total variation, we represent the total variation between the two measures as the supremum of differences in rewards under two Markov reward process induced by $P_{\theta,\mathsf{alg}}(\boldsymbol{\mu}, \tau_H)$ and $P_{\theta,\mathsf{alg}'}(\boldsymbol{\mu}, \tau_H)$. The decomposition then follows from a careful application of the performance difference lemma. See Appendix B.7 for the full proof. $\square$

In the second stage of the proof, we apply the following rather general lemma.

**Lemma B.10** (Fundamental De-conditioning Lemma)**.** *Let $Q$ and $Q'$ be two measures on a pair of random variables $(X, Y)$ such that the conditionals of $X$ given $Y$ coincide: $Q(X \mid Y) = Q'(X \mid Y)$ almost surely. Then,*

$$\mathbb{E}_{X \sim Q} \mathrm{TV}(Q(Y \mid X) \| Q'(Y \mid X)) \leq 2\mathrm{TV}(Q(Y) \| Q'(Y)).$$

*Proof of Lemma B.10.* We first review the essential properties of total variation used in the proof; we then turn to applying said properties to establish the lemma.

**Properties of Total Variation.** Let $P$ and $P'$ over jointly distributed random variables $(X, Y)$. First, if the marginals under $X$ coincide, then their total variation can be expressed as the expected total variation between the conditions $Y \mid X$; that is,

$$\text{if } P(X) = P'(X), \quad \text{then} \quad \mathrm{TV}(P(X, Y) \| P'(X, Y)) = \mathbb{E}_{X \sim P} \mathrm{TV}(P(Y \mid X) \| P'(Y \mid X)). \tag{B.4}$$

On the other hand, if their conditionals of $Y \mid X$ coincide, then we have the following simplification:

$$\text{if } P(Y \mid X) = P'(Y \mid X), \quad \text{then} \quad \mathrm{TV}(P(X, Y) \| P'(X, Y)) = \mathrm{TV}(P(X) \| P'(X)). \tag{B.5}$$

Equation (B.4) is established in Lemma B.2, and Equation (B.5) in Lemma B.3. We shall also use that the total variation satisfies the triangle inequality ($\mathrm{TV}(P \| P') \leq \mathrm{TV}(P \| P'') + \mathrm{TV}(P' \| P'')$) and the data-processing inequality ($\mathrm{TV}(P(X) \| P'(X)) \leq \mathrm{TV}(P(X, Y) \| P(X, Y))$), stated formally and proven in Lemma B.1.

**Main proof.** We introduce an interpolating law $Q_{\rightarrow}$ such that $Q_{\rightarrow}(X) = Q(X)$ and $Q_{\rightarrow}(Y \mid X) = Q'(Y \mid X)$. Then

$$\mathbb{E}_{X \sim Q} \mathrm{TV}(Q(Y \mid X) \| Q'(Y \mid X)) = \mathbb{E}_{X \sim Q} \mathrm{TV}(Q(Y \mid X) \| Q_{\rightarrow}(Y \mid X))$$

$$\overset{(i)}{=} \mathrm{TV}(Q(X, Y) \| Q_{\rightarrow}(X, Y))$$

$$\overset{(ii)}{\leq} \underbrace{\mathrm{TV}(Q_{\rightarrow}(X, Y) \| Q'(X, Y))}_{(a)} + \underbrace{\mathrm{TV}(Q(X, Y) \| Q'(X, Y))}_{(b)},$$

where equality $(i)$ uses (B.4) given the fact that $Q(X) = Q_{\rightarrow}(X)$, and where $(ii)$ applies the triangle inequality. This leaves us with two terms, $(a)$ and $(b)$. First we upper bound term $(a)$ by term $(b)$:

$$\mathrm{TV}(Q_{\rightarrow}(X, Y) \| Q'(X, Y)) = \mathrm{TV}(Q_{\rightarrow}(X) \| Q'(X))$$

$$= \mathrm{TV}(Q(X) \| Q'(X)) \leq \mathrm{TV}(Q(X, Y) \| Q'(X, Y)) =: (b).$$

Above, the first equality uses $Q_{\to}(Y \mid X) = Q'(Y \mid X)$ to invoke (B.5), the second the fact that $Q_{\to}(X) = Q(X)$, and the final equality applies the data-processing inequality described above. Hence,

$$\mathbb{E}_{X \sim Q} \mathrm{TV}(Q(Y \mid X) \parallel Q'(Y \mid X)) \leq 2\mathrm{TV}(Q(X,Y) \parallel Q'(X,Y)).$$

Finally, since $Q(X \mid Y) = Q'(X \mid Y)$, Equation (B.5) entails that $\mathrm{TV}(Q(X,Y) \parallel Q'(X,Y)) = \mathrm{TV}(Q(Y) \parallel Q'(Y))$. $\qquad\square$

Using our shorthand $P_H = P_{\theta,\mathsf{alg}(\theta)}(\boldsymbol{\mu}, \tau_H)$ and $P'_H = P_{\theta,\mathsf{alg}(\theta')}(\boldsymbol{\mu}, \tau_H)$, we can finish the proof of Proposition 3.4 as follows:

$$
\begin{aligned}
\mathrm{TV}(P_H \parallel P'_H) &\overset{(i)}{\leq} \sum_{h=1}^{H} \mathbb{E}_{\tau_{h-1} \sim P_{\theta,\mathsf{alg}(\theta)}} \left[ \mathrm{TV}(P_{\mathsf{alg}(\theta)}(a_h \mid \tau_{h-1}) \parallel P_{\mathsf{alg}(\theta')}(a_h \mid \tau_{h-1})) \right] \\
&\overset{(ii)}{\leq} n \sum_{h=1}^{H} \mathbb{E}_{\tau_{h-1} \sim P_{\theta,\mathsf{alg}(\theta)}} \left[ \mathrm{TV}(P_{\theta}(\boldsymbol{\mu} \mid \tau_{h-1}) \parallel P_{\theta'}(\boldsymbol{\mu} \mid \tau_{h-1})) \right] \\
&\overset{(iii)}{=} n \sum_{h=1}^{H} \mathbb{E}_{\tau_{h-1} \sim P_{\theta,\mathsf{alg}(\theta)}} \left[ \mathrm{TV}(P_{\theta,\mathsf{alg}(\theta)}(\boldsymbol{\mu} \mid \tau_{h-1}) \parallel P_{\theta',\mathsf{alg}(\theta)}(\boldsymbol{\mu} \mid \tau_{h-1})) \right] \\
&\overset{(iv)}{\leq} 2n \sum_{h=1}^{H} \mathrm{TV}(P_{\theta}(\boldsymbol{\mu}) \parallel P_{\theta'}(\boldsymbol{\mu})) = 2nH \cdot \mathrm{TV}(P_{\theta}(\boldsymbol{\mu}) \parallel P_{\theta'}(\boldsymbol{\mu})).
\end{aligned}
$$

Here, $(i)$ follows from Lemma B.9, $(ii)$ is the definition of $n$-Monte Carlo, $(iii)$ follows from the observation that the conditional distribution of $\boldsymbol{\mu}$ given $\tau_{h-1}$ does not depend on the algorithm that helped generate $\tau_{h-1}$, and $(iv)$ follows from Lemma B.10, where we have used the fact that $P_{\theta,\mathsf{alg}(\theta)}(\tau_{h-1} \mid \boldsymbol{\mu}) = P_{\theta',\mathsf{alg}(\theta)}(\tau_{h-1} \mid \boldsymbol{\mu})$, i.e. the conditional distribution of the trajectories does not depend on the prior when conditioning on the true mean $\boldsymbol{\mu}$.

## B.7 Proof of Lemma B.9

In the interest of generalizing our results to POMDPs (Appendix E), we will prove Lemma B.9 by establishing a nearly identical lemma which makes explicit the exact properties of the trajectories $\tau_h$ needed for Lemma B.9 to hold:

**Lemma B.11.** *Let $P, P'$ be two laws over abstract random variables $\boldsymbol{\mu}$, $\tau_1, \ldots, \tau_H$ and $a_1, \ldots, a_H$ such that the following properties hold:*

1. *$\tau_{h-1}$ is a deterministic function of $\tau_h$.*
2. *The conditional distributions of $\tau_h$ given $a_h, \tau_{h-1}$ and $\boldsymbol{\mu}$ are the same: $P'(\tau_h \mid a_h, \tau_{h-1}, \boldsymbol{\mu}) = P(\tau_h \mid a_h, \tau_{h-1}, \boldsymbol{\mu})$*
3. *Under both $P$ and $P'$, $a_h$ is independent of $\boldsymbol{\mu}$ given $\tau_{h-1}$.*

*Then, the following inequality holds:*

$$\mathrm{TV}(P(\boldsymbol{\mu}, \tau_H) \parallel P'(\boldsymbol{\mu}, \tau_H)) \leq \sum_{h=1}^{H} \mathbb{E}_{\tau_{h-1} \sim P} \left[ \mathrm{TV}(P(a_h \mid \tau_{h-1}) \parallel P'(a_h \mid \tau_{h-1})) \right].$$

Immediately, Lemma B.9 is obtained by taking $P = P_{\theta,\mathsf{alg}}$ and $P' = P_{\theta,\mathsf{alg}'}$:

1. Condition 1 is clear.
2. Condition 2 holds because the only part of $\tau_h$ not determined by $(a_h, \tau_{h-1}, \boldsymbol{\mu})$ is the reward $r_h$, and under both $P_{\theta,\mathsf{alg}}$ and $P'_{\theta,\mathsf{alg}}$, $r_h \sim \mathcal{D}(\boldsymbol{\mu})\big|_{a_h}$ is drawn from the same conditional distribution.
3. For Condition 3, first suppose there is no random seed $\boldsymbol{\xi}$. Then Condition 3 holds because the distribution of $a_h \sim f_h(\cdot \mid \tau_{h-1})$ is just a function of $\tau_{h-1}$. If there is a random seed, then letting

$P = P_{\theta,\mathrm{alg}}$, we have

$$
\begin{aligned}
P(a_h, \boldsymbol{\mu} \mid \tau_{h-1}) &= \mathbb{E}\left[\mathbb{E}\left[P(a_h, \boldsymbol{\mu} \mid \tau_{h-1}, \boldsymbol{\xi}) \mid \boldsymbol{\xi}\right] \mid \tau_{h-1}\right] \\
&= \mathbb{E}\left[\mathbb{E}\left[f_h(a_h \mid \tau_{h-1}, \boldsymbol{\xi}) P(\boldsymbol{\mu} \mid \tau_{h-1}, \boldsymbol{\xi}) \mid \boldsymbol{\xi}\right] \mid \tau_{h-1}\right] \\
&= \mathbb{E}\left[\mathbb{E}\left[f_h(a_h \mid \tau_{h-1}, \boldsymbol{\xi}) \mid \boldsymbol{\xi}\right] P(\boldsymbol{\mu} \mid \tau_{h-1}) \mid \tau_{h-1}\right] \\
&= P(a_h \mid \tau_{h-1}) P(\boldsymbol{\mu} \mid \tau_{h-1})
\end{aligned}
$$

where the second equality follows from the fact that $a_h \sim f_h(\cdot \mid \boldsymbol{\xi}, \tau_{h-1})$ and the third line follows from the fact that $\boldsymbol{\mu}$ is independent of $\boldsymbol{\xi}$ conditioned on $\tau_{h-1}$. The same argument holds symmetrically for $P' = P_{\theta,\mathrm{alg}'}$. Thus, Condition 3 holds regardless of whether or not there is a random seed.

*Proof of Lemma B.11.* For brevity, we define augmented trajectories containing the (unknown) mean parameter $\bar{\tau}_h = (\boldsymbol{\mu}, \tau_h)$ for $h = 1, 2, \ldots, H$. We further define $E$ and $E'$ to be the expectations under $P$ and $P'$, respectively. Fix any event $\mathcal{E}$ in the $\sigma$-algebra generated by $\bar{\tau}_H$; the total variation

$$
\mathrm{TV}(P(\bar{\tau}_H) \parallel P'(\bar{\tau}_H)) = \sup_{\mathcal{E}} P[\bar{\tau}_H \in \mathcal{E}] - P'[\bar{\tau}_H \in \mathcal{E}]
$$

can be expressed as the supremal difference over such events (Lemma B.1). We can then view this difference as the difference in rewards between two time-inhomogeneous Markov reward processes, with states $\bar{\tau}_h$ at step $h$ (note that the Markov property is trivially satisfied because $\tau_{h-1}$ is assumed to be deterministic function of $\tau_h$ by assumption), with identical rewards: at step $H$ the reward is $r_H(\bar{\tau}_H) = \mathbb{I}\{\bar{\tau}_H \in \mathcal{E}\}$, and steps $h < H$, the reward is zero. Let $V'_h(\cdot)$ denote the value function of step $h$ under the $P'$ reward process, the performance difference lemma [Kak03] then yields

$$
P[\bar{\tau}_H \in \mathcal{E}] - P'[\bar{\tau}_H \in \mathcal{E}] = \sum_{h=1}^{H} \mathbb{E}_{\bar{\tau}_{h-1} \sim P} \left[ E[V'_h(\bar{\tau}_h) \mid \tau_{h-1}] - E'[V'_h(\bar{\tau}_h) \mid \tau_{h-1}] \right]. \tag{B.6}
$$

Since the total reward collected is at most 1 and no less than 0, $V'_h(\cdot) \in [0,1]$. Hence, by the variational characterization of total variation (Lemma B.1),

$$
E[V'_h(\bar{\tau}_h) \mid \bar{\tau}_{h-1}] - E'[V'_h(\bar{\tau}_h) \mid \bar{\tau}_{h-1}] \leq \mathrm{TV}(P(\bar{\tau}_h \mid \bar{\tau}_{h-1}) \parallel P'(\bar{\tau}_h \mid \bar{\tau}_{h-1}))
$$

To conclude, it suffices to verify the inequality

$$
\mathrm{TV}(P(\bar{\tau}_h \mid \bar{\tau}_{h-1}) \parallel P'(\bar{\tau}_h \mid \bar{\tau}_{h-1})) \leq \mathrm{TV}(P(a_h \mid \tau_{h-1}) \parallel P'(a_h \mid \tau_{h-1})). \tag{B.7}
$$

To verify (B.7), let us fix a step $h$ and realization of $\bar{\tau}_{h-1}$, and set $Q = P(\cdot \mid \bar{\tau}_{h-1})$ and $Q' = P'(\cdot \mid \bar{\tau}_{h-1})$. Further, applying the data-processing inequality (Lemma B.1) followed by Lemma B.3 with $X = a_h$ and $Y = \bar{\tau}_h$ gives that $\mathrm{TV}(Q(Y) \parallel Q'(Y)) \leq \mathrm{TV}(Q(X,Y) \parallel Q'(X,Y)) = \mathrm{TV}(Q(X) \parallel Q'(X))$. Undoing the notational subsitutions, we have shown

$$
\mathrm{TV}(P(\bar{\tau}_h \mid \bar{\tau}_{h-1}) \parallel P'(\bar{\tau}_h \mid \bar{\tau}_{h-1})) \leq \mathrm{TV}(P(a_h \mid \bar{\tau}_{h-1}) \parallel P'(a_h \mid \bar{\tau}_{h-1})).
$$

Finally, we have

$$
\begin{aligned}
\mathbb{E}_{\bar{\tau}_{h-1} \sim P} \mathrm{TV}(P(a_h \mid \bar{\tau}_{h-1}) \parallel P'(a_h \mid \bar{\tau}_{h-1})) &= \mathbb{E}_{(\boldsymbol{\mu}, \tau_{h-1}) \sim P} \mathrm{TV}(P(a_h \mid \boldsymbol{\mu}, \tau_{h-1}) \parallel P'(a_h \mid \boldsymbol{\mu}, \tau_{h-1})) \\
&= \mathbb{E}_{(\boldsymbol{\mu}, \tau_{h-1}) \sim P} \mathrm{TV}(P(a_h \mid \tau_{h-1}) \parallel P'(a_h \mid \tau_{h-1})) \\
&= \mathbb{E}_{\tau_{h-1} \sim P} \mathrm{TV}(P(a_h \mid \tau_{h-1}) \parallel P'(a_h \mid \tau_{h-1})),
\end{aligned}
$$

where the first equality follows from the definition of $\bar{\tau}_{h-1} = (\boldsymbol{\mu}, \tau_{h-1})$, the second equality follows from the assumption that $a_h$ is independent of $\boldsymbol{\mu}$ given $\tau_{h-1}$, and the third equality follows from marginalization.

Concluding, we have shown that

$$
P[\bar{\tau}_H \in \mathcal{E}] - P'[\bar{\tau}_H \in \mathcal{E}] \leq \sum_{h=1}^{H} \mathbb{E}_{\tau_{h-1} \sim P} \mathrm{TV}(P(a_h \mid \tau_{h-1}) \parallel P(a_h \mid \tau_{h-1})) \tag{B.8}
$$

By the definition of total variation, $\mathrm{TV}(P(\bar{\tau}_H) \parallel P'(\bar{\tau}_H))$ is the supremum of the left-hand side over events $\mathcal{E}$ (Definition B.1), and we have defined $\bar{\tau}_H = (\boldsymbol{\mu}, \tau_H)$. Thus,

$$
\mathrm{TV}(P(\boldsymbol{\mu}, \tau_H) \parallel P'(\boldsymbol{\mu}, \tau_H)) \leq \sum_{h=1}^{H} \mathbb{E}_{\tau_{h-1} \sim P} \mathrm{TV}(P(a_h \mid \tau_{h-1}) \parallel P'(a_h \mid \tau_{h-1})). \qquad \square
$$

## B.8 Proof of Theorem B.1

We now turn to the proof of Theorem B.1. Recall the statement of the theorem.

**Theorem B.1.** Let $\mathsf{alg}(\cdot)$ be an $n$-Monte Carlo family of algorithms on horizon $H \in \mathbb{N}$, and let $\theta, \theta' \in \Theta$. Then, setting $\varepsilon = \mathrm{TV}(P_\theta \,\|\, P_{\theta'})$,

$$|R(\theta, \mathsf{alg}(\theta)) - R(\theta, \mathsf{alg}(\theta'))| \le 2nH^2 \varepsilon \cdot \Psi_\theta(2nH\varepsilon).$$

Proposition 3.4 states that

$$\mathrm{TV}(P_H \,\|\, P'_H) \le 2nH \cdot \mathrm{TV}(P_\theta \,\|\, P_{\theta'}),$$

where $P_H = P_{\theta, \mathsf{alg}(\theta)}(\boldsymbol{\mu}, \tau_H)$ and $P'_H = P_{\theta, \mathsf{alg}(\theta')}(\boldsymbol{\mu}, \tau_H)$. Thus, Theorem B.1 is a direct consequence of Proposition 3.4, monotonicty of $p \mapsto p \cdot \Psi_\theta(p)$ (Lemma B.5) and the following lemma.

**Lemma B.12.** *Given two algorithms $\mathsf{alg}$ and $\mathsf{alg}'$ and ground truth parameter $\theta$, let $\delta = \mathrm{TV}(P_H \,\|\, P'_H)$, where again $P_H = P_{\theta, \mathsf{alg}}(\boldsymbol{\mu}, \tau_H)$ and $P'_H = P_{\theta, \mathsf{alg}'}(\boldsymbol{\mu}, \tau_H)$ denote the marginals over induced trajectories and means. Then,*

$$|R(\theta, \mathsf{alg}) - R(\theta, \mathsf{alg}')| \le H \cdot \delta \cdot \Psi_\theta(\delta).$$

*Proof.* Recall that

$$R(\theta, \mathsf{alg}) - R(\theta, \mathsf{alg}') = E_{\theta, \mathsf{alg}} \left[ \sum_{h=1}^{H} \mu_{a_h} \right] - E_{\theta, \mathsf{alg}'} \left[ \sum_{h=1}^{H} \mu_{a_h} \right]$$

To bound this difference, we place all random variables on the same probability space. Adopt the shorthand $P_H = P_{\theta, \mathsf{alg}}(\boldsymbol{\mu}, \tau_H)$ and $P'_H = P_{\theta, \mathsf{alg}'}(\boldsymbol{\mu}, \tau_H)$. Since $P_H(\boldsymbol{\mu}) = P'_H(\boldsymbol{\mu})$, Lemma B.4 ensures the existence of a coupling $Q(\boldsymbol{\mu}, \tau_H, \tau'_H)$ such that

$$Q(\boldsymbol{\mu}, \tau_H) = P_H, \quad Q(\boldsymbol{\mu}, \tau'_H) = P'_H, \quad Q[\tau_H \ne \tau'_H] = \mathrm{TV}(P_H \,\|\, P'_H) = \delta. \tag{B.9}$$

Letting $E_Q$ denote expectations under this coupling, and $a'_h$ the actions within $\tau'_H$, we then have

$$R(\theta, \mathsf{alg}) - R(\theta, \mathsf{alg}') = E_Q \left[ \sum_{h=1}^{H} \mu_{a_h} - \mu_{a'_h} \right]$$

$$\le E_Q \left[ \sum_{h=1}^{H} \mathrm{diam}(\boldsymbol{\mu}) \mathbb{I}\{a_h \ne a'_h\} \right]$$

$$\le H E_Q[\mathrm{diam}(\boldsymbol{\mu}) \mathbb{I}\{\tau_H \ne \tau'_H\}].$$

Note that $\mathbb{I}\{\tau_H \ne \tau'_H\} \in [0, 1]$ and, by construction of the coupling $Q$, $E_Q[\mathbb{I}\{\tau_H \ne \tau'_H\}] = \delta$. Hence, by the definition of the tail expectation, $E_Q[\mathrm{diam}(\boldsymbol{\mu}) \mathbb{I}\{\tau_H \ne \tau'_H\}] \le \delta \Psi_\theta(\delta)$. The bound follows. $\qquad\square$

# C Specification of Monte Carlo Algorithms

In this appendix, we elaborate upon various examples of algorithms satisfying the $n$-Monte Carlo property.

$k$**-Shot Thompson Sampling.** The first is a natural generalization of Thompson Sampling, where one draws not one but $k \in \mathbb{N}$ mean vectors $\tilde{\boldsymbol{\mu}}^{(i)}$ from the posterior at each step $h$, and selects the action for which one of the $k$ draws attains the highest observed realization: $a_h \in \arg\max_a \max_i \tilde{\mu}_a^{(i)}$. See Algorithm 1.

---

**Algorithm 1** $k$-Shot Thompson Sampling ($k$-TS($\theta$))

---

1: **Input:** Prior $\theta$, sample size $k \in \mathbb{N}$
2: **for** $h = 1, \ldots, H$ **do**
       // action selection at step $h$
3:     Sample $\tilde{\boldsymbol{\mu}}^{(1)}, \ldots, \tilde{\boldsymbol{\mu}}^{(k)}$ independently from the posterior $P_\theta[\cdot \mid \tau_{h-1}]$
4:     Select action $a_h \in \arg\max_a \max\{\tilde{\mu}_a^{(1)}, \ldots, \tilde{\mu}_a^{(k)}\}$.

---

We show that $k$-TS($\theta$) is $k$-Monte Carlo.

**Lemma C.1.** *For every $k \geq 1$, $k$-TS($\cdot$) is $k$-Monte Carlo. In particular, TS($\cdot$) is 1-Monte Carlo.*

**Generalized Posterior Sampling.** Lemma C.1 follows from an analysis of a more general recipe for $n$-Monte Carlo algorithms. Algorithm 2 describes describe a family of posterior sampling algorithms $(k, f_{1:H})$-PosteriorSample($\theta$), parameterized by prior $\theta$ and determined by a sample size $k \in \mathbb{N}$ and functions $f_1, \ldots, f_H : \mathbb{R}^{\mathcal{A} \times n}$ from $\mathbb{R}^{\mathcal{A} \times k}$ to probability distribution $\Delta^{\mathcal{A}}$ over actions. At each step $h$, $k$ means $\tilde{\boldsymbol{\mu}}^{(1)}, \ldots, \tilde{\boldsymbol{\mu}}^{(k)}$ are sampled from the posterior, and an action $a_h$ is drawn from the probability distribution $f_h(\cdot \mid \tilde{\boldsymbol{\mu}}^{(i)}, \ldots, \tilde{\boldsymbol{\mu}}^{(k)})$ induced by evaluating $f_h$ on the sampled means.

---

**Algorithm 2** $(k, f_{1:H})$-Posterior Sampling $((k, f_{1:H})$-PosteriorSample($\theta$))

---

1: **Input:** Prior $\theta$, sample size $k \in \mathbb{N}$, functions $f_1, \ldots, f_H : \mathbb{R}^{\mathcal{A} \times k} \to \Delta^{\mathcal{A}}$.
2: **for** $h = 1, \ldots, H$ **do**
       // action selection at step $h$
3:     Sample $\tilde{\boldsymbol{\mu}}^{(1)}, \ldots, \tilde{\boldsymbol{\mu}}^{(k)}$ independently from the posterior $P_\theta[\cdot \mid \tau_{h-1}]$
4:     Select action $a_h \sim f_h(\cdot \mid \tilde{\boldsymbol{\mu}}^{(1)}, \ldots, \tilde{\boldsymbol{\mu}}^{(k)})$.

---

Note that $k$-TS($\theta$) corresponds to the special case where $f_h = f$ is constant across $h$, and places a dirac mass on the action for which $\max_i \tilde{\mu}_a^{(i)}$ is largest (with a suitable tie-breaking rule). In particular, TS($\theta$) is a special case of $(k, f_{1:H})$-PosteriorSample($\theta$) with $k = 1$. Other algorithms in the family include the rule which selects the arm with largest sample average/sum $a_h \in \sum_i \tilde{\mu}_a^{(i)}$, or a policy which selects $a_h$ according to a softmax distribution on the sums $\sum_i \tilde{\mu}_a^{(i)}$.

The following lemma shows that, regardless of the functions $f_1, \ldots, f_H$, $(k, f_{1:H})$-PosteriorSample($\theta$) is $k$-Monte Carlo. Note that Lemma C.1 follows as a special case.

**Lemma C.2.** *For any $k \in \mathbb{N}$ and $f_1, \ldots, f_H : \mathbb{R}^{\mathcal{A} \times k} \to \Delta^{\mathcal{A}}$, the family of Bayesian algorithms given by $(k, f_{1:H})$-PosteriorSample($\cdot$) is $k$-Monte Carlo.*

The proof of Lemma C.2 is quite intuitive, and is given in Appendix C.1.

**Receding Horizon Control.** Some Bayesian bandit algorithms do not exactly satisfy the conditions of Lemma C.2 but are nonetheless $n$-Monte Carlo. As an example, we consider a sampling-based implementation of two-stage receding horizon control, 2-RHC($\theta$), detailed Algorithm 3. 2-RHC($\theta$) selects an action $a$ which maximizes $V_a$, which can be thought of as a discounted two-step value function, balancing (a) selection of actions with large posterior means and (b) selection of actions that are sufficiently informative such that the best action for "look-ahead" means sampled from the next stage yield large reward. This balance is controlled by a discount parameter $\alpha \in [0, 1]$.

---

**Algorithm 3** Two-Step Receding Horizon Control (2-RHC($\theta$))

---

1: **Input:** Prior $\theta$, discount parameter $\alpha \in [0, 1]$, sample sizes $k_1, k_2 \in \mathbb{N}$.
2: **for** $h = 1, \ldots, H$ **do**
   `// action selection at step` $h$
3:     **for** actions $a \in \mathcal{A}$ and $i = 1, \ldots, k_1$ **do**
4:         Sample mean $\tilde{\boldsymbol{\mu}}^{(a,i)} \sim P_\theta[\cdot \mid \tau_{h-1}]$
5:         Sample reward vector $\tilde{\mathbf{r}}^{(a,i)} \sim \mathcal{D}(\tilde{\boldsymbol{\mu}}^{(a,i)})$
6:         **for** $j = 1, 2, \ldots, k_2$ **do**
7:             Sample "look-ahead" means $\hat{\boldsymbol{\mu}}^{(a,i,j)} \sim P_\theta[\in \cdot \mid \tau_{h-1} \text{ and } \{r_{h+1,a} = \tilde{r}_a^{(a,i)}\}]$
8:     **Select** action $a_h \in \arg\max_a V_{a,h}$, where

$$V_a := \sum_{i=1}^{k_1} \left( (1-\alpha)\tilde{\mu}_a^{(a,i)} + \alpha \left( \max_{a'} \frac{1}{k_2} \sum_{j=1}^{k_2} \hat{\mu}_{a'}^{(a,i,j)} \right) \right). \tag{C.1}$$

---

At the one extreme $\alpha = 0$, 2-RHC($\theta$) has no look-ahead, and is a special case of $(k, f_{1:H})$-Posterior Sampling with $k = |\mathcal{A}|k_1$. At the other extreme $\alpha = 1$, 2-RHC($\theta$) disregards means sampled from the posterior, and only evaluates actions $a$ by how informative they are about the look-ahead means. This latter case, $\alpha = 1$, in fact gives a Monte Carlo approximation of the classical knowledge gradient algorithm [RPF12]. The following lemma verifies the Monte Carlo property for all choices of $\alpha$.

**Lemma C.3.** 2-RHC($\cdot$) *with budgets* $k_1, k_2$ *and finite action set is* $n$-*Monte Carlo for* $n = |\mathcal{A}| \cdot k_1 \cdot (2k_2 + 3)$, *regardless of discount* $\alpha \in [0, 1]$.

The proof of the above lemma is provided in Appendix C.2.

## C.1 Proof of Lemma C.2.

Lemma B.1 guarantees the existence of a *maximal coupling* $Q(\boldsymbol{\mu}, \boldsymbol{\mu}')$ between $P := P_\theta(\boldsymbol{\mu} \mid \tau_{h-1})$ and $P' := P_{\theta'}(\boldsymbol{\mu} \mid \tau_{h-1})$; that is, a joint law over $(\boldsymbol{\mu}, \boldsymbol{\mu}')$ with marginals $\boldsymbol{\mu} \sim P$ and $\boldsymbol{\mu}' \sim P'$, and for which $\mathbb{P}_{(\boldsymbol{\mu}, \boldsymbol{\mu}') \sim Q}[\boldsymbol{\mu} \neq \boldsymbol{\mu}'] = \mathrm{TV}(P \parallel P')$. Using $Q$, we construct a coupling $\bar{Q}$ of $P_{\mathsf{alg}(\theta)}(a_h \mid \tau_{h-1})$ and $P_{\mathsf{alg}(\theta')}(a_h \mid \tau_{h-1})$:

- Draw $(\boldsymbol{\mu}_1, \boldsymbol{\mu}_1'), \ldots, (\boldsymbol{\mu}_n, \boldsymbol{\mu}_n')$ i.i.d. from $Q$.
- If $\boldsymbol{\mu}_1 = \boldsymbol{\mu}_1', \ldots, \boldsymbol{\mu}_n = \boldsymbol{\mu}_n'$, then draw $a_h \sim f_h(\cdot \mid \boldsymbol{\mu}_1, \ldots, \boldsymbol{\mu}_n)$ and let $a_h' = a_h$.
- Otherwise, let $a_h \sim f_h(\cdot \mid \boldsymbol{\mu}_1, \ldots, \boldsymbol{\mu}_n)$ and $a_h' \sim f_h(\cdot \mid \boldsymbol{\mu}_1', \ldots, \boldsymbol{\mu}_n')$ independently.

It is easily verified that this defines a valid coupling of $P_{\mathsf{alg}(\theta)}(a_h \mid \tau_{h-1})$ and $P_{\mathsf{alg}(\theta')}(a_h \mid \tau_{h-1})$. Hence, Lemma B.1 ensures that

$$\mathrm{TV}(P_{\mathsf{alg}(\theta')}(a_h \mid \tau_{h-1}) \parallel P_{\mathsf{alg}(\theta')}(a_h \mid \tau_{h-1})) \leq \bar{Q}(a_h \neq a_h').$$

Continuing, we conclude

$$\bar{Q}(a_h \neq a_h') \leq \mathbb{P}(\exists i : \boldsymbol{\mu}_i \neq \boldsymbol{\mu}_i') \leq \sum_{i=1}^n Q(\boldsymbol{\mu}_i \neq \boldsymbol{\mu}_i') = n \cdot \mathrm{TV}(P_\theta(\boldsymbol{\mu} \mid \tau_{h-1}) \parallel P_{\theta'}(\boldsymbol{\mu} \mid \tau_{h-1})).$$

Thus, $\mathsf{alg}(\theta)$ is $n$-Monte Carlo. $\qquad\square$

## C.2 Monte Carlo Property of 2-RHC (Lemma C.3)

Let us briefly recall the specified of 2-RHC (Algorithm 3); we include an explicit dependence on horizon to avoid confusion. At each step $h$, we

- Sample means $\tilde{\boldsymbol{\mu}}_h^{(a,i)} \sim P_\theta[\cdot \mid \tau_{h-1}]$ for each $a \in \mathcal{A}$ and $i \in [k_1]$
- For each such $(a, i)$, we samplea reward vector $\tilde{\mathbf{r}}_h^{(a,i)} \sim \mathcal{D}(\tilde{\boldsymbol{\mu}}^{(a,i)})$

- For each $(a, i)$ and $j \in [k_2]$, we sample "look-ahead" means $\hat{\boldsymbol{\mu}}_h^{(a,i,j)} \sim P_\theta[\in \cdot \mid \tau_{h-1}$ and $\{r_{h+1,a} = \tilde{r}_a^{(a,i)}\}]$
- The action selected is a deterministic function of the vector $(\tilde{\boldsymbol{\mu}}_h^{(a,i)}, \hat{\boldsymbol{\mu}}_h^{(a,i,j)})_{a \in \mathcal{A}, i \in [k_1], j \in [k_2]}$.

Continuing, fix a step $h \in \mathbb{N}$, and introduce the shorthand

$$\mathbf{Z}_{h,a,i} := (\tilde{\boldsymbol{\mu}}^{(a,i)}, (\hat{\boldsymbol{\mu}}^{(a,i,j)})_{j \in [k_2]}), \quad \text{and} \quad \mathbf{Z}_h = (\mathbf{Z}_{h,a,i})_{a \in \mathcal{A}, i \in [k_1]}.$$

The 2-RHC decision rule is then a deterministic function of $\mathbf{Z}_h$, so it suffices to bound

$$\mathrm{TV}(P_{\mathsf{alg}(\theta)}(\mathbf{Z}_h \mid \tau_{h-1}) \parallel P_{\mathsf{alg}(\theta')}(\mathbf{Z}_h \mid \tau_{h-1})).$$

Moreover, given $\tau_{h-1}$, $\mathbf{Z}_{h,a,i}$ are independent across $a \in \mathcal{A}$ and $i \in [k_1]$. Thus, by the tensorization property of total variation (Lemma B.1),

$$\mathrm{TV}(P_{\mathsf{alg}(\theta)}(\mathbf{Z}_h \mid \tau_{h-1}) \parallel P_{\mathsf{alg}(\theta')}(\mathbf{Z} \mid \tau_{h-1}))$$
$$\leq \sum_{a \in \mathcal{A}} \sum_{i \in [k_1]} \mathrm{TV}(P_{\mathsf{alg}(\theta)}(\mathbf{Z}_{h,a,i} \mid \tau_{h-1}) \parallel P_{\mathsf{alg}(\theta')}(\mathbf{Z}_{h,a,i} \mid \tau_{h-1})). \tag{C.2}$$

We now decouple the summands above by appealing to the following property:

**Claim 1.** *For the given $h \in [H]$, fix $a \in \mathcal{A}$ and $i \in [k_1]$, and introduce the short hand*

$$X_0 = \tilde{\boldsymbol{\mu}}_h^{(a,i)}, \quad X_j = \hat{\boldsymbol{\mu}}_h^{(a,i,j)}, \ j \in [k_1], \quad \text{and} \quad Y = \tilde{r}_{h,a}^{(a,i)},$$

*where all random variables above are those simulated by* 2-RHC *at step $h$. Then,*

(a) *Under $P_{\mathsf{alg}(\theta)}(\cdot \mid \tau_{h-1}, Y)$ (and similarly under $\theta'$), $(X_0, X_1, \ldots, X_{k_2})$ are independent and identically distributed.*

(b) *The distributions $P_{\mathsf{alg}(\theta)}(Y \mid \tau_{h-1}, X_0)$ and $P_{\mathsf{alg}(\theta')}(Y \mid \tau_{h-1}, X_0)$ are identical.*

*Proof.* Let us start with point (b). Recall that $\tilde{r}_a^{(a,i)}$ denotes the action-$a$ entry of $\tilde{\mathbf{r}}^{(a,i)}$, which is drawn from the distribution $\mathcal{D}(\tilde{\boldsymbol{\mu}}^{(a,i)})$, regardless of the parameter $\theta$. Hence, given $X_0 = \tilde{\boldsymbol{\mu}}^{(a,i)}$, the distribution of $Y$ is identical under $\mathsf{alg}(\theta)$ and $\mathsf{alg}(\theta')$.

Let us turn to part (a). We focus on $P_\theta$, as the argument for $P_{\theta'}$ is identical. We notice that under a given $\theta$,

$$P_\theta(\tilde{\boldsymbol{\mu}}_h^{(a,i)} \mid \tau_{h-1}, \tilde{r}_{h,a}^{(a,i)} = r) = P_\theta(\boldsymbol{\mu} \mid \tau_{h-1}, r_{h,a} = r);$$

in words, the posterior of the simulated mean $\tilde{\boldsymbol{\mu}}^{(a,i)}$ given simulated reward $\tilde{r}_a^{(a,i)}$ is equal to the posterior of the *true* mean given that the reward $r_{h,a}$. This is because

(a) $P_\theta(\tilde{\boldsymbol{\mu}}_h^{(a,i)} \mid \tau_{h-1}) = P_\theta(\boldsymbol{\mu} \mid \tau_{h-1})$ (that is, $\tilde{\boldsymbol{\mu}}^{(a,i)}$ is drawn from the true posterior given $\tau_{h-1}$)

(b) The condition distribution of $\tilde{r}_a^{(a,i)} \mid \tilde{\boldsymbol{\mu}}^{(a,i)}, \tau_{h-1}$ is equal to the condition distribution of $r_{h,a} \mid \boldsymbol{\mu}, \tau_{h-1}$. Note that conditioning on the trajectory is immaterial to the draw of these reward, and both are given by the restriction of the reward distribution $\mathcal{D}$ to entry $a \in \mathcal{A}$.

Moreover definition of the 2-RHC procedure,

$$P_\theta(\hat{\boldsymbol{\mu}}^{(a,i,j)} \mid \tau_{h-1}, \tilde{r}_{h,a}^{(a,i)} = r) := P_\theta(\boldsymbol{\mu} \mid \tau_{h-1}, r_{h,a} = r)$$

for all $i \in [k_2]$ as well. Hence, $P_\theta(X_i \mid Y)$ are identically distributed.

To conclude part (a), we must verify independence. This holds since $X_1, \ldots, X_{k_2}$ are i.i.d. draws from $P_\theta(\cdot \mid \tau_{h-1}, (a_h, r_h) = (a, Y))$, regardless of the realization of $X_0$. Hence, $X_1, \ldots, X_{k_2}$ are conditionally independent of each other, and of $X_0$, given $Y, \tau_{h-1}$. $\qquad \square$

The next lemma lets us put the two properties in Claim 1 to use:

**Lemma C.4.** *Let $P$ and $P'$ be two probability distributions over random variables $(Y, X_0, X_1, \ldots, X_k)$ such that*

*(a)* $X_{0:k} \mid Y$ *are independent and identically distributed under both $P$ and $P'$*

*(b)* *The conditionals $P(Y \mid X_0) = P'(Y \mid X_0)$ are the same.*

*Then,* $\mathrm{TV}(P(X_{0:k}) \parallel P'(X_{0:k})) \leq (2k+3)\mathrm{TV}(P(X_0) \parallel P'(X_0))$.

Before we prove Lemma C.4, we show how it implies Lemma C.3. Fix indices $a \in \mathcal{A}$ and $i \in [k_1]$. Recall the random variables $(X_{0:k_2}, Y)$ defined in Claim 1, and note that $\mathbf{Z}_{a,i}$ is precisely given by $X_{0:k_2}$; that is,

$$\mathrm{TV}(P_{\mathsf{alg}(\theta)}(\mathbf{Z}_{a,i} \mid \tau_{h-1}) \parallel P_{\mathsf{alg}(\theta')}(\mathbf{Z}_{a,i} \mid \tau_{h-1})) = \mathrm{TV}(P_{\mathsf{alg}(\theta)}(X_{0:k_2} \mid \tau_{h-1}) \parallel P_{\mathsf{alg}(\theta')}(X_{0:k_2} \mid \tau_{h-1})).$$

Claim 1 further ensures that, under the history-conditioned laws $P \leftarrow P_\theta(\cdot \mid \tau_{h-1})$ and $P' \leftarrow P_\theta(\cdot \mid \tau_{h-1})$, $X_{0:k_2}$ satisfy the conditions of Lemma C.4. This implies that

$$\mathrm{TV}(P_{\mathsf{alg}(\theta)}(X_{0:k_2} \mid \tau_{h-1}) \parallel P_{\mathsf{alg}(\theta')}(X_{0:k_2} \mid \tau_{h-1}))$$
$$\leq (2k_2+3)\mathrm{TV}(P_{\mathsf{alg}(\theta)}(X_0 \mid \tau_{h-1}) \parallel P_{\mathsf{alg}(\theta')}(X_0 \mid \tau_{h-1})).$$

Finally, by construction, $X_0 = \tilde{\boldsymbol{\mu}}^{(a,i)}$ is drawn from $P_\theta(\boldsymbol{\mu} \mid \tau_{h-1})$. Hence, we conclude

$$\mathrm{TV}(P_{\mathsf{alg}(\theta)}(\mathbf{Z}_{a,i} \mid \tau_{h-1}) \parallel P_{\mathsf{alg}(\theta')}(\mathbf{Z}_{a,i} \mid \tau_{h-1})) \leq (2k_2+3)\mathrm{TV}(P_\theta(\boldsymbol{\mu} \mid \tau_{h-1}) \parallel P_{\theta'}(\boldsymbol{\mu} \mid \tau_{h-1})).$$

And by Equation (C.2),

$$\mathrm{TV}(P_{\mathsf{alg}(\theta)}(\mathbf{Z} \mid \tau_{h-1}) \parallel P_{\mathsf{alg}(\theta')}(\mathbf{Z} \mid \tau_{h-1})) \leq |\mathcal{A}| k_1 (2k_2+3)\mathrm{TV}(P_\theta(\boldsymbol{\mu} \mid \tau_{h-1}) \parallel P_{\theta'}(\boldsymbol{\mu} \mid \tau_{h-1})),$$

yielding the $n$-Monte Carlo property for $n = |\mathcal{A}| k_1 (2k_2+3)$.

*Proof of Lemma C.4.* Introduce the measure $\tilde{P}$ under which $\tilde{P}(Y) = P(Y)$ and $\tilde{P}(X_{0:k} \mid Y) = P'(X_{0:k} \mid Y)$. By the data processing and triangle inequalities

$$\mathrm{TV}(P(X_{0:k}) \parallel P'(X_{0:k})) \leq \mathrm{TV}(P(X_{0:k}, Y) \parallel P'(X_{0:k}, Y))$$
$$\leq \mathrm{TV}(P(X_{0:k}, Y) \parallel \tilde{P}(X_{0:k}, Y)) + \mathrm{TV}(P'(X_{0:k}, Y) \parallel \tilde{P}(X_{0:k}, Y)).$$

Using Lemma B.3 and the fact that $\tilde{P}(X_{0:k} \mid Y) = P'(X_{0:k} \mid Y)$, followed by the fact $\tilde{P}(Y) = P(Y)$ and the data-processing inequality again, we have

$$\mathrm{TV}(P'(X_{0:k}, Y) \parallel \tilde{P}(X_{0:k}, Y)) = \mathrm{TV}(P'(Y) \parallel \tilde{P}(Y))$$
$$= \mathrm{TV}(P'(Y) \parallel P(Y)) \leq \mathrm{TV}(P'(X_0, Y) \parallel P(X_0, Y)). \quad \text{(C.3)}$$

On the other hand, using Lemma B.2 and the fact that $\tilde{P}(Y) = P(Y)$.

$$\mathrm{TV}(P(X_{0:k}, Y) \parallel \tilde{P}(X_{0:k}, Y)) = \mathbb{E}_{Y \sim P(Y)} \mathrm{TV}(P(X_{0:k} \mid Y) \parallel \tilde{P}(X_{0:k} \mid Y)). \quad \text{(C.4)}$$

Now, observe that $\tilde{P}(X_{0:k} \mid Y) = P'(X_{0:k} \mid Y)$, and under both $P$ and $P'$, $X_{0:k} \mid Y$ are independent and identically distributed. Hence, from the decoupling property (Lemma B.1), Equation (C.4) yields

$$\mathrm{TV}(P(X_{0:k}, Y) \parallel \tilde{P}(X_{0:k}, Y)) = \sum_{i=0}^{k} \mathbb{E}_{Y \sim P(Y)} \mathrm{TV}(P(X_i \mid Y) \parallel \tilde{P}(X_i \mid Y))$$
$$= (k+1) \mathbb{E}_{Y \sim P(Y)} \mathrm{TV}(P(X_0 \mid Y) \parallel \tilde{P}(X_0 \mid Y)).$$

Reversing the decoupling property,

$$\mathrm{TV}(P(X_{0:k}, Y) \parallel \tilde{P}(X_{0:k}, Y)) = (k+1)\mathrm{TV}(P(X_0, Y) \parallel \tilde{P}(X_0, Y)).$$

Now, we invoke the triangle inequality once more to get

$$\mathrm{TV}(P(X_{0:k}, Y) \parallel \tilde{P}(X_{0:k}, Y))$$
$$\leq (k+1)\mathrm{TV}(P(X_0, Y) \parallel P'(X_0, Y)) + (k+1)\mathrm{TV}(\tilde{P}(X_0, Y) \parallel P'(X_0, Y)).$$

Invoking Equation (C.3), we have $\mathrm{TV}(\tilde{P}(X_0, Y) \parallel P'(X_0, Y)) \leq \mathrm{TV}(P(X_0, Y) \parallel P'(X_0, Y))$, yielding a final bound of

$$\mathrm{TV}(P(X_{0:k}) \parallel P'(X_{0:k})) \leq \mathrm{TV}(P(X_{0:k}, Y) \parallel P'(X_{0:k}, Y))$$
$$\leq \mathrm{TV}(P(X_{0:k}, Y) \parallel \tilde{P}(X_{0:k}, Y)) + \mathrm{TV}(P'(X_{0:k}, Y) \parallel \tilde{P}(X_{0:k}, Y))$$
$$\leq (2k+3)\mathrm{TV}(P(X_0, Y) \parallel P'(X_0, Y))$$

Finally, since $P(Y \mid X_0)$ and $P'(Y \mid X_0)$ coincide, Lemma B.3 yields that

$$\mathrm{TV}(P(X_0, Y) \parallel P'(X_0, Y)) = \mathrm{TV}(P(X_0) \parallel P'(X_0)).$$

The bound follows. $\qquad\square$

# D Lower Bounds

**Theorem D.1.** *There is a universal constant $c_0 \geq 1$ such that the following holds. Fix any $k \in \mathbb{N}$, and a tolerance $\eta \in (0, 1/4)$. Then for all horizons $H \geq \frac{c_0}{\eta}$ and errors $\epsilon \leq \frac{\eta}{c_0 k H}$, there are two priors $\theta, \theta'$ over bandit instances with $|\mathcal{A}| = H\lceil \frac{c_0}{\eta} \rceil$ arms such that (a) $\mathrm{TV}(P_\theta \parallel P_{\theta'}) = \epsilon$, (b) $\boldsymbol{\mu} \in [0,1]^{\mathcal{A}}$ with probability one under both $P_\theta$ and $P_{\theta'}$, and (c) the difference in rewards collected by $k$-$\mathrm{TS}(\theta)$ and $k$-$\mathrm{TS}(\theta')$ is at least*

$$R(\theta, k\text{-}\mathrm{TS}(\theta)) - R(\theta, k\text{-}\mathrm{TS}(\theta')) \geq \left(\frac{1}{2} - \eta\right) k\epsilon H^2.$$

Note that, by rescaling, an $\left(\frac{1}{2} - \eta\right) k\epsilon H^2 B$ bound holds against $B$-bounded priors for any $B > 0$. The proof of Theorem D.1 is given in Appendix D.1. A proof sketch is given below. The construction is somewhat involved, and relies on a carefully contrived prior and deterministic rewards.

In Appendix D.2, we also provide a simpler construction that provides a sharp converse to Proposition 3.4 and removes the deterministic rewards condition of Theorem D.1 to allow for Bernoulli rewards.

*Proof Sketch of Theorem D.1.* We construct a (rather contrived) prior $\theta$ over means $\boldsymbol{\mu}$ with $N + 1$ arms; the rewards are deterministically equal to the prior mean. Under $P_\theta$, a single arm $\bar{a} \in [N]$ is chosen uniformly at random to have reward close to 1, and the rest have zero reward. The $N + 1$-st arm has an $\epsilon$ probability of having reward exactly *equal to* 1, and thus an $\epsilon$ probability of being the best. Otherwise, the $N + 1$-st arm has low reward, but the value of its reward encodes the location of the optimal arm $\bar{a} \in [N]$.

At each stage $h$, we show that $k$-$\mathrm{TS}(\theta)$ has approximately $k\epsilon$ probability of selecting arm $N + 1$ under the hunch that it may be the best, only to find that it reveals the location of the best arm. After this revelation, the algorithm knows the best arm with certainty, and thus collects reward close to 1 for the remainder of the episode. Hence, at each step $h$, there is a close to $k\epsilon$ chance of accruing reward close to $H - h$ for the remaining steps. For small enough $\epsilon$, we show this yields cumulative reward at least about $k\epsilon \binom{H-1}{2} \approx k\epsilon H^2/2$.

We then construct an alternative prior $\theta'$ which places zero probability that arm $N + 1$ has the greatest reward, but otherwise coincides with $P_\theta$. Thus, $\mathrm{TV}(P_\theta \parallel P_{\theta'}) = \epsilon$, but $k$-$\mathrm{TS}(\theta')$ fails to sample arm $N + 1$, and misses out on the additional information about which arm is optimal. Without this information, $k$-$\mathrm{TS}(\theta')$ makes random guesses at the best arm $\bar{a} \in [N]$, and accumulates close to zero reward (in expectation) provided $N$ is sufficiently large. Naively, this argument would require $N$ to grow with $1/k\epsilon$. By coupling the behavior of $k$-$\mathrm{TS}(\theta)$ and $k$-$\mathrm{TS}(\theta')$, we only require $N$ to scale with $1/H$. $\qquad \square$

## D.1 Proof of Theorem D.1

By rescaling, we assume without loss of generality that $B = 1$. Fix a parameter $\delta < 2^{-5}$ to be chosen later, and consider $N + 1$ arms. We assume that the rewards are *noiseless*; that is, $r_{h,a} = \mu_a$ with probability 1.

Let $\theta$ denote the prior where an arm $\bar{a} \in [N]$ and a binary random variable $\bar{b} \in \{0, 1\}$ are drawn such that

$$\bar{a} \overset{\mathrm{unif}}{\sim} [N], \quad \bar{b} \sim \mathrm{Bernoulli}(\epsilon), \quad \hat{a} \perp b.$$

Given $(\bar{a}, \bar{b})$, the mean $\boldsymbol{\mu} = \boldsymbol{\mu}(\bar{a}, \bar{b})$, where we define

$$\boldsymbol{\mu}(\bar{a}, \bar{b})\big|_a = \begin{cases} 1 - \delta & a = \bar{a} \\ \delta & a \in [N] \setminus \{\bar{a}\} \\ \delta \frac{\bar{a}}{2N} & a = N + 1 \text{ and } b = 0 \\ 1 & a = N + 1 \text{ and } b = 1. \end{cases}$$

We make the following observations:

- $\mu_{N+1}$ uniquely determines the best arm. Indeed, if $\mu_{N+1} \leq \delta/2$, then the best arm is $\bar{a}$, which can be recovered by setting $\bar{a} = \frac{2N}{\delta} \cdot \mu_{N+1}$. Otherwise, $\mu_{N+1} = 1$, and it is the best arm. Hence, given any trajectory containing $a_h = N + 1$, there is a unique best arm under the posterior.
- Given any trajectory $\tau_h$ which *does not* contain $a_h = N + 1$, no information is communicated about the Bernoulli variable $b$. Moreover, if $\tau_h$ also does not include $a_h = \bar{a}$, then $\bar{a}$ is uniform on $[N] \setminus \{a_1, \ldots, a_h\}$.

Using these facts, we derive an implementation for $k$-TS$(\theta)$ in Algorithm 4.

**Claim 2.** *The pseudocode given by Algorithm 4 is a valid implementation of $k$-TS$(\theta)$.*

*Proof.* Consider any trajectory $\tau_{h-1}$. If $\tau_{h-1}$ contains any action $a_{h'} = N + 1$, $h' < h$, then as noted above, the best action is given by the selection in Line 4. Otherwise, $\tau_{h-1}$ only contains actions $a \in [N]$. There are now two cases:

- *Case 1.* $\tau_{h-1}$ contains a reward $r_{h'} = 1 - \delta$. We continue to let $h'$ denote this time step. In this case, $a_{h'}$ must be the index $\bar{a}$, and $\bar{a}$ is yields maximal reward over all actions $a \in [N]$, and thus the posterior on rewards for arms $a \in [N]$ satisfies that $\mu_{a_{h'}} = 1 - \delta$ and $\mu_a = \delta$ for $a \in [N] \setminus \{a_{h'}\}$. On the other hand, because $\tau_{h-1}$ only has actions $a \in [N]$, the posterior on $b$ is still Bernoulli with parameter $\epsilon$. Hence, a draw $\hat{\boldsymbol{\mu}} \sim P_\theta[\cdot \mid \tau_{h-1}]$ has distribution $\boldsymbol{\mu}(a_{h'}, \hat{b})$, where $\hat{b}$ is uniform Bernoulli. If $\hat{b} = 1$, then $\max_a \boldsymbol{\mu}(a_{h'}, \hat{b})\big|_a = 1$ for $a = N + 1$; otherwise, $\max_a \boldsymbol{\mu}(a_{h'}, \hat{b})\big|_a = 1 - \delta$, attained by $a = a_{h'}$. Hence, the update rules in Lines 6 and 8 are equivalent to $k$-TS$(\theta)$.
- *Case 2.* $\tau_{h-1}$ contains no action $r_{h'} = 1 - \delta$. Hence, it can be ruled out that the top action is not among $a_1, \ldots, a_{h-1}$, and thus $(\bar{a}, \bar{b}) \sim P_\theta[\cdot \mid \tau_{h-1}]$ are distributed independently as $\bar{a} \overset{\text{unif}}{\sim} [N] \setminus \{a_1, \ldots, a_{h-1}\}$ and $\bar{b} \sim \text{Bernoulli}(\epsilon)$. Thus, the update rules in Lines 6 and 11 correctly execute $k$-TS$(\theta)$.

$\square$

---

**Algorithm 4** $k$-TS sampling under $\theta$ (lower bound construction)

---

1: **for** $h = 1, \ldots, H$ **do**
2:     Sample $\hat{b}_{h,1}, \ldots, \hat{b}_{h,k} \sim \text{Bernoulli}(\epsilon)$.
3:     **if** there exists $h' < h$ with $a_{h'} = N + 1$ **then**
4:         Select action `// having played` $a = N + 1$ `in the past gives away the best arm`

$$a_h = \begin{cases} N + 1 & r_{h'} = 1 \\ \frac{2N}{\delta} r_{h'} & r_{h'} \neq 1. \end{cases}$$

5:     **else if** $\max_{i \in [k]} b_{h,i} = 1$ **then**
6:         Select action $a_h = N + 1$ `//` $\arg\max_a \max_i \boldsymbol{\mu}(\bar{a}, b_{h,i})\big|_a = N + 1$, for any reference action $\bar{a} \in [N]$.
7:     **else if** there exists $h' < h$ such that $r_{h'} = 1 - \delta$ **then**
8:         Select action $a_h = a_{h'}$, where $h'$ has $r_{h'} = 1 - \delta$.
9:     **else**
10:        Sample $\hat{a}_{h,1}, \ldots, \hat{a}_{h,k} \overset{\text{unif}}{\sim} [N] \setminus \{a_1, \ldots, a_{h-1}\}$
11:        Select $a_h$ be any element of $\{\hat{a}_{h,1}, \ldots, \hat{a}_{h,k}\}$.

---

**Alternative Instance** We now construct an alternative instance $\theta'$ by agreeing with $\theta$ but always setting $b = 0$:

$$\boldsymbol{\mu} = \boldsymbol{\mu}(\bar{a}, 0), \quad \bar{a} \overset{\text{unif}}{\sim} [N].$$

It is clear that $\text{TV}(\theta \parallel \theta') = \epsilon$, because the two differ only in the coin-flip of $b$. Under $\theta'$, $a = N + 1$ never has the largest reward and is therefore never sampled. Therefore, defining the event $\bar{\mathcal{E}} = \{\exists h \in [H] : a_h = N + 1\}$, and its complement

$$\bar{\mathcal{E}}^c := \{a_h \neq N + 1 \,\forall h \in [H]\},$$

we see that

$$E_{\theta,k\text{-TS}(\theta)}\left[\sum_{h=1}^{H} r_h \mid \bar{\mathcal{E}}^c\right] = R(\theta, k\text{-TS}(\theta')).$$

That is, the conditional expected reward garnered by well-specified $k$-TS($\theta$) under the event that $a_h \neq N+1$ for all $h$ is equal to the expected reward of misspecified $k$-TS($\theta'$).

**Comparing the instances**   To compare the instances, write out $\mathcal{E}_h := \{a_{h'} \neq N+1, \quad \forall h' < h \text{ and } a_h = 1\}$. Then, $\bar{\mathcal{E}}$ is the disjoint union of $\mathcal{E}_1, \ldots, \mathcal{E}_H$, so that

$$R(\theta, k\text{-TS}(\theta)) = \sum_{h=1}^{H} P_{\theta,k\text{-TS}(\theta)}[\mathcal{E}_h] \cdot E_{\theta,k\text{-TS}(\theta)}\left[\sum_{h=1}^{H} r_h \mid \mathcal{E}_h\right]$$

$$+ P_{\theta,k\text{-TS}(\theta)}[\bar{\mathcal{E}}^c] \cdot E_{\theta,k\text{-TS}(\theta)}\left[\sum_{h=1}^{H} r_h \mid \bar{\mathcal{E}}^c\right].$$

As noted above, $E_{\theta,k\text{-TS}(\theta)}[\sum_{h=1}^{H} r_h \mid \bar{\mathcal{E}}^c] = R(\theta, k\text{-TS}(\theta'))$. If $\mathcal{E}_h$ occurs, then the best action $a_h$ is identified, and reward at least $1 - \delta$ is accrued on stages $h' > h$. Introducing the shorthand $P[\mathcal{E}_h] = P_{\theta,k\text{-TS}(\theta)}[\mathcal{E}_h]$ and similarly for $\bar{\mathcal{E}}, \bar{\mathcal{E}}^c$, we then find

$$R(\theta, k\text{-TS}(\theta)) \geq (1 - \delta)\sum_{h=1}^{H} P[\mathcal{E}_h](H - h) + P[\bar{\mathcal{E}}^c]R(\theta, k\text{-TS}(\theta')).$$

Therefore, subtracting $R(\theta, k\text{-TS}(\theta'))$ from both sides,

$$R(\theta, k\text{-TS}(\theta)) - R(\theta, k\text{-TS}(\theta')) \geq (1 - \delta)\sum_{h=1}^{H} P[\mathcal{E}_h](H - h) - P[\bar{\mathcal{E}}]R(\theta, k\text{-TS}(\theta')).$$

Let us continue simplifying the above two terms. First, since all draws $b_{h,i}$ are i.i.d. Bernoulli with parameter $\epsilon \leq 1/kH$, we have

$$\begin{aligned}
P[\mathcal{E}_h] &= P[a_{h'} \neq N+1, \forall h' < h] \cdot P[a_h = N+1 \mid a_{h'} \neq N+1, \forall h' < h] \\
&= P[\max_i b_{h',i} = 0, \quad \forall h' < h] \cdot P[\max_i b_{h,i} = 1] \\
&\geq \left(1 - \sum_{h'=1}^{h-1}\sum_{i=1}^{k} P[b_{h',i} = 1]\right) \cdot P[\max_i b_{h,i} = 1] \\
&= (1 - k(h-1)\epsilon)(1 - (1-\epsilon)^k) \\
&\geq (1 - k(h-1)\epsilon)(1 - e^{-k\epsilon}) \\
&\geq (1 - k(h-1)\epsilon)(k\epsilon - \frac{(k\epsilon)^2}{2}) \\
&\geq (1 - kH\epsilon)(1 - k\epsilon)(k\epsilon) \\
&\geq (1 - kH\epsilon)^2(k\epsilon).
\end{aligned}$$

Thus,

$$(1 - \delta)\sum_{h=1}^{H} P[\mathcal{E}_h](H - h) \geq (1 - kH\epsilon)^2(1 - \delta) \cdot k\epsilon \sum_{h=1}^{H}(H - h) = (1 - kH\epsilon)^2(1 - \delta)k\epsilon\binom{H - 1}{2}.$$

$$(D.1)$$

On the other hand, by a union bound, we have $P[\bar{\mathcal{E}}] \leq kH\epsilon$. Moreover, we have

$$R(\theta, k\text{-TS}(\theta')) \leq HP_{\theta,k\text{-TS}(\theta')}[\exists h : a_h = \bar{a}] + H\delta,$$

since under $k$-TS$(\theta')$, if $a_h \neq \bar{a}$ for all $h$, then $k$-TS$(\theta')$ always selects actions $a \in [N] \setminus \bar{a}$, all of which return reward $\delta$. On the other hand,

$$
\begin{aligned}
P_{\theta, k\text{-TS}(\theta')}[\exists h : a_h = \bar{a}] &= \sum_{h=1}^{H} P_{\theta, k\text{-TS}(\theta')}[a_h = \bar{a} \mid a_{1:h-1} \neq \bar{a}] P_{\theta, k\text{-TS}(\theta')}[a_{1:h-1} \neq \bar{a}] \\
&\leq \sum_{h=1}^{H} P_{\theta, k\text{-TS}(\theta')}[a_h = \bar{a} \mid a_{1:h-1} \neq \bar{a}] \\
&= \sum_{h=1}^{H} \frac{1}{N - (h-1)} \leq \frac{H}{N - H}.
\end{aligned}
$$

Therefore,

$$
R(\theta, k\text{-TS}(\theta')) \leq \frac{H^2}{N - H} + H\delta, \quad P[\bar{\mathcal{E}}] R(\theta, k\text{-TS}(\theta')) \leq kH^2 \epsilon \left( \frac{H}{N - H} + \delta \right).
$$

Hence, combining with Equation (D.1), we conclude

$$
\begin{aligned}
R(\theta, k\text{-TS}(\theta)) - R(\theta, k\text{-TS}(\theta')) &\geq (1 - \delta) \sum_{h=1}^{H} P[\mathcal{E}_h](H - h) - P[\bar{\mathcal{E}}] R(\theta, k\text{-TS}(\theta')). \\
&\geq k\epsilon H^2 \left( (1 - kH\epsilon)^2 (1 - \delta) \frac{(H-1)(H-2)}{2H^2} - \frac{H}{N - H} - \delta \right).
\end{aligned}
$$

By tuning the above bound, we can see that there is a universal constant $c_0$ such that, for any $\eta \in (0, 1/4)$, taking $\epsilon^{-1} \geq \frac{\eta}{c_0 k H}$, $H \geq \frac{c_0}{\eta}$ and $N + 1 = H \lceil \frac{c_0}{\eta} \rceil$ and $\delta = \frac{\eta}{\mathcal{E}}$ ensures that

$$
R(\theta, k\text{-TS}(\theta)) - R(\theta, k\text{-TS}(\theta')) \geq \left( \frac{1}{2} - \eta \right) k\epsilon H^2.
$$

## D.2 A simple converse to Proposition 3.4

**Proposition D.2** (Lower Bound). *Let $H, k \geq 1$ be given. Then there exists a pair of priors, $\theta$ and $\theta'$ such that*

$$
\mathrm{TV}(P_{\theta, k\text{-TS}(\theta)}(\boldsymbol{\mu}, \tau_H) \parallel P_{\theta, k\text{-TS}(\theta')}(\boldsymbol{\mu}, \tau_H)) \geq \frac{kH}{2} \mathrm{TV}(P_\theta \parallel P_{\theta'}).
$$

*In particular, since Thompson sampling is $k$-TS for $k = 1$, we have*

$$
\mathrm{TV}(P_{\theta, \mathsf{TS}(\theta)}(\boldsymbol{\mu}, \tau_H) \parallel P_{\theta, \mathsf{TS}(\theta')}(\boldsymbol{\mu}, \tau_H)) \geq \frac{H}{2} \mathrm{TV}(P_\theta \parallel P_{\theta'}).
$$

*Moreover, the rewards are Bernoulli.*

*Proof.* Recall the $k$-TS$(\theta)$ selection rule at time $h$:

1. Sample means $\boldsymbol{\mu}^{(1)}, \ldots, \boldsymbol{\mu}^{(k)}$ from the posterior $P_\theta[\cdot \mid \tau_{h-1}]$.
2. Select action
$$
a_h \in \arg\max_{a \in \mathcal{A}} \max\{\mu_a^{(1)}, \ldots, \mu_a^{(k)}\}.
$$

Now we will show the lower bound for the following two prior distributions.

- $P_\theta$ places all its probability mass on the mean vector $(1/2, 0)$.
- $P_{\theta'}$ places $1 - \epsilon$ of its probability mass on the mean vector $(1/2, 0)$ and $\epsilon$ of its probability mass on the mean vector $(1/2, 1)$.

Clearly we have $\mathrm{TV}(P_\theta, P_{\theta'}) = \epsilon$. Moreover, we have the following three observations.

(a) $k$-TS($\theta$) will always pull arm 1. Thus, to give a lower bound on the total variation distance between the behavior of $k$-TS($\theta$) and $k$-TS($\theta'$), it suffices to lower bound the probability that $k$-TS($\theta'$) pulls arm 2 in the course of $H$ interactions.

(b) The posterior distribution under $\theta'$ remains unchanged when arm 1 is pulled. Thus, the probability that $k$-TS($\theta'$) never pulls arm 2 in the course of $H$ interactions is the $H$-fold product of the probability that $k$-TS($\theta'$) does not pull arm 2 in one round of interaction.

(c) The probability that $k$-TS($\theta'$) does not pull arm 2 in one round of interaction is exactly the probability that $k$ i.i.d. draws from $\theta'$ does not yield an instance of $(1/2, 1)$, i.e. $(1 - \epsilon)^k$.

Combining (a), (b), and (c), and assuming $\epsilon \leq \frac{1}{2Hk}$,

$$
\begin{aligned}
\mathrm{TV}(P_{\theta, k\text{-TS}(\theta)}(\boldsymbol{\mu}, \boldsymbol{\tau}_H) \,\|\, P_{\theta, k\text{-TS}(\theta')}(\boldsymbol{\mu}, \boldsymbol{\tau}_H)) &\geq 1 - (1 - \epsilon)^{Hk} \\
&\geq 1 - e^{-\epsilon Hk} \\
&\geq \epsilon Hk - (\epsilon Hk)^2 \\
&\geq \frac{1}{2} \epsilon Hk.
\end{aligned}
$$

In the above we have used the inequalities $1 + x \leq e^x$ for all $x \in \mathbb{R}$ and $e^{-x} \leq 1 - x + x^2$ for $x \in [0, 1]$, and the assumption that $\epsilon \leq \frac{1}{2Hk}$. $\qquad\square$

## E  General Bayesian Decision-Making

### E.1  POMDP Formalism and Special Cases

We begin by presenting a general formalism for Bayesian POMDPs and listing some illustrative examples. For a more thorough introduction to Bayesian reinforcement learning, we direct the reader to [GMPT15].

**Bayesian POMDP**   In a Bayesian POMDP, the priors $\{P_\theta : \theta \in \Theta\}$ are distributions over POMDP *environments* $\phi \in \Phi$ with (possibly unobserved) *states* $s_h \in \mathcal{S}$, *observations* $y_h \in \mathcal{Y}$, *actions* $a_h \in \mathcal{A}$, and *rewards* $r_h \in \mathbb{R}$. Common to all POMDP environments are (possibly time-varying) transition functions

$$
\mathsf{P}_h : \Phi \times \mathcal{A} \times \mathcal{S} \times \mathcal{Y} \to \Delta(\mathcal{S} \times \mathcal{Y} \times \mathbb{R})
$$

which induce distributions

$$
\mathsf{P}_h(s_{h+1}, y_{h+1}, r_h \mid a_h, s_h, y_h, \phi).
$$

There is also an initial distribution $\mathsf{P}_0 : \Phi \to \Delta(\mathcal{S} \times \mathcal{Y})$ giving an initial distribution of $(s_1, y_1) \sim \mathsf{P}_0(\cdot \mid \phi)$. The relevant definition of trajectories revealed to the learner are:

$$
\tau_h := (a_1, y_1, r_1, \ldots, a_h, y_h, r_h, y_{h+1}) \tag{E.1}
$$

**POMDP Algorithms**   A randomized POMDP algorithm is formally identical to a bandit one. A family of randomized POMDP algorithms $\mathsf{alg}(\theta)$ is a specified by a distribution $\mathcal{D}_{\mathrm{seed}}$ (independent of $\theta$), a domain $\Xi$ over random seeds $\boldsymbol{\xi}$, and step-wise mappings $f_1, \ldots, f_H$ from trajectories, the random seed, and parameters $\theta$ to distributions over actions:

$$
f_h(\tau_{h-1}, \boldsymbol{\xi} \mid \theta) : \{h\text{-trajectories}\} \times \Xi \times \Theta \to \Delta(\mathcal{A}).
$$

Each $\mathsf{alg}(\theta)$ operates as follows:

- $\boldsymbol{\xi}$ is drawn from $\mathcal{D}_{\mathrm{seed}}$ at the start of the episode before interaction.
- At each step $h$, $a_h$ is chosen independently according to $a_h \sim f_h(\tau_{h-1}, \boldsymbol{\xi} \mid \theta)$.

Note again the *two* sources of randomness: the draw of $a_h$ and the initial random seed $\boldsymbol{\xi}$; see Remark 3 for details. In short: the types of algorithms we allow for not only include those that are implemented only with the randomness in the choice of $a_h$ but also those that use initial random seeds $\boldsymbol{\xi}$ to induce correlations across steps $h$.

**Interaction Protocol**

- $\boldsymbol{\xi}$ is drawn from $\mathcal{D}_{\text{seed}}(\cdot)$ at the start of the episode before interaction.
- Then, an environment $\phi$ is drawn from $P_\theta$ (independent of $\boldsymbol{\xi}$)
- An initial state and observation are drawn as $(s_1, y_1) \sim \mathsf{P}_0(s, y \mid \phi)$, and $\tau_0 = (y_1)$ is revealed to the learner.
- Subsequently, for all $h \in \{1, 2, \ldots, H\}$,
  1. The learner selects action $a_h$ with independent randomness via $a_h \sim f_{h-1}(\tau_{h-1}, \boldsymbol{\xi} \mid \theta)$.
  2. The environment draws a state, observation, and reward
  $$(s_{h+1}, y_{h+1}, r_h) \sim \mathsf{P}_h(s, y, r \mid a_h, s_h, y_h, \phi)$$
  3. The agent observes reward $r_h$ and observation $y_{h+1}$. The triple $(r_h, a_h, y_{h+1})$ is then appended to $\tau_{h-1}$ to form $\tau_h$.

As in the bandit case, the reward accrued by alg is

$$R(\theta, \mathsf{alg}) := E_{\theta, \mathsf{alg}} \left[ \sum_{h=1}^H r_h \right],$$

where $E_{\theta, \mathsf{alg}}$ denotes expectations under $\phi \sim P_\theta$, the transitions $\mathsf{P}_h(\cdot \mid \cdot, \phi)$, and the choice of actions $a_h$ as above.

## E.2 Special Cases of Bayesian POMDPs

The Bayesian POMDP set up encompasses a number of special cases:

**Mean-Parametrized Bayesian Bandits:**   The Bayesian-Bandit setting considered in the main body can be viewed as a POMDP with no state, no observation, and where the instance $\phi$ is summarized by the mean parameter $\boldsymbol{\mu}$. The only randomness after $\boldsymbol{\mu}$ is selected is the generation of rewards, that is $r_h \sim \mathsf{P}(r \mid a_h, \boldsymbol{\mu})$, which is equivalent to the distribution $\mathcal{D}(\boldsymbol{\mu})$ described in the main text. For example, $\mathsf{P}(r \mid a_h = a, \boldsymbol{\mu}) = \mathcal{N}(\mu_a, \sigma^2)$ for some fixed $\sigma^2 > 0$.

Note that the distribution over mean vectors $\boldsymbol{\mu}$ may arise to form means with, for example, linear structure (e.g.[AL17]). For example, consider an instance where each action $a$ corrsponds to a vector $\mathbf{v}_a$, and each $\boldsymbol{\mu}$ to a vector $\mathbf{w}_{\boldsymbol{\mu}}$ drawn from a prior, say, $\mathcal{N}(0, \Sigma_\theta)$, for which

$$\mu_a = \langle \mathbf{v}_a, \mathbf{w}_{\boldsymbol{\mu}} \rangle.$$

**General-Reward Bayesian Bandits:**   More generally, we could consider Bayesian instances where the prior $P_\theta$ over models $\phi$ governs not only the reward means $\boldsymbol{\mu}$ but can encode general conditional distributions of rewards. For example, we may have priors over mean-variance vector pairs $\phi = (\boldsymbol{\mu}, \boldsymbol{\sigma}) \in \mathbb{R}^{2|\mathcal{A}|}$, and conditional rewards $\mathsf{P}(r \mid a_h = a, \phi) = \mathcal{N}(\mu_a, \sigma_a^2)$.

**Mean-Parametrized and General Contextual Bandits:**   Bayesian POMDPs also capture the contextual bandits formalism. Here, contexts $x_h \in \mathcal{X}$, are directly revealed to the learner and correspond to both states and observations (i.e. $x_h := y_h = s_h$ ), and are drawn i.i.d. from a law $\mathsf{P}_{\text{context}}(x \in \cdot \mid \phi)$. Then, the distribution of rewards is selected depending on the context $\mathsf{P}_{\text{reward}}(r_h \mid x_h, a, \phi)$. In other words, the transition distribution $\mathsf{P}(x_{h+1}, r_h \mid s_h, a, \phi)$ is the product distribution of $\mathsf{P}_{\text{context}}(x \mid \phi)$ and $\mathsf{P}_{\text{reward}}(r_h \mid x_h, a, \phi)$. Note that the next context $x_{h+1}$ is independent of all other randomness given the instance $\phi$, so the dynamics are trivial.

For example, we might have that contexts are vectors $\mathbf{x} \in \mathcal{R}^d$ (bolded to denote that they are vectors), actions are identified with vectors $\mathbf{v}_a \in \mathbb{R}^n$, $\phi = (\boldsymbol{\Sigma}_\phi, \mathbf{L}_\phi) \in \mathbb{R}^{d \times d} \times \mathbb{R}^{n \times d}$, and that contexts and rewards are drawn

$$\mathbf{x}_h \overset{\text{i.i.d.}}{\sim} \mathcal{N}(0, \boldsymbol{\Sigma}_\phi), \quad \mathsf{P}(r_h \mid \mathbf{s}_h, a_h = a) = \mathcal{N}\left(\mathbf{v}_a^\top \mathbf{L}_\phi \mathbf{x}_h, I\right).$$

The above is an example of Bayesian linear contextual bandits.

The special case of contextual bandits studied in the main text are where the model is parameterized by the mean ($\phi = \boldsymbol{\mu}$) and the distribution over contexts does not depend on the model: formally, contexts are drawn $x_h \sim \mathcal{D}_x$ which does not depend on the realized model $\phi$, and where (as in reward-parametrized bandits) rewards are drawn as $r_h \sim \mathcal{D}(\boldsymbol{\mu}, x_h)$ for some $\mathcal{D} : \mathbb{R}^{\mathcal{A}} \times \mathcal{X} \to \Delta(\mathbb{R})$.

**Bayesian MDPs** One final case is that of the Bayesian MDP, where the agent observes the state: $s_h = y_h$. Bayesian MDPs, and in particular, Bayesian Tabular MDPs have recieved extensive study [OVR17]. More general results were studied in [GMM14].

**Bayesian Control Problems** In addition, many online control settings – notably, the online Linear Quadratic Regulator – satisfy the Markov property, and hence are examples of Bayesian MDPs when formulated in the Bayesian setting [AYS15, AL18]. Bayesian control Kalman Filtering and Linear Quadratic Gaussian control may also be formulated as a PODMP.

## E.3 Formal Guarantees

We now state the formal guarantees for the Bayesian POMDP setting, which straightforwardly specialize to the bandit decision-making setting described in the main text.

**Monte Carlo Property** Mirroring the bandit case, we let $P_\theta(\phi \mid \tau_h)$ denote the conditional distribution of the POMDP environment $\phi$ given the trajector $\tau_h$ (using the generalization of trajectories stated in Equation (E.1)), and $P_{\mathsf{alg}}(a_h \mid \tau_h)$ the conditional distribution of actions $a_h$ under algorithm alg given the trajectory.

**Definition E.1** (Generalized $n$-Monte Carlo). We say that a family of Bayesian POMDP algorithms $\{\mathsf{alg}(\theta) : \theta \in \Theta\}$ satisfy the generalized $n$-Monte Carlo property if, for all possible trajectories $\tau_h$,

$$\mathrm{TV}(P_{\mathsf{alg}(\theta)}(a_h \mid \tau_{h-1}) \parallel P_{\mathsf{alg}(\theta')}(a_h \mid \tau_{h-1})) \le \mathrm{TV}(P_\theta(\phi \mid \tau_{h-1}) \parallel P_{\theta'}(\phi \mid \tau_{h-1})).$$

**Tail Expectations** Second, we require the relevant notion of *tail expectation*. We propose a slightly different definition than the one given for Bayesian bandits, due to the fact that different algorithms may visit different states under the same POMDP environment. We now introduce the average conditional reward (ACR):

$$\bar{r}_{H,\theta,\mathsf{alg}} := \frac{1}{H} \sum_{h=1}^{H} \mathsf{E}[r_h \mid a_h, s_h, y_h, \phi], \quad \text{where } (a_{1:H}, s_{1:H}, y_{1:H}) \sim P_{\theta,\mathsf{alg}}. \tag{E.2}$$

Above, we use $\mathsf{E}[r_h \mid a_h, s_h, y_h, \phi]$ to denote expectation over the law $\mathsf{P}(r_h \mid a_h, s_h, y_h, \phi)$, and note that the conditional *does not* depend on $\theta$ or the specification of alg. In the special case of bandits, notice that Equation (E.2) simplies to

$$\bar{r}_{H,\theta,\mathsf{alg}} := \frac{1}{H} \sum_{h=1}^{H} \mu_{a_h}.$$

**Definition E.2** (POMDP Tail expectation). Given an algorithm $\mathsf{alg}(\cdot)$ parametrized by $\theta \in \Theta$, we define

$$\bar{\Psi}_\theta(p) = \Psi_{|\bar{r}_{H,\theta,\mathsf{alg}(\theta)}|}(p), \quad \bar{\Psi}_{\theta/\theta'}(p) = \Psi_{|\bar{r}_{H,\theta,\mathsf{alg}(\theta')}|}(p),$$

where $\Psi_X(p)$ is the tail expectation of a nonnegative random variable $X$ as in Definition B.2.

Unlike the bandit tail expectations $\Psi_\theta$ in Definition B.2 which depend only on the mean parameter $\boldsymbol{\mu}$, the tail expectations above depend on both the ACR under $\theta$, as in $\bar{\Psi}_\theta$, and the ACR under $\theta, \mathsf{alg}(\theta')$ as in $\bar{\Psi}_{\theta/\theta'}$. In particular, both terms depend on the family of algorithms $\mathsf{alg}(\cdot)$. This is important in POMDP environments with unbounded state spaces (e.g. linear control), where sensitivity can be quite poor if the misspecified policy $\mathsf{alg}(\theta')$ visits much lower-reward states under $\theta$ than the well-specified $\mathsf{alg}(\theta)$. Nevertheless, for bandits, $\bar{\Psi}_\theta$ and $\Psi_\theta$ are qualitatively similar because

$$|\bar{r}_{H,\theta,\mathsf{alg}(\theta)}| \le \sup_a |\mu_a|,$$

and similarly for $\bar{r}_{H,\theta,\mathsf{alg}(\theta')}$.

**Strongly $B$-Bounded distributions.** To interpret the tail conditions, we consider the special case of *strongly $B$* bounded distributions.

**Definition E.3.** We say that $P_\theta$ is *strongly B-bounded* if, with probability 1, $\mathbb{E}_\theta[r_h \mid a_h, \tau_{h-1}, \phi] \in [-\frac{B}{2}, \frac{B}{2}]$ conditioned on any action $a_h$, trajectory $\tau_{h-1}$ and $\phi$.

In the special case of bandits, strong $B$-boundedness implies that $\mu_a \in [-\frac{B}{2}, \frac{B}{2}]$ with probability one, and is therefore slightly stronger than $B$-boundedness, which states that $\mathrm{diam}(\boldsymbol{\mu}) \le B$. Observe that if $P_\theta$ is strongly $B$-bounded (Definition E.3), then

$$|\bar{r}_{H,\theta,\mathsf{alg}(\theta)}| \le \frac{B}{2}, \quad \text{and} \quad |-\bar{r}_{H,\theta,\mathsf{alg}(\theta')}| \le \frac{B}{2}.$$

Therefore $\bar{\Psi}_\theta(p) + \bar{\Psi}_{\theta/\theta'}(p) \le B$.

We are now ready to state our general theorem, consisting of (a) a total variation bound, (b) a reward bound for strongly $B$-bounded rewards, and (c) a reward bound for general tail expectations:

**Theorem E.1.** *Let* $\mathsf{alg}(\cdot)$ *satisfy the $n$-Monte Carlo property on horizon $H$, and consider two priors* $\theta, \theta' \in \Theta$ *with* $\varepsilon = \mathrm{TV}(P_\theta \parallel P_{\theta'})$

*(a) Let $P_H = P_{\theta,\mathsf{alg}(\theta)}(\phi, \tau_H, s_{1:H+1})$ and $P'_H = P_{\theta,\mathsf{alg}(\theta')}(\phi, \tau_H, s_{1:H+1})$ denote the joint law of the environment $\phi$, trajectory $\tau_H$, and sequence of states $s_{1:H}$ under prior $P_{\theta,\mathsf{alg}(\theta)}$ and $P_{\theta,\mathsf{alg}(\theta')}$, respectively. Then,*

$$\mathrm{TV}(P_H \parallel P'_H) \le 2H\varepsilon.$$

*(b) If $P_\theta$ is strongly $B$-bounded, then $|R(\theta, \mathsf{alg}(\theta)) - R(\theta, \mathsf{alg}(\theta'))| \le 2nBH^2\varepsilon$.*
*(c) For general tail expectations, the following bound holds:*

$$|R(\theta, \mathsf{alg}(\theta)) - R(\theta, \mathsf{alg}(\theta'))| \le 2nH^2\varepsilon \cdot \left(\bar{\Psi}_\theta(2H\varepsilon) + \bar{\Psi}_{\theta/\theta'}(2H\varepsilon)\right).$$

Note that part (a) of the above theorem generalizes Proposition 3.4, part (b) generalizes Theorem 3.2(a).

*Proof of Theorem E.1.* We begin by establishing part (a), from which parts (b) and (c) follow.

**Part a.** Let us start by developing an analogoue of Lemma B.9. To do so, we invoke Lemma B.11, replacing $\boldsymbol{\mu}$ with the POMDP environment denoted by $\phi$, and with $P_H = P_{\theta,\mathsf{alg}(\theta)}$ and $P'_H = P_{\theta,\mathsf{alg}(\theta')}$, and with augmented trajectories $\tilde{\tau}_{h-1} = (\tau_{h-1}, s_{1:h})$ To invoke the latter lemma, we need to check three conditions.

1. *Condition 1: $\tilde{\tau}_{h-1}$ is a deterministic function of $\tilde{\tau}_h$.* This is definitionally true, even in the PODMP setting.
2. *Condition 2: The conditional distributions of $\tau_h$ given $a_h, \tau_{h-1}$ and $\phi$ are the same: $P'_H(\tau_h \mid a_h, \tilde{\tau}_{h-1}, \phi) = P_H(\tilde{\tau}_h \mid a_h, \tau_{h-1}, \phi)$.* This follows because $P(\tau_h \mid a_h, \tilde{\tau}_{h-1}, \phi) = \mathsf{P}(s_{h+1}, y_{h+1}, r_h \mid a_h, s_h, y_h, \phi)$ for $P \in \{P_H, P'_H\}$, were $\mathsf{P}$ is the transition function.
3. *Condition 3: Under both $P_H$ and $P'_H$, $a_h$ is independent of $\phi$ given $\tilde{\tau}_{h-1}$.* Following the same logic as in Lemma B.9 and letting $P = P_H$, we have

$$\begin{aligned}
P(a_h, \phi \mid \tau_{h-1}) &= \mathbb{E}\left[\mathbb{E}\left[P(a_h, \phi \mid \tilde{\tau}_{h-1}, \boldsymbol{\xi}) \mid \boldsymbol{\xi}\right] \mid \tilde{\tau}_{h-1}\right] \\
&= \mathbb{E}\left[\mathbb{E}\left[f_h(a_h \mid \tau_{h-1}, \boldsymbol{\xi})P(\phi \mid \tilde{\tau}_{h-1}, \boldsymbol{\xi}) \mid \boldsymbol{\xi}\right] \mid \tilde{\tau}_{h-1}\right] \\
&= \mathbb{E}\left[\mathbb{E}\left[f_h(a_h \mid \tau_{h-1}, \boldsymbol{\xi}) \mid \boldsymbol{\xi}\right] P(\phi \mid \tilde{\tau}_{h-1}) \mid \tilde{\tau}_{h-1}\right] \\
&= P(a_h \mid \tilde{\tau}_{h-1})P(\phi \mid \tilde{\tau}_{h-1})
\end{aligned}$$

where the second equality follows from the fact that $a_h \sim f_h(\cdot \mid \boldsymbol{\xi}, \tau_{h-1})$ and the third line follows from the fact that $\phi$ is independent of $\boldsymbol{\xi}$ conditioned on $\tau_{h-1}$. The same argument holds symmetrically for $P' = P'_H$.

As a consequence of these three conditions, it holds that for $P_H = P_{\theta,\mathsf{alg}(\theta)}(\boldsymbol{\mu}, \tau_H, s_{1:H+1})$ and $P'_H = P_{\theta,\mathsf{alg}(\theta')}(\boldsymbol{\mu}, \tau_H, s_{1:H+1})$,

$$\mathrm{TV}(P_H \| P'_H) \leq \sum_{h=1}^{H} \mathbb{E}_{\tilde{\tau}_{h-1} \sim P_H} \mathrm{TV}(P_H(a_h \mid \tilde{\tau}_{h-1}) \| P'_H(a_h \mid \tilde{\tau}_{h-1}))$$

$$= \sum_{h=1}^{H} \mathbb{E}_{\tau_{h-1} \sim P_H} \mathrm{TV}(P_H(a_h \mid \tau_{h-1}) \| P'_H(a_h \mid \tau_{h-1})).$$

where we use the fact that $P_H(a_h \mid \tilde{\tau}_{h-1}) = P_H(a_h \mid \tau_{h-1})$, and similarly under $P'_H$. Next, using the $n$-Monte Carlo Property, and that $P_H(a_h \mid \tau_{h-1}) = P_{\mathsf{alg}(\theta)}(a_h \mid \tau_{h-1})$ (and similarly for $P'_H$ and $P_{\mathsf{alg}(\theta')}$),

$$\mathrm{TV}(P_H \| P'_H) \leq n \sum_{h=1}^{H} \mathrm{TV}(P_\theta(\phi \mid \tau_{h-1}) \| P_{\theta'}(\phi \mid \tau_{h-1})).$$

By invoking the de-conditioning lemma, Lemma B.10, with $Q = P_{\theta,\mathsf{alg}(\theta)}$, $Q' = P_{\theta',\mathsf{alg}(\theta')}$, $X = \tau_{h-1}$ and $Y = \phi$, we have that

$$\mathrm{TV}(P_H \| P'_H) \leq n \sum_{h=1}^{H} 2\mathrm{TV}(Q(Y) \| Q'(Y)) = 2Hn\mathrm{TV}(P_\theta \| P_{\theta'}).$$

This establishes part (a).

**Parts b and c.** Part (b) is a consequence of part (c) and the fact that, for strongly $B$-bounded $P_\theta$, $\bar{\Psi}_\theta(\cdot) + \bar{\Psi}_{\theta/\theta'}(\cdot) \leq B$. We conclude by proving part (c), and keep the notation $\tilde{\tau}_h = (\tau_h, s_{1:h+1})$. Let $P, E$ and $P', E'$ denote probability and expectation operators under $P_{\theta,\mathsf{alg}(\theta)}$ and $P_{\theta,\mathsf{alg}(\theta')}$, respectively. Introduce the conditional reward function

$$\bar{\mu}_h(a, s, y, \phi) = \mathbb{E}[r_h \mid a_h = a, s_h = s, y_h = y, \phi].$$

By the tower rule,

$$R(\theta, \mathsf{alg}(\theta)) - R(\theta, \mathsf{alg}(\theta')) = E\left[\sum_{h=1}^{H} \bar{\mu}_h(a_h, s_h, y_h, \phi)\right] - E'\left[\sum_{h=1}^{H} \bar{\mu}_h(a_h, s_h, y_h, \phi)\right].$$

By Lemma B.4 and the fact that $P(\phi) = P'(\phi)$, there exists a coupling $Q$ such over random variables $(\phi, \tilde{\tau}_H, \tilde{\tau}'_H)$ such that both

$$Q(\phi, \tilde{\tau}_H) = P(\phi, \tilde{\tau}_H), \quad Q(\phi, \tilde{\tau}'_H) = P'(\phi, \tilde{\tau}_H),$$

and

$$Q[\tilde{\tau}_H \neq \tilde{\tau}'_H] = \mathrm{TV}(P(\phi, \tilde{\tau}_H) \| P'(\phi, \tilde{\tau}_H)).$$

Letting $(a_h, s_h, y_h)$ and $(a'_h, s'_h, y'_h)$ denote states and actions corresponding to $\tilde{\tau}_H$ and $\tilde{\tau}'_H$, and let $E_Q$ denote expectations under the coupling $Q$, we then have

$R(\theta, \mathsf{alg}(\theta)) - R(\theta, \mathsf{alg}(\theta'))$

$$= E_Q\left[\sum_{h=1}^{H} \bar{\mu}_h(a_h, s_h, y_h, \phi) - \bar{\mu}_h(a'_h, s'_h, y'_h, \phi)\right]$$

$$\overset{(i)}{=} E_Q\left[\mathbb{I}\{\tilde{\tau}_H \neq \tilde{\tau}'_H\}\left(\sum_{h=1}^{H} \bar{\mu}_h(a_h, s_h, y_h, \phi) - \bar{\mu}_h(a'_h, s'_h, y'_h, \phi)\right)\right]$$

$$= H\left(E_Q\left[\underbrace{\mathbb{I}\{\tilde{\tau}_H \neq \tilde{\tau}'_H\}}_{:=Y} \cdot \underbrace{\frac{1}{H}\sum_{h=1}^{H} \bar{\mu}_h(a_h, s_h, y_h, \phi)}_{:=X}\right] - E_Q\left[\underbrace{\mathbb{I}\{\tilde{\tau}_H \neq \tilde{\tau}'_H\}}_{:=Y} \cdot \underbrace{\frac{1}{H}\sum_{h=1}^{H} \bar{\mu}_h(a'_h, s'_h, y'_h, \phi)}_{:=X'}\right]\right)$$

where $(i)$ uses that $\sum_{h=1}^{H} \bar{\mu}_h(a_h, s_h, y_h, \phi) - \bar{\mu}_h(a'_h, s'_h, y'_h, \phi) = 0$ whenever $\tilde{\tau}_H = \tilde{\tau}'_H$. By the triangle inequality,

$$|R(\theta, \mathsf{alg}(\theta)) - R(\theta, \mathsf{alg}(\theta'))| \le H \left( \mathbb{E}\left[|X| \cdot Y\right] + E_Q\left[|X'| \cdot Y\right] \right).$$

Since $|X|$ and $|X'|$ are nonnegative, and that $Y \in [0, 1]$ satisfies

$$\mathbb{E}[Y] = Q[\tilde{\tau}_H \ne \tilde{\tau}'_H] = \mathrm{TV}(P(\phi, \tilde{\tau}_H) \parallel P'(\phi, \tilde{\tau}_H)) \le \delta := 2nH\varepsilon,$$

where the first equality is from our choice of coupling $Q$ and the inequality follows from part (a) of the theorem. Hence, by definition of the tail expectation functional which maximizes the correlation with $[0, 1]$-bounded random variables $Y$ satisfying the above constraints (Definition B.2),

$$|R(\theta, \mathsf{alg}(\theta)) - R(\theta, \mathsf{alg}(\theta'))| \le H\delta \left( \Psi_{|X|}(\delta) + \Psi_{|X'|}(\delta) \right).$$

Finally, we observe that $X$ has the same distribution of $\bar{r}_{H,\theta,\mathsf{alg}(\theta)}$ and $X'$ the distribution of $\bar{r}_{H,\theta,\mathsf{alg}(\theta')}$. Hence, $\Psi_{|X|}(\delta) = \bar{\Psi}_\theta(\delta)$ and $\Psi_{|X'|}(\delta) = \bar{\Psi}_{\theta/\theta'}(\delta)$. We therefore conclude

$$|R(\theta, \mathsf{alg}(\theta)) - R(\theta, \mathsf{alg}(\theta'))| \le H\delta \left( \bar{\Psi}_\theta(\delta) + \bar{\Psi}_{\theta/\theta'}(\delta) \right)$$
$$= 2nH^2\varepsilon \cdot \left( \bar{\Psi}_\theta(2H\varepsilon) + \bar{\Psi}_{\theta/\theta'}(2H\varepsilon) \right). \qquad \square$$

### E.4 The Monte-Carlo property in POMDPs

We conclude this section by mentioning that, as in the bandit setting, any algorithm whose actions depend only on independent samples of environments drawn from the posterior distribution is $n$-Monte Carlo. Thus, Algorithm 5, which is the POMDP generalization of Algorithm 2, is $k$-Monte Carlo, where $k$ is the number of environments sampled from the posterior. The proof of this fact is identical to the proof of Lemma C.2. Similarly, along the lines of Lemma C.3, one can establish the Monte-Carlo property for a suitable generalizations of the 2-RHC algorithm (Algorithm 3); for brevity we omit details.

---

**Algorithm 5** $(k, f_{1:H})$-Posterior Sampling $((k, f_{1:H})$-PosteriorSample$(\theta))$

---

1: **Input:** Prior $\theta$, sample size $k \in \mathbb{N}$, functions $f_1, \ldots, f_H : \mathbb{R}^{\Phi \times k} \to \Delta^{\mathcal{A}}$.
2: **for** $h = 1, \ldots, H$ **do**
   `// action selection at step` $h$
3: $\quad$ Sample $\phi^{(1)}, \ldots, \phi^{(k)}$ independently from the posterior $P_\theta[\cdot \mid \tau_{h-1}]$
4: $\quad$ Select action $a_h \sim f_h(\cdot \mid \phi^{(1)}, \ldots, \phi^{(k)})$.

---

## F Accuracy of moment estimators

### F.1 Beta priors and Bernoulli rewards

We first show how to translate sufficiently good error bounds in parameter estimation for regular exponential families into bounds on total variation error.

**Lemma F.1.** *Let $\{P_{\boldsymbol{\theta}} : \boldsymbol{\theta} \in \Theta\}$ be a standard exponential family with natural parameter space $\Theta \subset \mathbb{R}^p$. For any $\boldsymbol{\theta} \in \Theta$, there exist $C, c > 0$ depending only on $\boldsymbol{\theta}$ such that $\mathrm{TV}(P_{\boldsymbol{\theta}} \parallel P_{\boldsymbol{\theta}'}) \le C \cdot \|\boldsymbol{\theta}' - \boldsymbol{\theta}\|_2$ for all $\boldsymbol{\theta}' \in \Theta$ satisfying $\|\boldsymbol{\theta}' - \boldsymbol{\theta}\|_2 \le c$.*

*Proof.* Let $A$ be the log-partition function for the exponential family, which is infinitely-differentiable on $\Theta$ [Bro86], and let $\mathrm{D}_A(\boldsymbol{\theta}' \parallel \boldsymbol{\theta}) = A(\boldsymbol{\theta}') - A(\boldsymbol{\theta}) - \langle \nabla A(\boldsymbol{\theta}), \boldsymbol{\theta}' - \boldsymbol{\theta} \rangle$ be its corresponding Bregman divergence. By Pinsker's inequality and properties of Bregman divergences [BMD$^+$05, Appendix A], we have

$$2\,\mathrm{TV}(P_{\boldsymbol{\theta}} \parallel P_{\boldsymbol{\theta}'})^2 \le \mathrm{KL}(P_{\boldsymbol{\theta}} \parallel P_{\boldsymbol{\theta}'}) = \mathrm{D}_A(\boldsymbol{\theta}' \parallel \boldsymbol{\theta}) = A(\boldsymbol{\theta}') - A(\boldsymbol{\theta}) - \langle \nabla A(\boldsymbol{\theta}), \boldsymbol{\theta}' - \boldsymbol{\theta} \rangle.$$

Let $g(t) = A\left(t \cdot \boldsymbol{\theta}' + (1 - t) \cdot \boldsymbol{\theta}\right)$. By Taylor's theorem, there exists $\xi \in [0, 1]$ such that $A(\boldsymbol{\theta}') = A(\boldsymbol{\theta}) + \langle \nabla A(\boldsymbol{\theta}), \boldsymbol{\theta}' - \boldsymbol{\theta} \rangle + \frac{1}{2}(\boldsymbol{\theta}' - \boldsymbol{\theta})^{\mathsf{T}} \nabla^2 A\left(\xi \boldsymbol{\theta}' + (1 - \xi)\boldsymbol{\theta}\right)(\boldsymbol{\theta}' - \boldsymbol{\theta})$. We can therefore take any $C$ and $c$ such that the Hessian has eigenvalues bounded by $4C^2$ in a Euclidean ball of radius $c$ around $\boldsymbol{\theta}$, upon which we have $\mathrm{D}_A(\boldsymbol{\theta}' \parallel \boldsymbol{\theta}) \le 2C^2 \|\boldsymbol{\theta}' - \boldsymbol{\theta}\|_2^2$. $\qquad \square$

Now we argue that the method-of-moments estimator of [TGG94] for the Beta-Binomial distribution gives accurate parameter estimates of the Beta component parameters (i.e., $\alpha$ and $\beta$) provided a large enough sample size. The bound is given for the parameters corresponding to a single arm; applying the result for all arms $a \in \mathcal{A}$ with a union bound delivers the final sample complexity claim.

**Lemma F.2.** *Let $\hat{m}_1$ and $\hat{m}_2$ be empirical moments based on $N$ i.i.d. draws from a Beta-Binomial distribution with parameters $(\alpha, \beta, n)$ where $n \geq 2$. Let $(\hat{\alpha}, \hat{\beta})$ be the method-of-moments estimate of $(\alpha, \beta)$ obtained using*

$$\hat{\alpha} := \frac{n\hat{m}_1 - \hat{m}_2}{n(\frac{\hat{m}_2}{\hat{m}_1} - \hat{m}_1 - 1) + \hat{m}_1} \quad and \quad \hat{\beta} := \frac{(n - \hat{m}_1)(n - \frac{\hat{m}_2}{\hat{m}_1})}{n(\frac{\hat{m}_2}{\hat{m}_1} - \hat{m}_1 - 1) + \hat{m}_1}.$$

*There exists a positive constant $C > 0$ depending only on $(\alpha, \beta, n)$ such that for any $\epsilon > 0$ and $\delta \in (0, 1)$, if $N \geq C \log(1/\delta)/\epsilon^2$, then $\mathbb{P}(\max\{|\hat{\alpha} - \alpha|, |\hat{\beta} - \beta|\} \leq \epsilon) \geq 1 - \delta$.*

*Proof.* Let $X_1, \ldots, X_N$ denote an i.i.d. sample from the Beta-Binomial distribution with parameters $(\alpha, \beta, n)$, and let $m_1 := \mathbb{E}[X_1]$ and $m_2 := \mathbb{E}[X_1^2]$. First, by Hoeffding's inequality and union bounds, with probability at least $1 - \delta$, we have

$$|\hat{m}_1 - m_1| \leq n\sqrt{\frac{2\ln(4/\delta)}{N}} \quad and \quad |\hat{m}_2 - m_2| \leq n^2\sqrt{\frac{2\ln(4/\delta)}{N}},$$

where $\hat{m}_1 := \frac{1}{N}\sum_{i=1}^N X_i$ and $\hat{m}_2 := \frac{1}{N}\sum_{i=1}^N X_i^2$. Let us henceforth condition on this $1 - \delta$ probability event. Now, treating $\hat{\alpha}(\hat{m}_1, \hat{m}_2)$ and $\hat{\beta}(\hat{m}_1, \hat{m}_2)$ as functions of $(\hat{m}_1, \hat{m}_2)$, we have by Taylor's theorem that

$$\hat{\alpha}(\hat{m}_1, \hat{m}_2) = \hat{\alpha}(m_1, m_2) + \frac{\partial \hat{\alpha}}{\partial \hat{m}_1}(\tilde{m}_1) \cdot (\hat{m}_1 - m_1) + \frac{\partial \hat{\alpha}}{\partial \hat{m}_2}(\tilde{m}_2) \cdot (\hat{m}_2 - m_2)$$

$$\hat{\beta}(\hat{m}_1, \hat{m}_2) = \hat{\beta}(m_1, m_2) + \frac{\partial \hat{\beta}}{\partial \hat{m}_1}(\tilde{m}_1) \cdot (\hat{m}_1 - m_1) + \frac{\partial \hat{\beta}}{\partial \hat{m}_2}(\tilde{m}_2) \cdot (\hat{m}_2 - m_2)$$

where $(\tilde{m}_1, \tilde{m}_2) = (1 - \xi)(\hat{m}_1, \hat{m}_2) + \xi(m_1, m_2)$ for some $\xi \in [0, 1]$. It can be verified using properties of the Beta-Binomial distribution that $\hat{\alpha}(m_1, m_2) = \alpha$ and $\hat{\beta}(m_1, m_2) = \beta$. Moreover, since the functions $\hat{\alpha}(\hat{m}_1, \hat{m}_2)$ and $\hat{\beta}(\hat{m}_1, \hat{m}_2)$ are analytic, it follows that there is a Euclidean ball of radius (say) $c' > 0$ around $(m_1, m_2)$ on which the gradients of $\hat{\alpha}$ and $\hat{\beta}$ are uniformly bounded by (say) $C' > 0$ in Euclidean norm. Here, both $C'$ and $c'$ depend only on $m_1$ and $m_2$. So, as long as $\sqrt{(\hat{m}_1 - m_1)^2 + (\hat{m}_2 - m_2)^2} \leq c'$, we have

$$|\hat{\alpha}(\hat{m}_1, \hat{m}_2) - \alpha| \leq C\sqrt{(\hat{m}_1 - m_1)^2 + (\hat{m}_2 - m_2)^2}$$
$$|\hat{\beta}(\hat{m}_1, \hat{m}_2) - \beta| \leq C\sqrt{(\hat{m}_1 - m_1)^2 + (\hat{m}_2 - m_2)^2}$$

by Cauchy-Schwarz. The claim now follows by choosing $N \geq C \log(1/\delta)/\epsilon^2$ for some $C$ depending only on $C'$, $c'$, and $n$. $\square$

## F.2 Gaussian priors and Gaussian rewards

We directly bound the KL-divergence between two multivariate Gaussian distributions in terms of distances between their corresponding parameters.

**Lemma F.3** (Gaussian KL-divergence). *Let $P := \mathcal{N}(\boldsymbol{\nu}_\star, \boldsymbol{\Psi}_\star)$ and $\widehat{P} := \mathcal{N}(\widehat{\boldsymbol{\nu}}, \widehat{\boldsymbol{\Psi}})$ be multivariate Gaussian distributions in $\mathbb{R}^\mathcal{A}$. Then*

$$\mathrm{KL}(\widehat{P} \parallel P) = \frac{1}{2}\left\{\mathrm{tr}(\boldsymbol{\Psi}_\star^{-1/2}\widehat{\boldsymbol{\Psi}}\boldsymbol{\Psi}_\star^{-1/2} - I) - \ln\det(\boldsymbol{\Psi}_\star^{-1/2}\widehat{\boldsymbol{\Psi}}\boldsymbol{\Psi}_\star^{-1/2}) + \|\boldsymbol{\Psi}_\star^{-1/2}(\widehat{\boldsymbol{\nu}} - \boldsymbol{\nu}_\star)\|_2^2\right\}.$$

*Moreover, if*

$$\|\boldsymbol{\Psi}_\star^{-1/2}\widehat{\boldsymbol{\Psi}}\boldsymbol{\Psi}_\star^{-1/2} - I\|_2 \leq \frac{2}{3},$$

*then*

$$\mathrm{KL}(\widehat{P} \parallel P) \leq \frac{1}{2}\left\{|\mathcal{A}| \cdot \|\boldsymbol{\Psi}_\star^{-1/2}\widehat{\boldsymbol{\Psi}}\boldsymbol{\Psi}_\star^{-1/2} - I\|_2^2 + \|\boldsymbol{\Psi}_\star^{-1/2}(\widehat{\boldsymbol{\nu}} - \boldsymbol{\nu}_\star)\|_2^2\right\}.$$

*Proof.* The formula for the KL-divergence is standard. Now suppose that $\|\boldsymbol{\Psi}_\star^{-1/2}\widehat{\boldsymbol{\Psi}}\boldsymbol{\Psi}_\star^{-1/2} - I\|_2 \leq 2/3$. This means that all of the eigenvalues $\lambda_1, \ldots, \lambda_K$ of $\boldsymbol{\Psi}_\star^{-1/2}\widehat{\boldsymbol{\Psi}}\boldsymbol{\Psi}_\star^{-1/2}$ are contained in the interval $[1/3, 5/3]$. In this case, we have

$$
\begin{aligned}
\operatorname{tr}(\boldsymbol{\Psi}_\star^{-1/2}\widehat{\boldsymbol{\Psi}}\boldsymbol{\Psi}_\star^{-1/2} - I) - \ln\det(\boldsymbol{\Psi}_\star^{-1/2}\widehat{\boldsymbol{\Psi}}\boldsymbol{\Psi}_\star^{-1/2}) &= \sum_{i=1}^K \{\lambda_i - 1\} - \ln\prod_{i=1}^K \lambda_i \\
&= \sum_{i=1}^K \{\lambda_i - 1 - \ln\lambda_i\} \\
&\leq \sum_{i=1}^K (\lambda_i - 1)^2 \\
&\leq K \cdot \|\boldsymbol{\Psi}_\star^{-1/2}\widehat{\boldsymbol{\Psi}}\boldsymbol{\Psi}_\star^{-1/2} - I\|_2^2,
\end{aligned}
$$

where the first inequality uses the fact $\ln(1+x) \geq x - x^2$ for all $x \geq -2/3$. Plugging this inequality into the KL-divergence formula gives the claimed inequality. $\square$

Lemma F.3 and Pinsker's inequality imply that, to obtain an estimate of $\mathcal{N}(\boldsymbol{\nu}_\star, \boldsymbol{\Psi}_\star)$ that is $\varepsilon$-close in total variation distance, it suffices to obtain estimates $\widehat{\boldsymbol{\nu}}$ and $\widehat{\boldsymbol{\Psi}}$ such that

$$
\|\boldsymbol{\Psi}_\star^{-1/2}(\widehat{\boldsymbol{\nu}} - \boldsymbol{\nu}_\star)\|_2 \leq \varepsilon, \quad \|\boldsymbol{\Psi}_\star^{-1/2}(\widehat{\boldsymbol{\Psi}} - \boldsymbol{\Psi}_\star)\boldsymbol{\Psi}_\star^{-1/2}\|_2 \leq \frac{\varepsilon}{\sqrt{|\mathcal{A}|}}.
$$

Below, we give estimators $\widehat{\boldsymbol{\nu}}$ and $\widehat{\boldsymbol{\Psi}}$ that satisfy these inequalities with probability at least $1 - \delta$ provided that

$$
T \geq C' \cdot \frac{d \cdot (|\mathcal{A}|^4 + |\mathcal{A}|^3 \log(1/\delta))}{\varepsilon^2},
$$

where $d$ is defined in Lemma F.6, and $C'$ is an absolute constant. We note that if $\boldsymbol{\Psi}_\star$ is known and does not need to be estimated, then the requirement improves to

$$
T \geq C'' \cdot \frac{d_2 \cdot (|\mathcal{A}|^2 + |\mathcal{A}| \log(1/\delta))}{\varepsilon^2},
$$

where $d_2$ is defined in Lemma F.4, and $C''$ is another absolute constant.

**Mean estimation.** We first consider the estimate of $\boldsymbol{\nu}_\star$. To do so, we assume the first round in each of $T$ episodes is chosen uniformly at random from $\mathcal{A}$. In episode $t$:

1. let $\boldsymbol{\mu}_t \sim P = \mathcal{N}(\boldsymbol{\nu}_\star, \boldsymbol{\Psi}_\star)$ denote the mean reward vector;
2. let $a_t \sim \operatorname{Uniform}(\mathcal{A})$ be the action taken in the first round (independent of $\boldsymbol{\mu}_t$);
3. let $\mathbf{r}_t$ be the reward vector for the first round, so

$$
\mathbf{r}_t \mid (\boldsymbol{\mu}_t, a_t) \sim \mathcal{N}(\boldsymbol{\mu}_t, \sigma^2\mathbf{I}).
$$

The reward observed (and accrued) in the first round of episode $t$ is $\mathbf{r}_t^\top\mathbf{e}_{a_t}$. Our estimate of prior mean $\boldsymbol{\nu}_\star$ is

$$
\widehat{\boldsymbol{\nu}} := \frac{|\mathcal{A}|}{T}\sum_{t=1}^T (\mathbf{r}_t^\top\mathbf{e}_{a_t})\mathbf{e}_{a_t}. \tag{F.1}
$$

**Lemma F.4** (Gaussian mean estimation). *There exists a universal constant $C > 0$ such that the following holds. Consider any multivariate Gaussian distribution $P := \mathcal{N}(\boldsymbol{\nu}_\star, \boldsymbol{\Psi}_\star)$ in $\mathbb{R}^\mathcal{A}$. Let*

$$
(\boldsymbol{\mu}_1, a_1, \mathbf{r}_1), (\boldsymbol{\mu}_2, a_2, \mathbf{r}_2), \ldots, (\boldsymbol{\mu}_T, a_T, \mathbf{r}_T)
$$

*be $T$ iid random variables, with*

$$
\begin{aligned}
(\boldsymbol{\mu}_t, a_t) &\sim P \otimes \operatorname{Uniform}(\mathcal{A}), \\
\mathbf{r}_t \mid (\boldsymbol{\mu}_t, a_t) &\sim \mathcal{N}(\boldsymbol{\mu}_t, \sigma^2\mathbf{I});
\end{aligned}
$$

*and define $\widehat{\boldsymbol{\nu}}$ as in* (F.1). *For any $\delta \in (0,1)$, with probability at least $1 - \delta$,*

$$\|\boldsymbol{\Psi}_\star^{-1/2}(\widehat{\boldsymbol{\nu}} - \boldsymbol{\nu}_\star)\|_2 \leq C \left( \sqrt{\frac{d_2(|\mathcal{A}|^2 + |\mathcal{A}| \log(1/\delta))}{T}} + \frac{d_\infty(|\mathcal{A}|^2 + |\mathcal{A}| \log(1/\delta))}{T} \right),$$

*where*

$$d_2 := \lambda_{\max} \left( \boldsymbol{\Psi}_\star^{-1/2} \left( \mathrm{diag}(\boldsymbol{\Psi}_\star) + \sigma^2 \mathbf{I} + \mathrm{diag}(\boldsymbol{\nu}_\star)^2 \right) \boldsymbol{\Psi}_\star^{-1/2} \right),$$

$$d_\infty := \max_{a \in \mathcal{A}} \sqrt{(\boldsymbol{\Psi}_\star^{-1})_{a,a}((\boldsymbol{\Psi}_\star)_{a,a} + \sigma^2 + (\boldsymbol{\nu}_\star)_a^2)}.$$

*Proof.* First, since

$$\mathbb{E}\left[(\mathbf{r}_t^\intercal \mathbf{e}_{a_t})\mathbf{e}_{a_t}\right] = \frac{1}{|\mathcal{A}|} \sum_{a \in \mathcal{A}} \mathbb{E}\left[(\mathbf{r}_t^\intercal \mathbf{e}_{a_t})\mathbf{e}_{a_t} \mid a_t = a\right] = \frac{1}{|\mathcal{A}|} \mathbb{E}\left[\mathbf{r}_t\right] = \frac{1}{|\mathcal{A}|} \mathbb{E}\left[\boldsymbol{\mu}_t\right] = \frac{1}{|\mathcal{A}|} \boldsymbol{\nu}_\star,$$

it follows by linearity that $\mathbb{E}[\widehat{\boldsymbol{\nu}}] = \boldsymbol{\nu}_\star$. Next, we show that for any unit vector $\mathbf{u} \in S^{|\mathcal{A}|-1}$, the random variable

$$X_{\mathbf{u},t} := (\boldsymbol{\Psi}_\star^{-1/2}\mathbf{u})^\intercal \left( (\mathbf{r}_t^\intercal \mathbf{e}_{a_t})\mathbf{e}_{a_t} - \frac{1}{|\mathcal{A}|}\boldsymbol{\nu}_\star \right)$$

$$= (\boldsymbol{\Psi}_\star^{-1/2}\mathbf{u})^\intercal \mathbf{e}_{a_t}\mathbf{e}_{a_t}^\intercal(\mathbf{r}_t - \boldsymbol{\nu}_\star) + (\boldsymbol{\Psi}_\star^{-1/2}\mathbf{u})^\intercal \mathbf{e}_{a_t}\mathbf{e}_{a_t}^\intercal \boldsymbol{\nu}_\star - \frac{(\boldsymbol{\Psi}_\star^{-1/2}\mathbf{u})^\intercal \boldsymbol{\nu}_\star}{|\mathcal{A}|}$$

is $(4d_2/|\mathcal{A}|, 2d_\infty)$-subexponential. Consider $\lambda \in \mathbb{R}$ such that $|\lambda| \leq 1/(2d_\infty)$, and let $\mathbf{v} := \boldsymbol{\Psi}_\star^{-1/2}\mathbf{u}$. Then

$$\frac{\lambda^2 v_{a_t}^2}{2} \left( \mathbf{e}_{a_t}^\intercal \boldsymbol{\Psi}_\star \mathbf{e}_{a_t} + \sigma^2 \right) + \lambda v_{a_t} \mathbf{e}_{a_t}^\intercal \boldsymbol{\nu}_\star$$

$$= \frac{\lambda^2 (\mathbf{u}^\intercal \boldsymbol{\Psi}_\star^{-1/2}\mathbf{e}_{a_t})^2 (\mathbf{e}_{a_t}^\intercal \boldsymbol{\Psi}_\star \mathbf{e}_{a_t} + \sigma^2)}{2} + \lambda(\mathbf{u}^\intercal \boldsymbol{\Psi}_\star^{-1/2}\mathbf{e}_{a_t})\mathbf{e}_{a_t}^\intercal \boldsymbol{\nu}_\star$$

$$\leq \frac{\lambda^2 \mathbf{e}_{a_t}^\intercal \boldsymbol{\Psi}_\star^{-1}\mathbf{e}_{a_t}(\mathbf{e}_{a_t}^\intercal \boldsymbol{\Psi}_\star \mathbf{e}_{a_t} + \sigma^2)}{2} + |\lambda| \sqrt{\mathbf{e}_{a_t}^\intercal \boldsymbol{\Psi}_\star^{-1}\mathbf{e}_{a_t}} |\mathbf{e}_{a_t}^\intercal \boldsymbol{\nu}_\star|$$

$$\leq \frac{\lambda^2 d_\infty^2}{2} + |\lambda| d_\infty \leq 1$$

where the first inequality follows by Cauchy-Schwarz, the second inequality follows by definition of $d_\infty$, and the third inequality follows by assumption on $\lambda$. Further, observe that $\mathbf{r}_t$ has the same distribution as $\boldsymbol{\nu}_\star + \boldsymbol{\Psi}_\star^{1/2}\mathbf{x} + \sigma\mathbf{y}$, where $(\mathbf{x}, \mathbf{y}) \overset{\text{i.i.d.}}{\sim} \mathcal{N}(\mathbf{0}, \mathbf{I})^{\otimes 2}$, independent of $a_t$. Since $a_t$ and $(\mathbf{x}, \mathbf{y})$ are independent,

$$\mathbb{E}\exp(\lambda X_{\mathbf{u},t}) = \mathbb{E}\left[ \mathbb{E}\left[ \exp\left( \lambda v_{a_t}\mathbf{e}_{a_t}^\intercal(\boldsymbol{\Psi}_\star^{1/2}\mathbf{x} + \sigma\mathbf{y}) + \lambda\mathbf{v}^\intercal \mathbf{e}_{a_t}\mathbf{e}_{a_t}^\intercal \boldsymbol{\nu}_\star - \frac{\lambda\mathbf{v}^\intercal \boldsymbol{\nu}_\star}{|\mathcal{A}|} \right) \mid a_t \right] \right]$$

$$= \mathbb{E}\left[ \exp\left( \frac{\lambda^2 v_{a_t}^2 (\mathbf{e}_{a_t}^\intercal \boldsymbol{\Psi}_\star \mathbf{e}_{a_t} + \sigma^2)}{2} + \lambda\mathbf{v}^\intercal \mathbf{e}_{a_t}\mathbf{e}_{a_t}^\intercal \boldsymbol{\nu}_\star - \frac{\lambda\mathbf{v}^\intercal \boldsymbol{\nu}_\star}{|\mathcal{A}|} \right) \right]$$

$$\leq \exp\left( \sum_{a \in \mathcal{A}} \frac{2\lambda^2 v_a^2 (\mathbf{e}_a^\intercal \boldsymbol{\Psi}_\star \mathbf{e}_a + \sigma^2 + (\mathbf{e}_a^\intercal \boldsymbol{\nu}_\star)^2)}{|\mathcal{A}|} \right)$$

$$= \exp\left( \frac{2\lambda^2 \mathbf{u}^\intercal \boldsymbol{\Psi}_\star^{-1/2} \left( \mathrm{diag}(\boldsymbol{\Psi}_\star) + \sigma^2 \mathbf{I} + \mathrm{diag}(\boldsymbol{\nu}_\star)^2 \right) \boldsymbol{\Psi}_\star^{-1/2}\mathbf{u}}{|\mathcal{A}|} \right)$$

$$\leq \exp\left( \frac{2d_2\lambda^2}{|\mathcal{A}|} \right),$$

where we have used the moment generating function of standard Gaussian random variables, Lemma F.7 with the inequality from the previous display, and the definition of $d_2$. Thus $X_{\mathbf{u},t}$ is $(4d_2/|\mathcal{A}|, 2d_\infty)$-subexponential. By independence, $\sum_{t=1}^{T} X_{\mathbf{u},t}$ is $(4Td_2/|\mathcal{A}|, 2d_\infty)$-subexponential.

For any $\delta' \in (0, 1)$, a Bernstein inequality for subexponential random variables [Ver18, Theorem 2.8.1] gives, with probability at least $1 - \delta'$,

$$\sum_{t=1}^{T} X_{\mathbf{u},t} \leq \frac{C}{2} \left( \sqrt{\frac{T d_2 \log(1/\delta')}{|\mathcal{A}|}} + d_\infty \log(1/\delta') \right).$$

Combining with a union bound over all choices of $\mathbf{u}$ from a $(1/2)$-net $N$ of $S^{|\mathcal{A}|-1}$ shows that with probability at least $1 - |N|\delta'$, the inequality in the previous display holds for all $\mathbf{u} \in N$. A standard volume argument shows that we can take $|N| \leq 5^{|\mathcal{A}|}$ [Ver18, Corollary 4.2.13]. Therefore, the claim follows by choosing $\delta' := \delta/5^{|\mathcal{A}|}$ and observing that [Ver18, Exercise 4.4.2]

$$\|\boldsymbol{\Psi}_\star^{-1/2}(\widehat{\boldsymbol{\nu}} - \boldsymbol{\nu}_\star)\|_2 = \frac{|\mathcal{A}|}{T} \sup_{\mathbf{u} \in S^{|\mathcal{A}|-1}} \sum_{t=1}^{T} X_{\mathbf{u},t} \leq \frac{2|\mathcal{A}|}{T} \sup_{\mathbf{u} \in N} \sum_{t=1}^{T} X_{\mathbf{u},t}. \qquad \square$$

**Covariance estimation.** We now consider the estimate of $\boldsymbol{\Psi}_\star$. To do so, we first consider the case where $\boldsymbol{\nu}_\star$ is already known, so only $\boldsymbol{\Psi}_\star$ needs to be estimated. We assume the first two rounds in each of $T$ episodes are chosen independently and uniformly at random from $\mathcal{A}$. In episode $t$:

1. let $\boldsymbol{\mu}_t \sim P = \mathcal{N}(\boldsymbol{\nu}_\star, \boldsymbol{\Psi}_\star)$ denote the mean reward vector;
2. let $a_t, b_t \overset{\text{i.i.d.}}{\sim} \text{Uniform}(\mathcal{A})$ be the actions taken in the first two rounds (independent of $\boldsymbol{\mu}_t$);
3. let $\mathbf{r}_t$ and $\mathbf{s}_t$ be the reward vectors for the first two rounds, so

$$\mathbf{r}_t, \mathbf{s}_t \mid (\boldsymbol{\mu}_t, a_t, b_t) \overset{\text{i.i.d.}}{\sim} \mathcal{N}(\boldsymbol{\mu}_t, \sigma^2 \mathbf{I}).$$

The rewards observed (and accrued) in the first two rounds of episode $t$ are $\mathbf{r}_t^\mathsf{T} \mathbf{e}_{a_t}$ and $\mathbf{s}_t^\mathsf{T} \mathbf{e}_{b_t}$. Our estimate of prior covariance $\boldsymbol{\Psi}_\star$ is

$$\widehat{\boldsymbol{\Psi}} := \frac{|\mathcal{A}|^2}{2T} \sum_{t=1}^{T} (\mathbf{r}_t - \boldsymbol{\nu}_\star)^\mathsf{T} \mathbf{e}_{a_t} (\mathbf{s}_t - \boldsymbol{\nu}_\star)^\mathsf{T} \mathbf{e}_{b_t} \left( \mathbf{e}_{a_t} \mathbf{e}_{b_t}^\mathsf{T} + \mathbf{e}_{b_t} \mathbf{e}_{a_t}^\mathsf{T} \right). \tag{F.2}$$

**Lemma F.5** (Gaussian covariance estimation with known mean). *There exists a universal constant $C > 0$ such that the following holds. Consider any multivariate Gaussian distribution $P := \mathcal{N}(\boldsymbol{\nu}_\star, \boldsymbol{\Psi}_\star)$ in $\mathbb{R}^{\mathcal{A}}$. Let*

$$(\boldsymbol{\mu}_1, a_1, b_1, \mathbf{r}_1, \mathbf{s}_1), (\boldsymbol{\mu}_2, a_2, b_2, \mathbf{r}_2, \mathbf{s}_2), \ldots, (\boldsymbol{\mu}_T, a_T, b_T, \mathbf{r}_T, \mathbf{s}_T)$$

*be $T$ iid random variables, with*

$$(\boldsymbol{\mu}_t, a_t, b_t) \sim P \otimes \text{Uniform}(\mathcal{A}) \otimes \text{Uniform}(\mathcal{A}),$$
$$(\mathbf{r}_t, \mathbf{s}_t) \mid (\boldsymbol{\mu}_t, a_t, b_t) \sim \mathcal{N}(\boldsymbol{\mu}_t, \sigma^2 \mathbf{I}) \otimes \mathcal{N}(\boldsymbol{\mu}_t, \sigma^2 \mathbf{I});$$

*and define $\widehat{\boldsymbol{\Psi}}$ as in (F.2). For any $\delta \in (0, 1)$, with probability at least $1 - \delta$,*

$$\|\boldsymbol{\Psi}_\star^{-1/2}(\widehat{\boldsymbol{\Psi}} - \boldsymbol{\Psi}_\star)\boldsymbol{\Psi}_\star^{-1/2}\|_2 \leq C\sqrt{d} \left( \sqrt{\frac{|\mathcal{A}|^3 + |\mathcal{A}|^2 \log(1/\delta)}{T}} + \frac{|\mathcal{A}|^3 + |\mathcal{A}|^2 \log(1/\delta)}{T} \right),$$

*where*

$$d := \frac{\sigma^4 + \max_{a \in \mathcal{A}} (\boldsymbol{\Psi}_\star)_{a,a}^2}{\lambda_{\min}(\boldsymbol{\Psi}_\star)^2}.$$

*Proof.* The proof is very similar to that of Lemma F.4. We first observe that

$$\mathbb{E}\left[ \frac{1}{2} (\mathbf{r}_t - \boldsymbol{\nu}_\star)^\mathsf{T} \mathbf{e}_{a_t} (\mathbf{s}_t - \boldsymbol{\nu}_\star)^\mathsf{T} \mathbf{e}_{b_t} \left( \mathbf{e}_{a_t} \mathbf{e}_{b_t}^\mathsf{T} + \mathbf{e}_{b_t} \mathbf{e}_{a_t}^\mathsf{T} \right) \right] = \frac{1}{|\mathcal{A}|^2} \boldsymbol{\Psi}_\star.$$

We claim that for any unit vector $\mathbf{u} \in S^{|\mathcal{A}|-1}$,

$$X_{\mathbf{u},t} := (\boldsymbol{\Psi}_\star^{-1/2} \mathbf{u})^\mathsf{T} \left( \frac{1}{2} (\mathbf{r}_t - \boldsymbol{\nu}_\star)^\mathsf{T} \mathbf{e}_{a_t} (\mathbf{s}_t - \boldsymbol{\nu}_\star)^\mathsf{T} \mathbf{e}_{b_t} \left( \mathbf{e}_{a_t} \mathbf{e}_{b_t}^\mathsf{T} + \mathbf{e}_{b_t} \mathbf{e}_{a_t}^\mathsf{T} \right) - \frac{1}{|\mathcal{A}|^2} \boldsymbol{\Psi}_\star \right) (\boldsymbol{\Psi}_\star^{-1/2} \mathbf{u})$$

is $(v, c)$-subexponential with $v = O(d/|\mathcal{A}|^2)$ and $c = O(\sqrt{d})$. We defer this argument until the end. By independence, $\sum_{t=1}^{T} X_{\mathbf{u},t}$ is $(Tv, c)$-subexponential. For any $\delta' \in (0, 1)$, a Bernstein inequality for subexponential random variables [Ver18, Theorem 2.8.1] gives, with probability at least $1 - \delta'$,

$$\left| \sum_{t=1}^{T} X_{\mathbf{u},t} \right| \leq \frac{C}{2} \left( \sqrt{\frac{Td \log(1/\delta')}{|\mathcal{A}|^2}} + \sqrt{d} \log(1/\delta') \right).$$

Combining with a union bound over all choices of $\mathbf{u}$ from a $(1/4)$-net $N$ of $S^{|\mathcal{A}|-1}$ shows that with probability at least $1 - |N|\delta'$, the inequality in the previous display holds for all $\mathbf{u} \in N$. A standard volume argument shows that we can take $|N| \leq 9^{|\mathcal{A}|}$ [Ver18, Corollary 4.2.13]. Therefore, the claim follows by choosing $\delta' := \delta/9^{|\mathcal{A}|}$ and observing that [Ver18, Exercise 4.4.3(b)]

$$\|\boldsymbol{\Psi}_\star^{-1/2}(\widehat{\boldsymbol{\Psi}} - \boldsymbol{\Psi}_\star)\boldsymbol{\Psi}_\star^{-1/2}\|_2 = \frac{|\mathcal{A}|^2}{T} \sup_{\mathbf{u} \in S^{|\mathcal{A}|-1}} \left| \sum_{t=1}^{T} X_{\mathbf{u},t} \right| \leq \frac{2|\mathcal{A}|^2}{T} \sup_{\mathbf{u} \in N} \left| \sum_{t=1}^{T} X_{\mathbf{u},t} \right|$$

It remains to show that $X_{\mathbf{u},t}$ is $(v, c)$-subexponential with $v = O(d/|\mathcal{A}|^2)$ and $c = O(\sqrt{d})$. Observe that $(\mathbf{r}_t, \mathbf{s}_t, a_t, b_t)$ has the same joint distribution as $(\boldsymbol{\nu}_\star + \boldsymbol{\Psi}_\star^{1/2}\mathbf{x} + \sigma\mathbf{y}, \boldsymbol{\nu}_\star + \boldsymbol{\Psi}_\star^{1/2}\mathbf{x} + \sigma\mathbf{z}, a_t, b_t)$, where $\mathbf{x}, \mathbf{y}, \mathbf{z}$ are i.i.d. $\mathcal{N}(\mathbf{0}, \mathbf{I})$ random vectors in $\mathbb{R}^{\mathcal{A}}$, independent of $(a_t, b_t)$. Let $\mathbf{v} := \boldsymbol{\Psi}_\star^{-1/2}\mathbf{u}$ and $\mathbf{w} := \boldsymbol{\Psi}_\star^{1/2}\mathbf{x}$, so

$$X_{\mathbf{u},t} = (\boldsymbol{\Psi}_\star^{-1/2}\mathbf{u})^\mathsf{T} \left( \frac{1}{2}(\mathbf{r}_t - \boldsymbol{\nu}_\star)^\mathsf{T}\mathbf{e}_{a_t}(\mathbf{s}_t - \boldsymbol{\nu}_\star)^\mathsf{T}\mathbf{e}_{b_t} \left( \mathbf{e}_{a_t}\mathbf{e}_{b_t}^\mathsf{T} + \mathbf{e}_{b_t}\mathbf{e}_{a_t}^\mathsf{T} \right) - \frac{1}{|\mathcal{A}|^2}\boldsymbol{\Psi}_\star \right) (\boldsymbol{\Psi}_\star^{-1/2}\mathbf{u})$$

$$\overset{\text{dist}}{=} (\boldsymbol{\Psi}_\star^{-1/2}\mathbf{u})^\mathsf{T} \left( \frac{1}{2}(\mathbf{w} + \sigma\mathbf{y})^\mathsf{T}\mathbf{e}_{a_t}(\mathbf{w} + \sigma\mathbf{z})^\mathsf{T}\mathbf{e}_{b_t} \left( \mathbf{e}_{a_t}\mathbf{e}_{b_t}^\mathsf{T} + \mathbf{e}_{b_t}\mathbf{e}_{a_t}^\mathsf{T} \right) - \frac{1}{|\mathcal{A}|^2}\boldsymbol{\Psi}_\star \right) (\boldsymbol{\Psi}_\star^{-1/2}\mathbf{u})$$

$$= \left( v_{a_t}v_{b_t}(w_{a_t} + \sigma y_{a_t})(w_{b_t} + \sigma z_{b_t}) - \frac{1}{|\mathcal{A}|^2} \right).$$

Now we fix $\lambda \in \mathbb{R}$ such that $|\lambda| \leq 1/(C\sqrt{d})$ for some sufficiently large constant $C > 0$, and bound the moment generating function of $X_{\mathbf{u},t}$ at $\lambda$. To do so, we use the above characterization of the distribution of $X_{\mathbf{u},t}$ in terms of the independent Gaussian random vectors. First, taking expectation only with respect to $(\mathbf{y}, \mathbf{z})$ (i.e., conditional on $a_t, b_t, \mathbf{x}$):

$$\mathbb{E}\left[ \exp(\lambda X_{\mathbf{u},t}) \right]$$

$$= \mathbb{E}\left[ \mathbb{E}\left[ \exp\left( \lambda \left( v_{a_t}v_{b_t}(w_{a_t} + \sigma y_{a_t})(w_{b_t} + \sigma z_{b_t}) - \frac{1}{|\mathcal{A}|^2} \right) \right) \mid a_t, b_t, \mathbf{x} \right] \right]$$

$$= \mathbb{E}\left[ \exp\left( \eta w_{a_t}w_{b_t} + \frac{\eta^2\sigma^2 w_{a_t}^2 + \eta^2\sigma^2 w_{b_t}^2 + \eta^3\sigma^4 w_{a_t}w_{b_t}}{2(1 - \eta^2\sigma^4)} + \frac{1}{2}\ln\frac{1}{1 - \eta^2\sigma^4} - \frac{\lambda}{|\mathcal{A}|^2} \right) \right]$$

$$\leq \mathbb{E}\left[ \exp\left( \eta w_{a_t}w_{b_t} + \frac{\eta^2\sigma^2 w_{a_t}^2 + \eta^2\sigma^2 w_{b_t}^2 + \eta^3\sigma^4 w_{a_t}w_{b_t}}{2(1 - \eta^2\sigma^4)} + \eta^2\sigma^4 - \frac{\lambda}{|\mathcal{A}|^2} \right) \right]$$

where $\eta := \lambda v_{a_t}v_{b_t}$ satisfies $\eta^2\sigma^4 \leq 1/2$ (due to the assumption on $\lambda$). Next, we note that

$$\eta w_{a_t}w_{b_t} + \frac{\eta^2\sigma^2 w_{a_t}^2 + \eta^2\sigma^2 w_{b_t}^2 + \eta^3\sigma^4 w_{a_t}w_{b_t}}{2(1 - \eta^2\sigma^4)} = \mathbf{x}^\mathsf{T}\mathbf{A}\mathbf{x}$$

where $\mathbf{A}$ is the random symmetric matrix defined by

$$\mathbf{A} := \frac{\eta}{2(1 - \eta^2\sigma^4)}\boldsymbol{\Psi}_\star^{1/2} \left( (1 - \eta^2\sigma^4/2)(\mathbf{e}_{a_t}\mathbf{e}_{b_t}^\mathsf{T} + \mathbf{e}_{b_t}\mathbf{e}_{a_t}^\mathsf{T}) + \eta\sigma^2(\mathbf{e}_{a_t}\mathbf{e}_{a_t}^\mathsf{T} + \mathbf{e}_{b_t}\mathbf{e}_{b_t}^\mathsf{T}) \right) \boldsymbol{\Psi}_\star^{1/2}$$

$$= \eta \left( 1 + \frac{\eta^2\sigma^4}{2(1 - \eta^2\sigma^4)} \right) \boldsymbol{\Psi}_\star^{1/2} \left( \frac{1}{2}(\mathbf{e}_{a_t}\mathbf{e}_{b_t}^\mathsf{T} + \mathbf{e}_{b_t}\mathbf{e}_{a_t}^\mathsf{T}) \right) \boldsymbol{\Psi}_\star^{1/2}$$

$$+ \frac{\eta^2\sigma^2}{1 - \eta^2\sigma^4}\boldsymbol{\Psi}_\star^{1/2} \left( \frac{1}{2}(\mathbf{e}_{a_t}\mathbf{e}_{a_t}^\mathsf{T} + \mathbf{e}_{b_t}\mathbf{e}_{b_t}^\mathsf{T}) \right)$$

(where the randomness comes from $a_t, b_t$). Since $\mathbf{A}$ is symmetric, it has real eigenvalues $\lambda_1, \lambda_2, \ldots, \lambda_{|\mathcal{A}|}$. We shall ensure via the assumption on $\lambda$ that $\|\mathbf{A}\|_2 \leq 1/3$, which implies that $|\lambda_i| \leq 1/3$ for all $i$. The rotational invariance of $\mathcal{N}(\mathbf{0}, \mathbf{I})$ implies that the distribution of $\mathbf{x}^\mathsf{T}\mathbf{A}\mathbf{x}$ (conditional on $a_t, b_t$) is the same as that of $\sum_i \lambda_i t_i$, where $t_1, t_2, \ldots, t_{|\mathcal{A}|}$ are i.i.d. $\chi^2(1)$ random variables. This implies that

$$\mathbb{E}\left[\exp\left(\mathbf{x}^\mathsf{T}\mathbf{A}\mathbf{x}\right) \mid a_t, b_t\right] = \exp\left(\frac{1}{2}\sum_i \ln \frac{1}{1 - 2\lambda_i}\right)$$

$$\leq \exp\left(\sum_i \lambda_i + 2\lambda_i^2\right) = \exp\left(\mathrm{tr}(\mathbf{A}) + 2\|\mathbf{A}\|_{\mathrm{F}}^2\right)$$

where the inequality uses the bound $|\lambda_i| \leq 1/3$. We expand $\mathrm{tr}(\mathbf{A})$ to reveal its dependence on $a_t, b_t$:

$$\mathrm{tr}(\mathbf{A}) = \eta(\mathbf{\Psi}_\star)_{a_t, b_t} + \frac{\eta^3\sigma^4(\mathbf{\Psi}_\star)_{a_t, b_t} + \eta^2\sigma^2((\mathbf{\Psi}_\star)_{a_t, a_t} + (\mathbf{\Psi}_\star)_{b_t, b_t})}{2(1 - \eta^2\sigma^4)}.$$

And we bound $\|\mathbf{A}\|_{\mathrm{F}}^2$ as follows:

$$\|\mathbf{A}\|_{\mathrm{F}}^2 \leq 2\eta^2\left(1 + \frac{\eta^2\sigma^4}{2(1 - \eta^2\sigma^4)}\right)^2 \left\|\mathbf{\Psi}_\star^{1/2}\left(\frac{1}{2}(\mathbf{e}_{a_t}\mathbf{e}_{b_t}^\mathsf{T} + \mathbf{e}_{b_t}\mathbf{e}_{a_t}^\mathsf{T})\right)\mathbf{\Psi}_\star^{1/2}\right\|_{\mathrm{F}}^2$$

$$+ 2\eta^2\left(\frac{\eta\sigma^2}{1 - \eta^2\sigma^4}\right)^2 \left\|\mathbf{\Psi}_\star^{1/2}\left(\frac{1}{2}(\mathbf{e}_{a_t}\mathbf{e}_{a_t}^\mathsf{T} + \mathbf{e}_{b_t}\mathbf{e}_{b_t}^\mathsf{T})\right)\mathbf{\Psi}_\star^{1/2}\right\|_{\mathrm{F}}^2$$

$$= \eta^2\left(1 + \frac{\eta^2\sigma^4}{2(1 - \eta^2\sigma^4)}\right)^2 \left((\mathbf{\Psi}_\star)_{a_t, b_t}^2 + (\mathbf{\Psi}_\star)_{a_t, a_t}(\mathbf{\Psi}_\star)_{b_t, b_t}\right)$$

$$+ \frac{\eta^2}{2}\left(\frac{\eta\sigma^2}{1 - \eta^2\sigma^4}\right)^2 \left((\mathbf{\Psi}_\star)_{a_t, a_t}^2 + 2(\mathbf{\Psi}_\star)_{a_t, b_t}^2 + (\mathbf{\Psi}_\star)_{b_t, b_t}^2\right)$$

$$\leq 4\eta^2\left(1 + \frac{\eta^2\sigma^4}{2(1 - \eta^2\sigma^4)}\right)^2 \left((\mathbf{\Psi}_\star)_{a_t, a_t}^2 + (\mathbf{\Psi}_\star)_{b_t, b_t}^2\right)$$

$$+ \eta^2\left(\frac{\eta\sigma^2}{1 - \eta^2\sigma^4}\right)^2 \left((\mathbf{\Psi}_\star)_{a_t, a_t}^2 + (\mathbf{\Psi}_\star)_{b_t, b_t}^2\right)$$

$$\leq 11\eta^2\left((\mathbf{\Psi}_\star)_{a_t, a_t}^2 + (\mathbf{\Psi}_\star)_{b_t, b_t}^2\right).$$

Above the first inequality follows by the triangle inequality and the fact $(a + b)^2 \leq 2(a^2 + b^2)$ for $a, b \geq 0$; the second inequality uses Cauchy-Schwarz and the AM/GM inequality; the third inequality uses the assumption $\eta^2\sigma^4 \leq 1/2$. Thus, we have shown that

$$\mathbb{E}\left[\exp(\lambda X_{\mathbf{u}, t})\right] \leq \mathbb{E}\left[\exp\left(\mathrm{tr}(\mathbf{A}) + 2\|\mathbf{A}\|_{\mathrm{F}}^2 + \eta^2\sigma^4 - \frac{\lambda}{|\mathcal{A}|^2}\right)\right]$$

$$\leq \mathbb{E}\left[\exp\left(\eta(\mathbf{\Psi}_\star)_{a_t, b_t} + \frac{\eta^3\sigma^4(\mathbf{\Psi}_\star)_{a_t, b_t} + \eta^2\sigma^2((\mathbf{\Psi}_\star)_{a_t, a_t} + (\mathbf{\Psi}_\star)_{b_t, b_t})}{2(1 - \eta^2\sigma^4)}\right.\right.$$

$$\left.\left. + 11\eta^2\left((\mathbf{\Psi}_\star)_{a_t, a_t}^2 + (\mathbf{\Psi}_\star)_{b_t, b_t}^2\right) - \frac{\lambda}{|\mathcal{A}|^2}\right)\right].$$

Define

$$\alpha_{a,b} := \beta_{a,b} + \gamma_{a,b}$$

$$\beta_{a,b} := \lambda v_a v_b (\mathbf{\Psi}_\star)_{a,b}$$

$$\gamma_{a,b} := \frac{\lambda^3 v_a^3 v_b^3 \sigma^4 (\mathbf{\Psi}_\star)_{a,b} + \lambda^2 v_a^2 v_b^2 \sigma^2 ((\mathbf{\Psi}_\star)_{a,a} + (\mathbf{\Psi}_\star)_{b,b})}{2(1 - \lambda^2 v_a^2 v_b^2 \sigma^4)}$$

$$+ \lambda^2 v_a^2 v_b^2 (11((\mathbf{\Psi}_\star)_{a,a}^2 + (\mathbf{\Psi}_\star)_{b,b}^2) + \sigma^4).$$

Observe that

$$\frac{1}{|\mathcal{A}|^2} \sum_{a,b \in \mathcal{A}} \beta_{a,b} = 1.$$

The assumptions $\lambda$ ensure that $|\beta_{a,b}| + |\gamma_{a,b}| \leq 1$, so we can apply Lemma F.7 to bound the final expression in the previous display to obtain the inequality

$$\mathbb{E}\left[\exp(\lambda X_{\mathbf{u},t})\right] \leq \exp\left(\frac{1}{|\mathcal{A}|^2}\sum_{a,b\in\mathcal{A}}\left(\alpha_{a,b} + \alpha_{a,b}^2\right) - \frac{\lambda}{|\mathcal{A}|^2}\right)$$

$$\leq \exp\left(\frac{1}{|\mathcal{A}|^2}\sum_{a,b\in\mathcal{A}}\left(4\gamma_{a,b} + \beta_{a,b}^2\right)\right)$$

$$\leq \exp\left(\frac{8\|\mathbf{v}\|_2^4(\sigma^4 + \|\operatorname{diag}(\boldsymbol{\Psi}_\star)\|_2^2)}{|\mathcal{A}|^2}\left[\frac{|\lambda|\|\mathbf{v}\|_\infty^2\sigma^2 + 2}{4(1 - \lambda^2\|\mathbf{v}\|_\infty^4\sigma^4)} + 23\right]\frac{\lambda^2}{2}\right)$$

$$\leq \exp\left(\frac{200\|\mathbf{v}\|_2^4(\sigma^4 + \|\operatorname{diag}(\boldsymbol{\Psi}_\star)\|_2^2)}{|\mathcal{A}|^2}\cdot\frac{\lambda^2}{2}\right),$$

where the inequalities use the bounds on $|\beta_{a,b}|$ and $|\gamma_{a,b}|$, and the additional bound $|\lambda|\|\mathbf{v}\|_\infty^2(\sigma^2 + \|\operatorname{diag}(\boldsymbol{\Psi}_\star)\|_2) \leq 1/10$ which is implied by the assumption on $\lambda$. The final bound is $\exp(C(d/|\mathcal{A}|^2)\cdot \lambda^2/2)$ for a sufficiently large absolute constant $C > 0$. $\qquad\square$

**Covariance estimation, redux.** Now we consider the case where both $\boldsymbol{\nu}_\star$ and $\boldsymbol{\Psi}_\star$ are unknown and need to be estimated. A standard approach to estimating $\boldsymbol{\Psi}_\star$ is to simply estimate the second moment of $\boldsymbol{\mu}_t$ (instead of its covariance), and then to subtract $\widehat{\boldsymbol{\nu}}\widehat{\boldsymbol{\nu}}^\mathsf{T}$ using some estimate $\widehat{\boldsymbol{\nu}}$ of $\boldsymbol{\nu}_\star$. However, the quality of our estimate of $\boldsymbol{\nu}_\star$ (described above) depends on properties of $\boldsymbol{\nu}_\star$ itself, which should not be necessary. Below, we instead analyze an estimator of $\boldsymbol{\Psi}_\star$ based on differences, essentially leveraging the fact that the variance of a random variable $X$ is half the expected squared difference between $X$ and an independent copy of itself.

We assume the first two rounds in each of $2T$ episodes are chosen independently and uniformly at random from $\mathcal{A}$. However, we use the same two chosen actions in two consecutive episodes. That is, in episodes $2t - 1$ and $2t$:

1. let $\boldsymbol{\mu}_t, \tilde{\boldsymbol{\mu}}_t \overset{\text{i.i.d.}}{\sim} P = \mathcal{N}(\boldsymbol{\nu}_\star, \boldsymbol{\Psi}_\star)$ denote the mean reward vectors for episodes $2t - 1$ and $2t$;
2. let $a_t, b_t \overset{\text{i.i.d.}}{\sim} \operatorname{Uniform}(\mathcal{A})$ be the actions taken in the first two rounds (independent of $\boldsymbol{\mu}_t, \tilde{\boldsymbol{\mu}}_t$);
3. let $\mathbf{r}_t$ and $\mathbf{s}_t$ be the reward vectors for the first two rounds of episode $2t - 1$, and let $\tilde{\mathbf{r}}_t$ and $\tilde{\mathbf{s}}_t$ be the reward vectors for the first two rounds of episode $2t$, so
$$(\mathbf{r}_t, \mathbf{s}_t, \tilde{\mathbf{r}}_t, \tilde{\mathbf{s}}_t) \mid (\boldsymbol{\mu}_t, \tilde{\boldsymbol{\mu}}_t, a_t, b_t) \sim \mathcal{N}(\boldsymbol{\mu}_t, \sigma^2\mathbf{I}) \otimes \mathcal{N}(\boldsymbol{\mu}_t, \sigma^2\mathbf{I}) \otimes \mathcal{N}(\tilde{\boldsymbol{\mu}}_t, \sigma^2\mathbf{I}) \otimes \mathcal{N}(\tilde{\boldsymbol{\mu}}_t, \sigma^2\mathbf{I}).$$

The rewards observed (and accrued) in the first two rounds of episode $2t - 1$ are $\mathbf{r}_t^\mathsf{T}\mathbf{e}_{a_t}$ and $\mathbf{s}_t^\mathsf{T}\mathbf{e}_{b_t}$, and the rewards observed (and accrued) in the first two rounds of episode $2t$ are $\tilde{\mathbf{r}}_t^\mathsf{T}\mathbf{e}_{a_t}$ and $\tilde{\mathbf{s}}_t^\mathsf{T}\mathbf{e}_{b_t}$. Our estimate of prior covariance $\boldsymbol{\Psi}_\star$ is

$$\widehat{\boldsymbol{\Psi}} := \frac{|\mathcal{A}|^2}{4T}\sum_{t=1}^{T}(\mathbf{r}_t - \tilde{\mathbf{r}}_t)^\mathsf{T}\mathbf{e}_{a_t}(\mathbf{s}_t - \tilde{\mathbf{s}}_t)^\mathsf{T}\mathbf{e}_{b_t}\left(\mathbf{e}_{a_t}\mathbf{e}_{b_t}^\mathsf{T} + \mathbf{e}_{b_t}\mathbf{e}_{a_t}^\mathsf{T}\right). \tag{F.3}$$

**Lemma F.6** (Gaussian covariance estimation with unknown mean)**.** *There exists a universal constant $C > 0$ such that the following holds. Consider any multivariate Gaussian distribution $P := \mathcal{N}(\boldsymbol{\nu}_\star, \boldsymbol{\Psi}_\star)$ in $\mathbb{R}^\mathcal{A}$. Let*

$$(\boldsymbol{\mu}_1, \tilde{\boldsymbol{\mu}}_1, a_1, b_1, \mathbf{r}_1, \mathbf{s}_1, \tilde{\mathbf{r}}_1, \tilde{\mathbf{s}}_1), (\boldsymbol{\mu}_2, \tilde{\boldsymbol{\mu}}_2, a_2, b_2, \mathbf{r}_2, \mathbf{s}_2, \tilde{\mathbf{r}}_2, \tilde{\mathbf{s}}_2), \ldots, (\boldsymbol{\mu}_T, \tilde{\boldsymbol{\mu}}_T, a_T, b_T, \mathbf{r}_T, \mathbf{s}_T, \tilde{\mathbf{r}}_T, \tilde{\mathbf{s}}_T),$$

*be $T$ iid random variables, with*

$$(\boldsymbol{\mu}_t, \tilde{\boldsymbol{\mu}}_t, a_t, b_t) \sim P \otimes P \otimes \operatorname{Uniform}(\mathcal{A}) \otimes \operatorname{Uniform}(\mathcal{A}),$$

$$(\mathbf{r}_t, \mathbf{s}_t, \tilde{\mathbf{r}}_t, \tilde{\mathbf{s}}_t) \mid (\boldsymbol{\mu}_t, \tilde{\boldsymbol{\mu}}_t, a_t, b_t) \sim \mathcal{N}(\boldsymbol{\mu}_t, \sigma^2\mathbf{I}) \otimes \mathcal{N}(\boldsymbol{\mu}_t, \sigma^2\mathbf{I}) \otimes \mathcal{N}(\tilde{\boldsymbol{\mu}}_t, \sigma^2\mathbf{I}) \otimes \mathcal{N}(\tilde{\boldsymbol{\mu}}_t, \sigma^2\mathbf{I});$$

*and define $\widehat{\boldsymbol{\Psi}}$ as in* (F.3)*. For any $\delta \in (0, 1)$, with probability at least $1 - \delta$,*

$$\|\boldsymbol{\Psi}_\star^{-1/2}(\widehat{\boldsymbol{\Psi}} - \boldsymbol{\Psi}_\star)\boldsymbol{\Psi}_\star^{-1/2}\|_2 \leq C\sqrt{d}\left(\sqrt{\frac{|\mathcal{A}|^3 + |\mathcal{A}|^2\log(1/\delta)}{T}} + \frac{|\mathcal{A}|^3 + |\mathcal{A}|^2\log(1/\delta)}{T}\right),$$

*where*

$$d := \frac{\sigma^4 + \max_{a \in \mathcal{A}} (\mathbf{\Psi}_\star)^2_{a,a}}{\lambda_{\min}(\mathbf{\Psi}_\star)^2}.$$

We omit the proof of Lemma F.6 since it is completely analogous to that of Lemma F.5.

The following lemma is used to bound the exponential moment of a discrete real-valued random variable.

**Lemma F.7.** *Let $Y$ be a random variable supported on $\{\alpha_1, \ldots, \alpha_K\} \subset \mathbb{R}$ with $\alpha_i \le 1$ and $p_i := \mathbb{P}(Y = \alpha_i)$ for all $i$. Then*

$$\mathbb{E}[\exp(Y)] \le \exp\left( \sum_{i=1}^{K} p_i \alpha_i + p_i \alpha_i^2 \right).$$

*Proof.* Since $e^t \le 1 + t + t^2$ for all $t \le 1$, we have

$$\mathbb{E}[\exp(Y)] \le \mathbb{E}[1 + Y + Y^2] = 1 + \sum_{i=1}^{K} q_i \alpha_i + \sum_{i=1}^{K} q_i \alpha_i^2.$$

The claim now follows since $1 + t \le e^t$ for all $t \in \mathbb{R}$. $\qquad\square$