# OpenReview forum: "Bayesian decision-making under misspecified priors with applications to meta-learning"
_NeurIPS.cc/2021/Conference — NeurIPS 2021 Spotlight_

### Official Review · Reviewer_BAyg · 2021-07-07

**Rating:** 5
**Confidence:** 3

**Summary:**

This article presents some very nice results on posterior-sampling-based bandit algorithms with mis-specified priors. The algorithms are those like Thompson sampling, in which the actions selected are functions of draws from the posterior distribution to date (an approximation of knowledge gradient is also analysed as part of a wider class of posterior-sample-based decision algorithms, and this is shown to be an improvement on TS). Results apply both to standard bandit settings and to meta-learning settings in which a series of short episodes are attempted. Theory and experiments are impressive.

**Limitations And Societal Impact:**

Yes

**Main Review:**

The article presents a nice general framework for learning, and a method for analysing performance when the prior used by the learning algorithm is not the same as the distribution that the environment "uses". There are philosophical considerations as to whether the environmental "prior" is a meaningful construct, but mis-specified priors are definitely worth considering. I like the generality of the framework and of the methods that are developed. The article is very nicely written, especially in the way it makes it clear that the mathematics can be used in a suite of related problems. I have minor comments on the presentation:
- It's slightly bizarre at the start of Theorem 4.1 to suddenly have "there exists an algorithm explore(t)", when the previous paragraph is describing an explicit exploration algorithm. Noting the relationship between the presented strategy and the strategy of the theorem would be better.
- It is somewhat disappointing that the strategies being deployed in the meta-learning are all purely exploitative. This feels like we are therefore solving the "wrong" problem in some sense. Some discussion of this would be useful.

I have not checked the proofs. I am afraid that begin expected to review 36 pages of appendix in the timeframe of a conference submission is not reasonable. I actually recommend that the article should not be published here only because of this, and instead submitted to a journal where proper review can be carried out.

**Time Spent Reviewing:**

4

---

> ### Author Response · Authors · 2021-08-11
> **Response**
>
> We appreciate the reviewer’s comment, and we would prefer a peer-review system that afforded more thorough reviewing of longer articles. Unfortunately, journal readership seems to be somewhat lower amongst our target audience than does readership of conference proceedings, and so we elected to submit to this venue. The appendix, while lengthy, is not anomalous for theory submissions to NeurIPS and similar conferences.
>
> Re: the statement of Theorem 4.1. We will revise the language to reference the aforementioned exploration strategy.
>
> Re: exploitative strategies. Yes, it would be interesting to consider more adaptive exploration strategies at the meta level. However, the main focus of the paper is understanding prior sensitivity in Bayesian bandit algorithms, and we focused on exploitative strategies so that meta-learning results could be derived as direct corollaries of our sensitivity bounds.

---

### Official Review · Reviewer_A8Q1 · 2021-07-18

**Rating:** 7
**Confidence:** 3

**Summary:**

This paper studies the effect of prior misspecification in Thompson sampling (TS) and related posterior sampling bandit algorithms. They are able to show that the regret due to such algorithms using a misspecified prior incurs additional regret that is linear in the total variation distance between the improper and true prior.

Additionally, the paper considers estimating such priors in a meta-learning setting, where an agent interacts sequentially with multiple bandit instances whose model parameters are sampled from the same unknown prior. The authors provide bounds on the total variation distance for the estimated priors when the prior is Beta or Gaussian.


**Limitations And Societal Impact:**

The paper is primarily theoretical, providing a novel sensitivity bound that shows how regret of TS algorithms changes when the prior is misspecified. As alluded to earlier, I believe the impact of the paper would be greater if the authors were able to use the result to propose new TS algorithms that would be more robust to such misspecification. The authors point to this as a potential avenue for future research.

**Main Review:**

The paper considers an important problem in Thompson sampling with a misspecified prior. To my knowledge, this is a novel consideration in bandit literature, as prior work in studying TS algorithms always assumes the true prior is given. The authors are able to show that TS with a misspecified prior incurs regret that scales with its total variation distance from the unknown, true prior, which is an original result. The paper also considers meta-learning a prior using sequential interactions across multiple bandit episodes, and derives bounds on the total variation distance for Beta and Gaussian priors. They also demonstrate empirically that meta-learning the prior can outperform using an uninformative prior in various bandit settings.

Overall, I thought that the paper was organized clearly and made a solid and original contribution to Bayesian bandit literature. I do question the practicality of the proposed meta-learning approach as it relies on instances of forced exploration in order to estimate the prior parameters. I would have liked to see a theoretical result that showed that the total regret across multiple episodes outperforms not refining the prior across episodes but avoiding the need for forced exploration, i.e. show that meta-learning can achieve regret that is sub-linear in the number of episodes. In addition, I would be interested in some discussion on whether the sensitivity bounds can lead to new TS algorithms when the prior is misspecified, such as increasing the covariance of the misspecified prior can lead. Nevertheless, I believe the paper is a solid submission and recommend that it be accepted.


**Time Spent Reviewing:**

1.5

---

> ### Author Response · Authors · 2021-08-11
> **Reply re: regret**
>
> Thanks for the careful review of our paper! Regarding your question about showing a sublinear regret bound, note that Theorems 4.1 and 4.2 immediately give sublinear regret via an explore-then-commit strategy. So our results do theoretically demonstrate that meta-learning, even with some uniform exploration, outperforms not refining the prior.

---

### Official Review · Reviewer_2kq1 · 2021-07-18

**Rating:** 7
**Confidence:** 1

**Summary:**

This paper focus on the performance of the Thompson sampling algorithm under the misspecification of the prior distribution. The authors propose a general family of n-Monte Carlo algorithm, which contains the canonical TS algorithm as a special case. The authors then proceed to show that the regret difference of a misspecified prior up to horizon H of an n-MC algorithm is of order H^2*eps, where eps is the amount of misspecification measured by the total variation distance. The results are applied to the meta-learning setting, in which one also aims to learn the prior. The paper concludes by a synthetic numerical experiment to show the benefit of meta-learning.

**Main Review:**

- The paper is well-written. The exposition is clear and easy to follow. I appreciate that the authors provide enough explanation and sketch of the proof for me to follow the main text.
- The submission is particularly long (48 pages) and I could not go through all the details of the proof in the appendix. However, through a superficial read of the proof, I am inclined that the results are correct.
- I would appreciate if the authors can also discuss and contrast the (prior) misspecification bounds in this paper with the (model) misspecification bounds for the UCB algorithm (and other non-Bayesian algorithms).
- Can the authors provide any theoretical justification for the efficiency of learning the prior in the meta-learning setting (compared to using a uniform prior)?

**Time Spent Reviewing:**

8

---

> ### Author Response · Authors · 2021-08-11
> **Reply and Clarifications**
>
> Thank you for your suggestions, we will be sure to include them in the updated version. The two notions of misspecification are quite different and incomparable on a technical level. Indeed, in our setting, we assume that we have a well-specified model of rewards conditioned on the mean vector. In contrast, frequentist misspecification bounds relax this assumption. For example: (1) rewards may be changing or corrupted with adversarial noise; (2) the tail decay of the reward distribution is unknown; or (3) a simple function class (e.g. linear models) are used as an approximation to a harder-to-model reward function. The Bayesian analog is that the *likelihood* is misspecified (not the prior), and an interesting question for future work is to understand if we can be robust to both forms of misspecification simultaneously. We will include this discussion in the revision.
>
> Theorems 4.1 and 4.2 provide the sample complexity for learning some standard non-uniform priors. If by efficiency you are referring to a regret bound, then these theorems can be easily translated to yield sublinear regret bounds via a standard argument. However, we would like to emphasize that because we treat H as fixed and consider multiple episodes (where the base learner is restarted), the purpose of non-uniform priors is *statistical consistency*, not statistical efficiency. As we demonstrate in the experiments, using the correct prior (e.g. accounting for covariances or using the correct prior means), is essential to obtaining optimal performance. If one runs Thompson Sampling with a uniform prior (or some other misspecified prior), one will not converge to the best attainable performance, or equivalently, one will incur linear-in-T regret in the analogous online learning setting.

---

> > ### Comment · Reviewer_2kq1 · 2021-08-22
> > **Score lowered to discourage long submission**
> >
> > I thank the authors for the reply. I have lowered the score to 7. I agree that there should be a mechanism to discourage long papers from being submitted to conferences like neurips. There are also more appropriate venues such as JMLR, Mathematics of Operations Research or Operations Research, etc. where the authors may consider submitting their papers.

---

### Official Review · Reviewer_Dwdc · 2021-07-19

**Rating:** 8
**Confidence:** 3

**Summary:**

In this paper, the authors study how prior mis-specification would impact the performance of bayesian bandit algorithms. In particular, the authors provide an upper and a lower bound for a wide class of algorithms, which include Thompson sampling algorithms as a special case. The results are complemented by applications in meta learning and numerical experiments.

**Limitations And Societal Impact:**

N.A.

**Main Review:**

The results in this paper are novel to the best of my knowledge. Prior sensitivity is a relatively open area in the analysis of Bayesian bandit algorithms. Since Thompson sampling algorithm, a particular Bayesian bandit algorithm, is widely used in many applications such as ads display and dynamic pricing, the results in this paper could be of broad interest.

Some comments for improvement. It looks like the key step in showing the upper bound is Proposition 3.4, it would thus be helpful if the proof sketch/intuition of this Proposition could be clearly provided.

Also, the coupling Q plays an important role in the proof of upper bound because the history of the two versions (one with correct prior one without) of the algorithm could have different history. I would suggest the authors to deliver a more constructive way to establish Q in appendix B.

**Time Spent Reviewing:**

1

---

> ### Author Response · Authors · 2021-08-11
> **Reply: re proof sketches**
>
> Thank you for your suggestions, we will be sure to include them (in particular a proof sketch of Prop 3.4) in the updated version. Regarding the coupling: the coupling in the proof may be somewhat intricate to construct, which is why we opt for the non-constructive version. However, we will provide an illustration of how such a coupling might look to expose why the coupling argument is necessary (which is precisely for the reason you describe: that two independent runs would generate different histories).

---

### Official Review · Reviewer_Du9Y · 2021-07-21

**Rating:** 7
**Confidence:** 4

**Summary:**

The paper establishes the theory for quantifying the sensitivity of a class of Bayesian bandit algorithms under misspecified prior, with Thompson sampling being a special case. It is also shown that the sensitivity bound is tight in the worst-case sense. Using the sensitivity analysis, the paper studies the sample complexity of Bayesian meta-learning in two special cases (Beta prior+Bernoulli reward and Gaussian prior+Gaussian reward). Finally, the aforementioned theory is generalized to the more general setting, the Bayesian POMDPs.



**Limitations And Societal Impact:**

A discussion on the potential negative societal impact of the work may be added.

**Main Review:**

This is a well-written paper with clear exposition, rigorous proof, and extensive simulations. My question is as follows:

- the authors claim that the theory is designed for meta-learning problems with short task horizons, but the lower bound seems to hold in a large-horizon regime. Is that correct?


**Time Spent Reviewing:**

4 hours

---

> ### Author Response · Authors · 2021-08-11
> **Reply Re: Lower Bound Horizon**
>
> Yes, that is correct, the upper and lower bounds apply for long horizons as well. However, the lower bound requires the two priors to be constructed as a function of the horizon H. On the other hand, long horizons are typically studied in the regime where H grows for a fixed prior. Fixing the prior and lengthening the horizon H will result in a O(sqrt{H})-convergence to the optimal action, via standard Thompson sampling regret. We can clarify this in the revision.

---

### Decision · Program_Chairs · 2021-09-27

**Decision:**

Accept (Spotlight)

**Comment:**

Thank you to the authors and the reviewers for their contributions to the conference! This paper introduces a novel sensitivity analysis of Bayesian bandit algorithms such as Thompson Sampling under misspecified priors. Misspecification is clearly an important problem in practice, and the review team unanimously appreciated the paper. I am happy to recommend acceptance.